# Autologous neutralizing antibodies and polyfunctional T cells contribute to long-term HIV-1 post-intervention control

Antiretroviral therapy (ART) interruption typically leads to rapid HIV-1 viral rebound in people with HIV-1. To develop an HIV-1 cure, insight into immunological mechanisms capable of preventing HIV-1 viral rebound is urgently needed. Here, we describe three exceptional post-intervention controllers (PICs) who maintained ART-free virological control for >6.5 years (ongoing), >7.5 years (ongoing) and 2.5 years following administration of broadly neutralizing antibodies. PICs had quantifiable genetically intact/inducible infectious proviral reservoirs that were increasingly clonal and located in nongenic/centromeric chromosomal regions, indicating immune-mediated selection. Potent autologous neutralizing antibodies and polyfunctional HIV-1-specific CD4[+] and CD8[+] T cell responses, pre-programmed for antigen response, were present before, and persisted during, ART interruption. In one PIC, viral rebound following 2.5 years of ART-free control was associated with accumulated viral mutations that resulted in escape from neutralizing antibody and T cell responses. Collectively, our findings support developing HIV-1 curative strategies aimed at enhancing pre-existing adaptive immune responses.

People living with HIV-1 (PLWH) must remain on lifelong ART to suppress HIV-1 replication, as plasma viral load rebounds within 2–4 weeks following ART interruption for most PLWH[1,2]. PICs maintain low or undetectable HIV-1 viral loads following ART interruption and administration of therapeutic interventions, such as broadly neutralizing anti-HIV-1 antibodies (bNAbs)[3], modeling ART-free virological control. Recent studies indicate that bNAb treatment in PLWH at ART initiation[4] or at ART interruption[5–10] leads to a substantial delay to or prevention of viral rebound despite bNAb concentrations dropping below suppressive levels[4–6,9,10]. In fact, the frequency of PICs following bNAb administration is estimated to be 10–20%, compared to a post-treatment control (PTC) frequency of 4% for PLWH who interrupt ART without therapeutic interventions[2]. Collectively, these observations suggest that bNAb administration can augment immune responses to effectively mediate ART-free virological control.

Despite extensive work characterizing factors contributing to ART-free virological control[11–15], the exact mechanisms that mediate

viral suppression remain elusive, and the role of multiple immunological and/or virological facets in the same individuals is largely unknown. Here, we performed an extensive characterization of the virological and HIV-1-specific immunological characteristics behind the control of viremia observed in three PICs.

## Results

Three PICs were identified following administration of bNAbs and subsequent analytical treatment interruption (ATI) as part of previously reported clinical trials[4,5,10]. All three PICs started ART relatively early following HIV-1 acquisition (Supplementary Table 1), and maintained long-term ART-free virological control, even after bNAb levels had dropped below suppressive levels (Supplementary Table 1 and Supplementary Fig. 1a–c). ID107 has remained off-ART with undetectable viral loads (<20 copies per ml) for 6.5 years (334 weeks ongoing), except for one minor blip (29 copies per ml; Fig. 1a), while ID9254 has maintained ART-free control for 7.5 years (388 weeks ongoing), except

✉ e-mail: olesoega@rm.dk

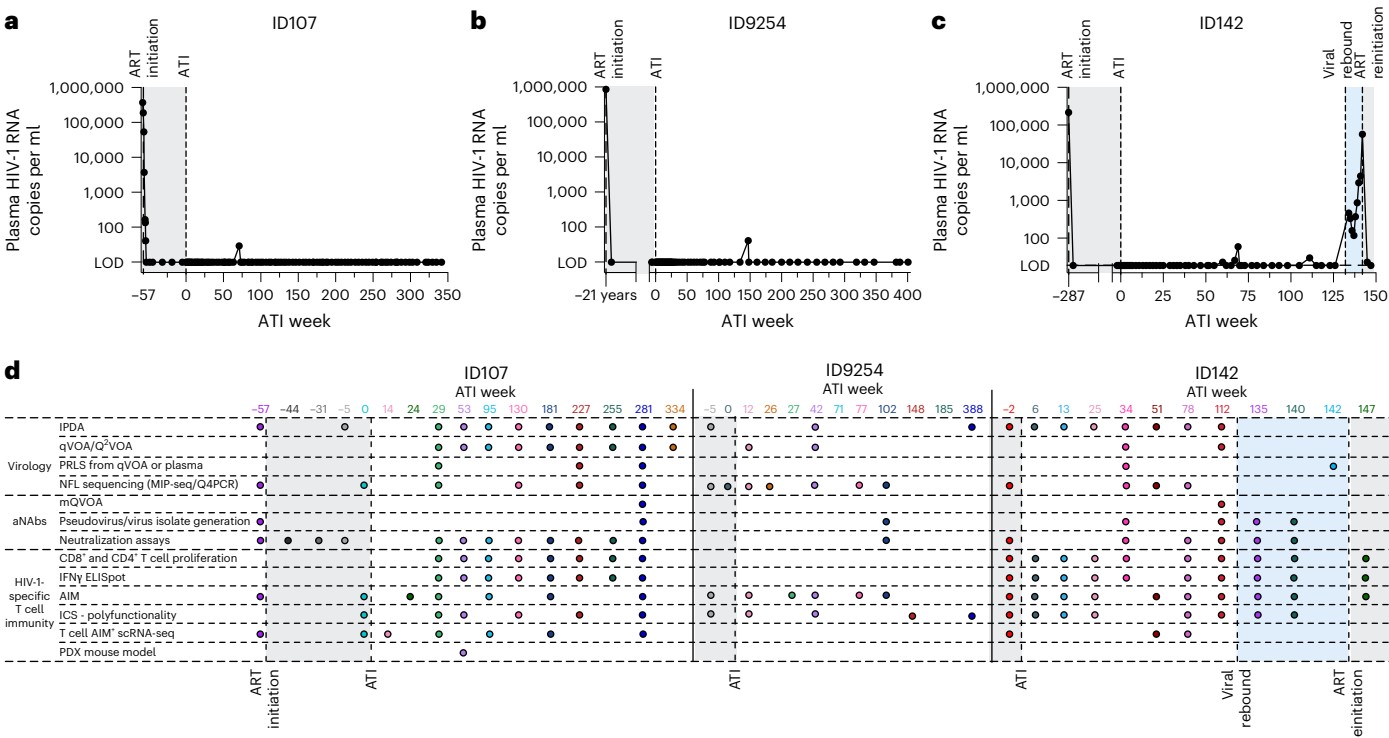

**Fig. 1 | Summary of assays performed. a–c**, Plasma HIV-1 RNA viral loads (copies per ml) for ID107 (**a**), ID9254 (**b**) and ID142 (**c**) relative to time following ART interruption (weeks). Gray shading indicates time on ART, blue shading indicates viremic time points for ID142. Limit of detection is 20 HIV-1 RNA copies per ml. **d**, Virological and immunological assays performed at each timepoint following ART interruption. Each assay performed is represented by a colored circle, relative to the time point in weeks following ART interruption. ICS, intracellular cytokine staining; mQVOA, modified qVOA; PRLS, plasma-derived HIV-1 RNA using long-range sequencing; Q4PCR, quadruplex quantitative PCR; SGS, single-genome sequencing; scRNA-seq, single-cell RNA sequencing.

for one minor blip (41 copies per ml; Fig. 1b). ID142 discontinued ART and maintained undetectable viral loads for 2.5 years (130 weeks), apart from two minor blips (Fig. 1c); however, ID142 reinitiated ART after experiencing viral rebound that reached 57,600 copies per ml at ATI week 142. All PICs had high viral loads before initiating ART and did not have HLA-I alleles B*27 or B*57, which have been previously associated with spontaneous control of HIV-1 (ref. 16) (Supplementary Table 1). All individuals had longitudinal cryopreserved peripheral blood mononuclear cells (PBMCs) available from pre-ATI throughout ATI, and we therefore sought to characterize the mechanisms behind long-term ART-free control of HIV-1 using a large selection of virological and immunological investigations (Fig. 1d).

**All PICs have a detectable genetically intact viral reservoir**

We first quantified the genetically intact proviral reservoir at multiple time points before and following ART interruption using a modified intact proviral DNA assay (IPDA)[17,18]. All three PICs had a genetically intact proviral reservoir that fell well within the range observed in individuals on suppressive ART[4,5] (Fig. 2b), indicating that low or undetectable genetically intact proviral reservoirs were not the cause of long-term virological control in these PICs.

**Genetically intact proviruses are increasingly clonal and located in nongenic/centromeric regions in PICs following ATI**

While the IPDA estimates the maximal size of the intact proviral reservoir, additional factors such as the integration site of genetically intact proviruses may also impact a provirus' potential to reactivate and produce virus following ATI[19,20]. We therefore used matched integration site and proviral sequencing (MIP-seq)[21] to identify genetically intact proviruses and their corresponding integration sites for ID107 and ID142 (Fig. 2c and Supplementary Table 2). For ID107, we observed that 50% of all intact proviruses identified at the pre-ART

time point (week −57) were integrated into genic regions, compared to 16% found in nongenic regions (Fig. 2d). Additionally, most proviruses with matching integration sites were only identified once and were not defined as clonal (Fig. 2c). By contrast, we observed a progressively increasing dominance of clonal intact proviruses integrated within nongenic, and specifically centromeric, genomic regions in the ATI time points (Fig. 2c,d and Supplementary Table 2). In fact, by ATI week 281, 93% of identified intact proviruses were found in a nongenic region (Fig. 2d). Many of these intact proviruses with identical integration sites were identified at multiple time points, indicating their clonality and long-term persistence during the ATI (Fig. 2c).

For ID142, despite the larger intact proviral reservoir detected by IPDA, only a small number of intact proviruses could be isolated by MIP-seq (Fig. 2c). One clone of intact proviruses was found to be integrated in a nongenic 'gene desert' region, and no integration sites could be determined for the other intact proviruses (Fig. 2c and Supplementary Table 2). These results may suggest that, at least during the period of virological control, intact proviruses for ID142 were integrated in highly inaccessible regions of the genome such as dense heterochromatin, for which chromosomal integration site identification is also difficult.

For ID9254, genetically intact proviruses generated by quadruplex qPCR (Q4PCR)[22] indicated that the genetically intact proviral reservoir in this individual was of a very low genetic diversity and was also dominated by large groups of genetically identical proviral sequences, suggestive of these proviruses also having identical integration sites (Extended Data Fig. 1). In support of this, previous integration site profiling of sampling from the pre-ATI time point for ID9254 indicated a dominance of the inducible proviral reservoir by proviruses with identical integration sites[23].

We conclude that genetically intact proviruses persisting during long-term ART-free control showed increasing clonality, low genetic

diversity, and integration into genomic regions less likely to support spontaneous proviral expression.

## Inducible proviruses persist during ART interruption

We next performed quantitative viral outgrowth assays (qVOA) during ATI for ID107 and ID142 to investigate whether the dominance of intact proviruses in transcriptionally silent chromosomal regions impeded their inducibility. For ID107, inducible virus was detected at all time points, ranging in size from 0.148 to 0.949 infectious units per million (IUPM) CD4[+] T cells (Fig. 2e), with no decrease in frequency over time. ID142 had a smaller but still inducible reservoir, ranging from 0.045 to 0.05 IUPM CD4[+] T cells (Fig. 2e). For ID9254, we quantified the reservoir of inducible infectious proviruses using the similar quantitative and qualitative viral outgrowth assay (Q²VOA)[24], and found that the size of the reservoir was larger than the other PICs, and ranged from 1.89 to 2.15 IUPM (Fig. 2e).

Next, we performed near-full-length (NFL) sequencing[25] of viral genomes present in positive qVOA cultures from ATI weeks 29, 227 and 281 for ID107, and from ATI week 34 for ID142, and compared these qVOA viruses to the proviral sequences generated by MIP-seq. For ID107, we found that three genetically intact proviral genomes matched HIV-1 RNA genomes isolated from qVOA wells (Fig. 2f). For ID142, both viral sequences sourced from qVOA wells were identical to a group of clonal genetically intact proviruses integrated within a nongenic region (Fig. 2g). Altogether, these results indicate that all PICs had reservoirs of inducible infectious proviruses that were not decreasing, and that theoretically could contribute to viral rebound in vivo.

## Residual viremia is detected during ART-free control

Next, we investigated whether residual viremia (viral load <20 copies per ml) could be detected despite lack of viral rebound. We sequenced 19 partial *env* single genomes (969 bp; HXB2 7008-8009) for ID107 from plasma sourced at ATI week 281, though we could not identify the proviral source of these plasma *env* genomes (Supplementary Fig. 2). These results indicate that viral antigens are produced during the ATI despite lack of viral rebound. Of note, the level of plasma inflammatory markers remained unchanged or were even lower after stopping ART compared to the on-ART time point (Supplementary Fig. 1d,e).

## Autologous neutralizing antibodies inhibit exponential viral outgrowth

We next investigated if low concentrations of contemporaneous autologous IgG antibodies could inhibit outgrowth of infectious virus using modified qVOAs[26]. For both ID107 and ID142, in the presence of 50 µg ml⁻¹ autologous IgG, no viral outgrowth was observed, whereas in the presence of HIV-negative donor IgG, exponential outgrowth was observed (Fig. 3a,b). This indicates that autologous IgG from ID107

and ID142 can inhibit outgrowth of contemporaneous infectious virus, suggesting a role for autologous neutralizing antibodies (aNAbs) in preventing viral rebound in vivo.

## Autologous IgG neutralizes inducible, infectious virus

Next, for ID107 and ID142, we directly assessed the potency of aNAbs against infectious virus by generating pseudoviruses expressing selected HIV-1 Env trimers (Extended Data Fig. 2). To evaluate antibody potency, we calculated both half-maximal inhibitory concentration (IC$_{50}$) and instantaneous inhibitory potential (IIP) values following incubation of pseudoviruses with serial dilutions of autologous IgG in direct TZM-bl neutralization assays (Supplementary Fig. 3a–c). IIP is the log reduction in single round infection events at a given antibody concentration, and can provide an estimate of in vivo efficacy of autologous antibodies at physiological concentrations[27,28].

For ID107, three pseudoviruses were generated from qVOA variants isolated from ATI week 281 (Extended Data Fig. 2a). Notably, autologous IgG obtained from the pre-ART time point (week −57) had low neutralizing capacity (IC$_{50}$ > 100 µg ml⁻¹), but aNAb potency stabilized and matured at 0.39–0.43 µg ml⁻¹ by 12 months on ART (Fig. 3c). During ATI, ID107 aNAbs consistently exhibited high levels of inhibition against all three pseudoviruses with IIP values at 10 mg ml⁻¹ values ranging from 5.6 to 7.3 (Fig. 3d). Notably, this level of inhibition against replication-competent virus is comparable to that of current combination ART regimens[27–29] (Supplementary Fig. 3e).

For ID142, three pseudoviruses representing the pre-rebound inducible proviral population were generated from qVOA variants from ATI week 112 (Extended Data Fig. 2b). These pseudoviruses were tested against autologous IgG purified from pre-ATI and ATI time points pre- and post-rebound (Fig. 3e). We found that autologous IgG from all time points potently neutralized replication-competent virus sourced from before rebound, with a mean IIP value at 10 mg ml⁻¹ of 6.82 (Fig. 3f).

For ID9254, we performed similar TZM-bl cell-based neutralization assays using autologous IgG sourced from ATI week 102, but with neutralization assessed against 19 replication-competent viral isolates sourced from Q²VOA experiments at ATI week 102. Replication-competent viral isolates are known to be typically more difficult to neutralize than pseudoviruses, even by potent bNAbs[30,31]. Regardless, we similarly observed that autologous IgG was able to neutralize contemporaneous replication-competent virus, with IC$_{50}$ values in the range of 10.8–23 µg ml⁻¹ (Fig. 3g).

Of note, for all three PICs, we observed no evidence of cross-neutralization capacity of the aNAbs against panels of pseudoviruses representing global HIV-1 variants, a different PIC's autologous virus and/or the laboratory-strain virus HXB2 (Supplementary Fig. 4 and Supplementary Tables 3 and 4). This indicates that the autologous antibodies are extraordinarily specific to the viral quasispecies of

---

**Fig. 2 | Genetically intact/inducible infectious proviruses persist during long-term ART-free virological control. a**, Summary of assays performed to characterize genetically intact and inducible infectious proviral reservoirs over time. Created in BioRender. Fisher, K. https://BioRender.com/tgthm7o (2026). **b**, Frequency of genetically intact proviruses/10⁶ CD4[+] T cells was quantified using a modified IPDA for ID107, ID9254 and ID142, and compared to those of two cohorts of ART-suppressed individuals[4,5] (eCLEAR and TITAN; median IPDA values highlighted by dashed lines). Each data point represents a single replicate per time point (longitudinal biological replicates within one individual). **c**, Circos plot highlighting individual genetically intact proviruses identified using MIP-seq for ID107 and ID142. Each square represents a single intact provirus, color-coded according to time point, and clones are identified by ribbons joining individual squares. Identified integration sites are shown next to the relevant intact provirus. Chromosomal coordinates are indicated using the Hg38 reference genome nomenclature. **d**, Pie charts representing the proportion of genetically intact proviruses isolated at each time point for ID107 according to the identified chromosomal region. **e**, Inducible infectious proviruses were quantified using the qVOA/Q²VOA as IUPM CD4[+] T cells through time following

ATI for ID107, ID9254 and ID142. Individual error bars represent the upper and lower limits for IUPM, calculated using limiting dilution analysis based on the number of cells assayed in each experiment and the number of positive qVOA or Q²VOA cultures as previously described[66]. All IUPM values were derived from a single qVOA experiment per individual and time point (longitudinal biological replicates within one individual) consisting of multiple culture wells, with each culture well assaying 50,000 CD4[+] T cells for the qVOA and 200,000 CD4[+] T cells for the Q²VOA. Each experiment assayed at least 30 × 10⁶ CD4[+] T cells in total (number of CD4[+] T cells assayed per experiment listed in Source Data). **f,g**, Maximum likelihood phylogenetic trees showing all genetically intact proviruses isolated by MIP-seq and inducible infectious HIV-1 RNA genomes sequenced from positive qVOA cultures for ID107 (**f**) and ID142 (**g**). Groups of sequences in boxes indicate genetically identical proviral and qVOA sequences. Identified integration sites are shown next to the relevant provirus. Scale bars represent nucleotide substitutions per site. Asterisk indicates branch support value > 70%. Gray shading indicates time on ART, integration sites in genic regions are blue, in nongenic/centromeric regions are green, and not identified are gray. Analyzed samples are color-coded according to time point.

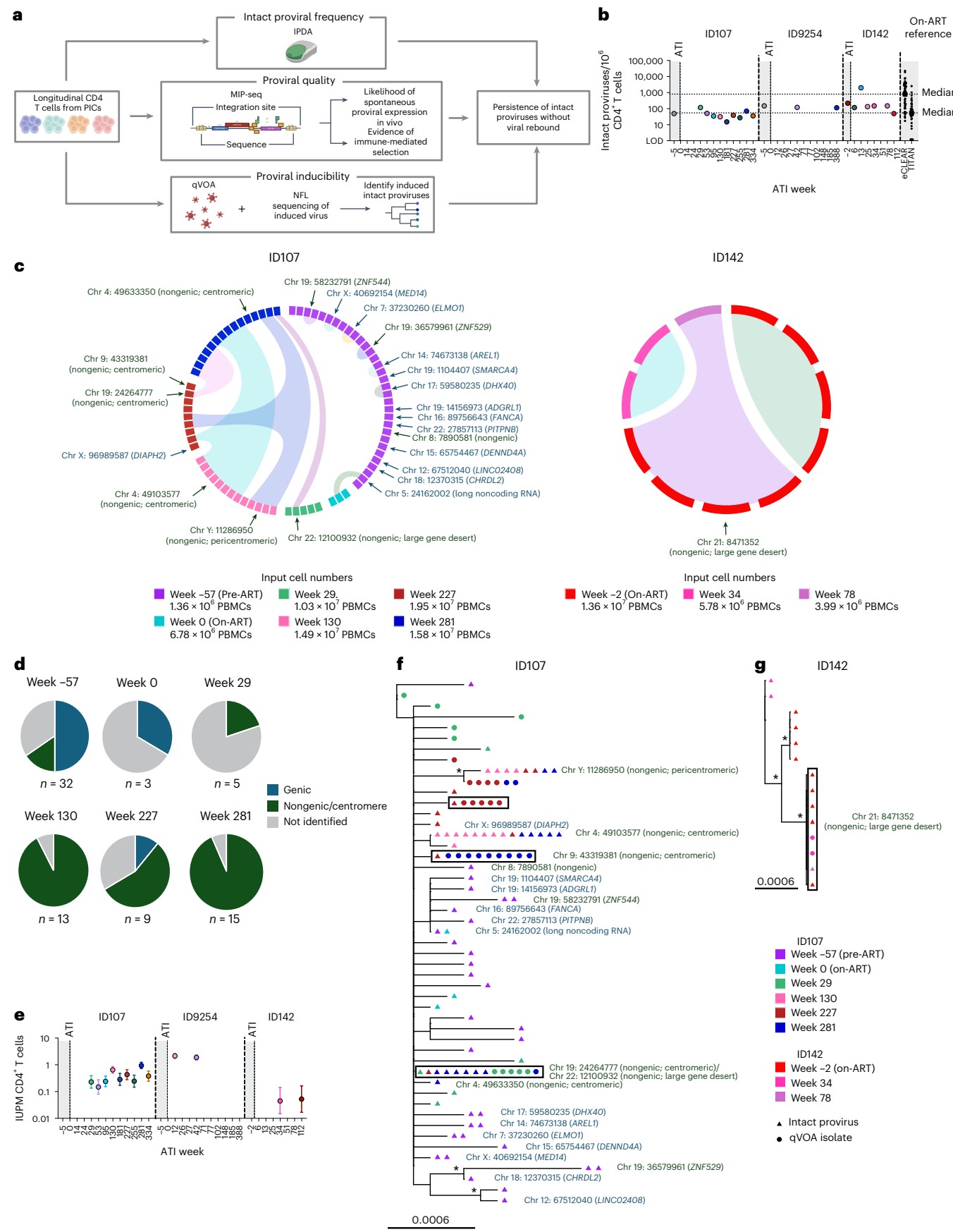

each individual. Altogether, these results indicate that potent and specific aNAbs contribute to the maintenance of ART-free control in all three PICs.

## HIV-1-specific CD4+ and CD8+ T cell responses are highly polyfunctional

Next, we used a spectral flow cytometry-based intracellular cytokine staining assay to characterize HIV-1-specific memory CD4+ and CD8+ T cell responses to stimulation with HIV-1 peptide pools representing Gag, Pol, Nef and Env (Fig. 4). For comparison, six individuals on suppressive ART were included, four of which also participated in the ATI during the eCLEAR trial[4] (Supplementary Table 5; detailed analysis of HIV-1-specific cells found in Extended Data Fig. 3a). We included individuals on suppressive ART for comparison to PICs ensure that comparisons of the frequency and characteristics of HIV-1-specific T cells are not biased by the presence of viremia in noncontrollers (NCs) following ART interruption.

First, we characterized differences in the frequency of monofunctional and polyfunctional HIV-1-specific T cells responding to HIV-1 Gag, Pol and Nef stimulations between PICs and ART-suppressed individuals at the on-ART time point (ATI week 0) (Fig. 4a). PICs showed higher frequencies of polyfunctional responses than ART-suppressed individuals, with the difference in CD4 memory responses reaching statistical significance ($P = 0.024$; Fig. 4a).

Stratification by HIV-1 protein revealed that polyfunctional CD8+ T cell responses in PICs, primarily expressing IFNγ, TNF and CD107a, were largely directed against a single HIV-1 protein, which differed among individuals (Fig. 4b,c). Specifically, ID107 exhibited a predominantly Gag-specific polyfunctional CD8+ T cell response, which declined during 6-year ATI, whereas ID9254 exhibited a predominantly Env-specific polyfunctional CD8 response, which increased following ART interruption and remained stable through 7 years off-ART (Fig. 4b,c). ID142 exhibited a predominantly Pol-specific polyfunctional CD8+ T cell response which remained stable during 2.5-year ATI (Fig. 4b,c). These observations were supported by T cell proliferation assays, IFNγ enzyme-linked immunosorbent spot assay (ELISPOT) assays and the activation-induced marker (AIM) assay (Extended Data Figs. 3b–e and 4).

All PICs exhibited a broader polyfunctional CD4+ T cell response toward Gag, Pol and Nef (Fig. 4b,c), with a significantly higher frequency of Gag-specific responses driving the overall increase in polyfunctional CD4+ T cell responses compared to ART-suppressed individuals at the pre-ART interruption time point ($P = 0.024$; Fig. 4b). These polyfunctional CD4+ T cells in PICs mostly expressed IFNγ, TNF and IL-2 (Fig. 4c). Altogether, we speculate that HIV-1-specific polyfunctional CD4+ and CD8+ T cells with enhanced cytokine production capacity at the pre-ART

interruption time point may be essential for maintaining long-term virological control.

## Pre-programming of HIV-1-specific CD8+ T cells before ART interruption supports rapid response to antigen exposure

To dissect how HIV-1-specific T cell responses evolved over time, we performed single-cell transcriptome and T cell receptor (TCR) sequencing on AIM+ sorted CD3+ T cells after stimulation with a pool of HIV-1 Gag, Pol and Nef peptides for ID107 and ID142 at time points pre-ATI and during ATI (Supplementary Fig. 5). The combined AIM+CD8+ and CD4+ T cell responses to HIV-1 Gag, Pol and Nef through time is shown in Extended Data Fig. 4g,h. For comparison, we included two non-controller individuals at the pre-ART, pre-ATI and viral rebound time points (ID104 and ID112; Supplementary Fig. 5 and Supplementary Table 5). Following dimensionality reduction, batch-correction and clustering at a resolution of 0.7, 11 cell clusters were identified and manually annotated (Fig. 5a).

As ID107 and ID142 showed particularly strong CD8 responses, we focused on the CD8 compartment. Our transcriptome analysis suggested substantial differences in activation states (Extended Data Fig. 5a). Clusters 1 and 3 showed the strongest signature of TCR-mediated activation, with an upregulation of genes associated with TCR-signaling (*EGR2*, *NR4A2* and *NFKB1*), co-stimulatory receptors (*CRTAM* and *TNFRSF9*) and activation-induced chemokines (*XCL1*, *XCL2* and *CCL4*) (Extended Data Fig. 5a). Many of these genes have previously been associated with antigen-responsive CD8+ T cells[32]. To further substantiate this, we calculated a module score for each cell based on a subset of these genes that has been previously used to quantify virus-specific activation[33]. The resulting scores confirmed that the coordinated expression of these genes was highest in these two clusters (Fig. 5b,c). Altogether, these data suggest that clusters 1 and 3 comprise CD8+ T cells that have recently engaged with their cognate antigen through the TCR and therefore are most likely HIV-1-specific. This was supported by surface protein profiling in PICs by antibody-derived tags (ADTs; Extended Data Fig. 5b and Supplementary File 1), which identified high expression of the early activation marker CD69 and, notably, the highest mean levels of the degranulation marker CD107a within clusters 1 and 3 (Fig. 5d and Extended Data Fig. 5c).

We next investigated whether ID107 and ID142 harbored higher abundances of these transcriptionally defined clusters compared to non-controllers. Of note, while there was no evidence of differential abundance during ATI (Extended Data Fig. 5d), PICs had significantly higher frequencies of cluster 1 at the beginning of ATI (adjusted $P = 0.002$; Fig. 5e). Frequencies of cluster 3 were also increased among PICs, although the difference did not reach statistical significance (adjusted $P = 0.134$; Fig. 5e).

---

**Fig. 3 | aNAb responses contribute to long-term ART-free virological control. a**, Viral outgrowth in qVOA wells from ID107 at ATI week 281 cultured with HIV-negative donor IgG (50 μg ml⁻¹) or contemporaneous autologous IgG (50 μg ml⁻¹). Each circle represents a single qVOA well and open circles indicate wells with no detectable outgrowth. **b**, Viral outgrowth in qVOA wells from ID142 at ATI week 113 cultured with no IgG, HIV-negative donor IgG (50 μg ml⁻¹) or contemporaneous autologous IgG (50 μg ml⁻¹). **c**, Dose–response curves for pseudoviruses ID107.QVOA.1, ID107.QVOA.2 and ID107.QVOA.3. Pseudoviruses were tested against dilutions of autologous IgG antibodies (10 ng ml⁻¹ to 100 μg ml⁻¹) purified from longitudinal time points (see color keys; one IgG sample per time point). Data represent mean ± s.d., based on two independent neutralization experiments each containing three replicates of culture wells (six total data points) containing pseudovirus and TZM-bl cells for each IgG concentration. For each experiment, control wells were included containing pseudovirus and TZM-bl cells with no IgG (six wells per experiment) and TZM-bl cells with no pseudovirus or IgG (six wells per experiment). The *env* sequences used to generate pseudoviruses ID107.QVOA.1, ID107.QVOA.2 and ID107.QVOA.3 are shown in Extended Data Fig. 2a. **d**, IIP values for ID107 at 10 mg ml⁻¹ of

autologous IgG are calculated for longitudinal time points for each pseudovirus and represented on the right *y* axis. The dashed line at IIP = 5 represents the threshold for effective suppression of in vivo viral replication, comparable to current ART regimens[27,28]. **e**, Dose–response curves for pseudoviruses ID142. QVOA1, ID142.QVOA2 and ID142.QVOA3. Pseudoviruses were tested against autologous IgG purified from longitudinal time points from pre- and post-rebound time points (color keys; one IgG sample per time point). Data represent mean ± s.d., based on three replicates of culture wells containing pseudovirus and TZM-bl cells for each IgG concentration. Control wells were also included as described in **c**. The *env* sequences used to generate pseudoviruses ID142.QVOA1, ID142.QVOA2 and ID142.QVOA3 are shown in Extended Data Fig. 2b. **f**, IIP values for ID142 at 10 mg ml⁻¹ of autologous IgG are calculated for longitudinal time points for each pseudovirus and represented on the right *y* axis. **g**, Neutralization dose–response curve for ID9254 IgG sourced from time point ATI week 102 against 19 replication-competent viral isolates sourced from viral outgrowth assays performed at ATI week 102. Each data point is based on duplicate wells of each IgG dilution and virus isolate, with error bars representing ± %CV.

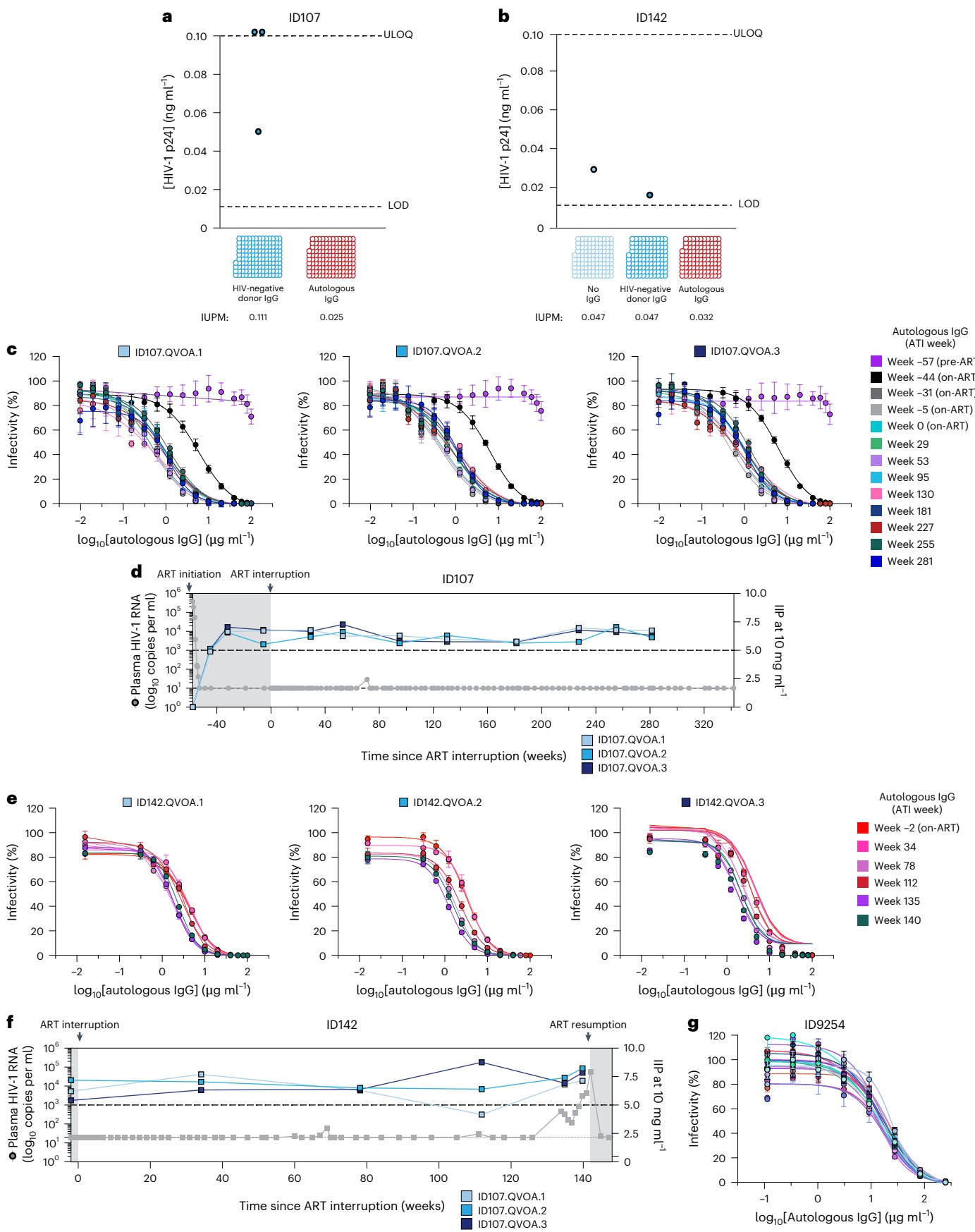

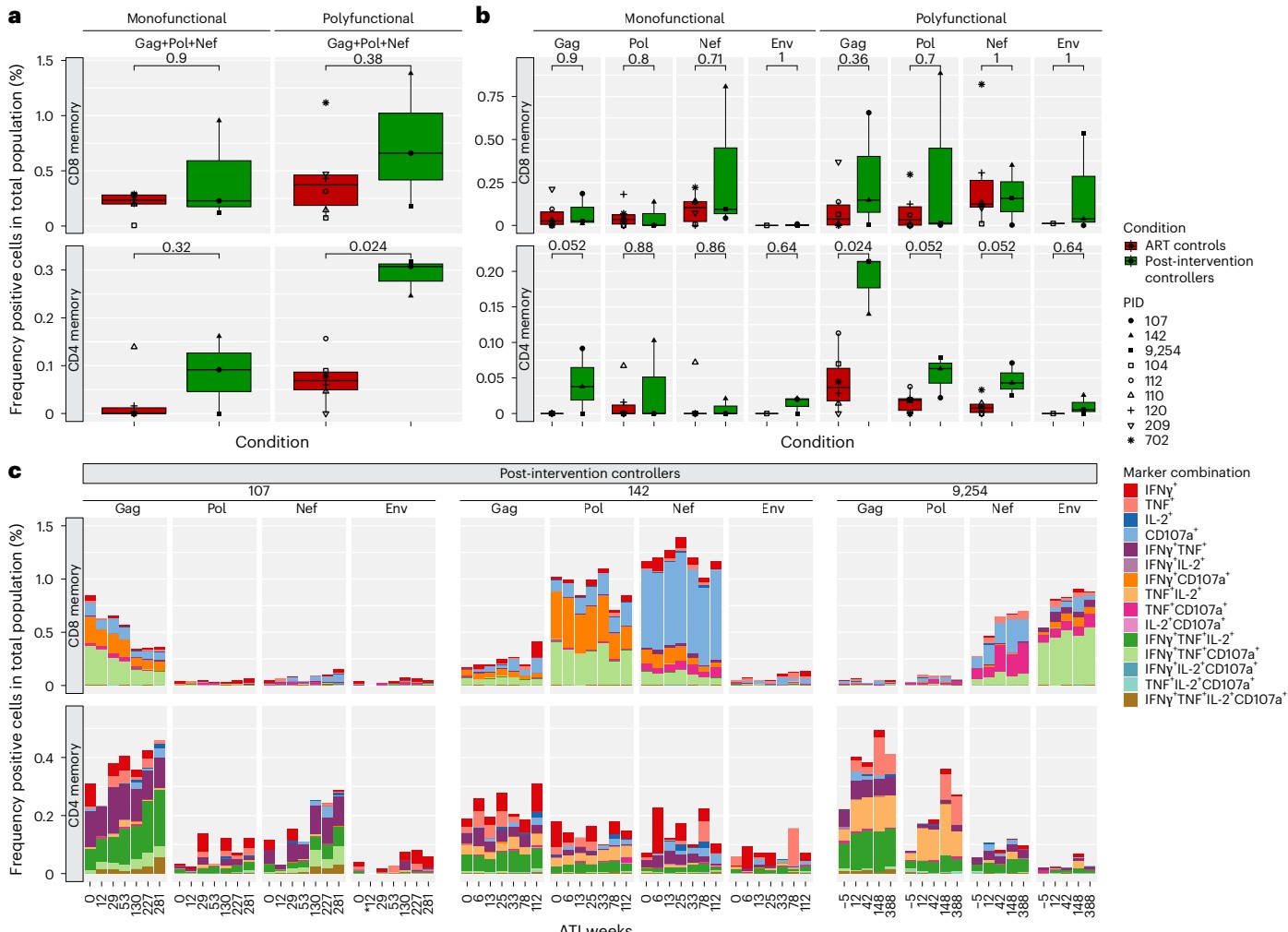

**Fig. 4 | HIV-1-specific CD8⁺ and CD4⁺ T cell responses in PICs are highly polyfunctional during long-term ART-free control.** Spectral flow cytometric assessment of cytokine production by memory CD8⁺ T cells (top) and memory CD4⁺ T cells (bottom) (Methods) in response to stimulation with HIV-1 Gag, Pol, Env or Nef peptide pools in PICs and ART-suppressed individuals. Asterisk denotes excluded data due to insufficient number of cells in the PBMC vials. **a,b**, Direct comparison of the percentage of memory CD4⁺ and CD8⁺ T cells responding to stimulation with a monofunctional response or a polyfunctional response at the ART-suppressed, pre-ART interruption timepoint, for Gag, Pol and Nef (as cells from some ART-suppressed individuals were not stimulated with Env peptides) stimulation combined (**a**) or stratified by HIV-1 peptide pool stimulation (**b**). Cells with a polyfunctional response were defined as those positive for two or more of the markers IFNγ, TNF, IL-2 and CD107a. Center line of box plots represents the median, edges of the box plot represent first and third quartiles, whiskers represent ±1.5 × IQR. To compare the frequencies of mono- and polyfunctional memory CD4⁺ and CD8⁺ T cells between PICs and ART controls (*n* = 3 and 6 biological replicates, respectively), two-sided, unpaired, nonparametric Wilcoxon signed-rank tests with a Bonferroni correction for multiple comparisons were used. Each data point represents a single experiment per individual and time point, utilizing a minimum of 0.8 × 10⁶ PBMCs per stimulation. Each data point is calculated by subtracting the value of responding cells to the specific stimulation by that within the negative control (stimulation of cells with dimethylsulfoxide (DMSO)). **c**, Specific cytokine production by responding CD8⁺ and CD4⁺ T cells in PICs ID107, ID9254 and ID142 at multiple time points before and after ART interruption. Color denotes positivity for cytokines and degranulation (CD107a), and the eight dominant responses are highlighted in bold in the legend.

Next, we investigated specific features distinguishing cluster 1 from clusters 3 and 10, which had the highest median virus-specific activation module scores (Fig. 5b) and the most similar transcriptional signatures (Extended Data Fig. 5a), to identify how high frequencies of cluster 1 may contribute to ART-free control in PICs. Cluster 1, like cluster 3, had higher expression of genes related to TCR-mediated effector responses and cytotoxic functions compared to cluster 10 (Fig. 5f,g). In addition, cluster 1, like cluster 10, showed increased expression of genes linked to ribosomal biogenesis and mitochondrial activity, as well as cell cycle- and DNA replication-associated genes, when compared to cluster 3 (Fig. 5g). These results suggest that cells in cluster 1 have increased biosynthetic and metabolic activity while undergoing differentiation processes that culminate in proliferation. In support of this, ADT data showed increased surface expression of CD25 (IL-2RA) and CD71 in

cluster 1, both of which are associated with activated, proliferating T cells[34] (Fig. 5d and Extended Data Fig. 5c). Collectively, these findings suggest that cluster 1 encompasses cells capable of both immediate cytotoxic and cytokine responses, and of proliferative expansion. Altogether, these findings suggest that ID107 and ID142 maintained a population of CD8⁺ T cells at pre-ATI primed to rapidly activate and differentiate upon HIV-1 antigen encounter.

## Single-cell TCR repertoire analysis of AIM⁺ T cells suggests that stable, oligoclonal T cell expansion may support virological control in ID107 and ID142

Next, we investigated the TCR repertoire of the AIM⁺ T cells. To compare the distribution of expanded clonal families across clusters, we ranked the clonotypes by their total cell counts across all visits, separately for

each participant. Of note, the top ten clonotypes consistently dominated the repertoire in clusters 1 and 3, as well as cluster 9, which had a similar effector profile to cluster 3 (Extended Data Fig. 5a), in ID107 and ID142, accounting for 49–92% of TCRs (Extended Data Fig. 6a). We hypothesized that this pattern could indicate that these clusters were highly clonal and interconnected through shared clonotypes.

We next calculated the median Gini coefficient to quantify the degree of clonality within each cluster and donor[35] (Extended Data Fig. 6b). We found that the median Gini coefficient values in clusters 1, 3 and 9 among ID107 and ID142 ranged between 0.47 and 0.76 (Extended Data Fig. 6b), indicating high TCR clonality. To quantify the overlap of TCRs between clusters, suggesting shared clonotypes, we calculated the Morisita–Horn index for cluster pairs for each donor[35] (Extended Data Fig. 6c). These results indicated consistent overlap between clusters 1, 3 and 9 in ID107 and ID142, and in non-controllers. Further, we found that the largest clones for ID107 and ID142 were already prevalent at pre-ATI and persisted throughout the ATI (Fig. 5h). By contrast, the clones that were expanded pre-ART among non-controllers and re-emerged at rebound were rare at pre-ATI (Fig. 5h).

Altogether, these findings show that oligoclonal TCR sequences, shared across clusters but not between participants, dominated the previously identified effector clusters that were abundant in ID107 and ID142. The repeated identification of these clonotypes as prevalent clones in AIM+ sorted T cells indicates that these cells may enable PICs to more efficiently mount a robust HIV-1-specific immune response upon ART interruption.

### HIV-1-specific CD8+ T cells derived from ID107 suppress viral replication in a HIV-1 participant-derived xenograft model

We next utilized a participant-derived xenograft (PDX) model of HIV-1 (ref. 36) to investigate whether HIV-1-specific CD8+ T cells sourced from ID107 can suppress active viral replication in vivo and hence prevent viral rebound (Fig. 6a). We used engraftments of memory CD4+ and CD8+ (mCD4+ and mCD8+) T cells derived from ID107 ATI week 53. Following initial engraftment of 19 mice with mCD4+ T cells from ID107, two mice experienced a spontaneous viral rebound (Fig. 6b), and the remaining mice were inoculated with plasma from the spontaneous viral rebound. One week after two rounds of inoculation, all but one mouse were robustly viremic (>10^5 copies per ml). A subset of mice then received an engraftment of autologous mCD8+ T cells from ID107 (+CD8+ T cells, $n = 7$) (Fig. 6c). Plasma HIV-1 RNA, CD4 counts and CD8 counts were monitored in all mice for a further 5 weeks (Fig. 6b,c and Extended Data Fig. 7a,b).

The engraftment of autologous mCD8+ T cells led to a 1,191-fold reduction in the median plasma HIV-1 viral load within 3 weeks compared to mice receiving mCD4+ T cells only, concurrent with an increase in CD8+ T cell counts (Fig. 6b, c). Following this sharp decline, viral loads

remained durably suppressed relative to controls for the remainder of the study, with a >2 − log difference still evident at week 11 ($P < 0.0001$; Extended Data Fig. 7c). These findings contrast with those from a similar experimental system using T cells from a progressor with HIV-1, in which the addition of CD8+ T cells produced only marginal reductions in viral load (Extended Data Fig. 7d). Altogether, these data indicate that mCD8+ T cells taken directly ex vivo from ID107 are sufficient to achieve potent suppression of autologous rebounding virus in vivo.

### Rebound viruses in ID142 show evidence of multiple immune escape mutations

To explore the mechanisms behind the viral rebound observed in ID142, we sequenced NFL plasma-derived genomes (~8.7 kb) from ATI week 142 (57,600 copies per ml)[25] (Fig. 7a), and compared these rebound viruses to genetically intact proviral sequences obtained from pre-rebound time points. We found that all rebound-derived sequences clustered separately from the pre-rebound genetically intact proviruses, suggesting that the rebound was caused by a genetically distinct viral variant (Fig. 7b).

Superinfection was ruled out by aligning all intact NFL sequences for ID142 with subtype B laboratory-strain viruses HXB2 and NL4-3, and ID142 *env* sequences with subtype B *env* sequences from six controls[4,5] (Extended Data Fig. 8). Furthermore, genotypic bNAb resistance assessments showed no evidence that the genetically distinct viral rebound population harbored variants with resistance mutations toward 3BNC117 or 10-1074.

### Rebound viruses are not neutralized by aNAbs

We next investigated the aNAb response during rebound. Plasma-derived HIV-1 *env* sequences recovered from rebound viruses possessed distinct point mutations in the hypervariable regions (V1–V5) of gp120, and in gp41, compared to proviral and qVOA *env* sequences isolated from before rebound (Extended Data Fig. 2b). To verify that these point mutations conferred escape from aNAbs, we generated pseudoviruses expressing plasma HIV-1 Env from weeks 135, 140 and 142 (ID142.Rebound1, ID142.Rebound2 and ID142.Rebound3, respectively). Autologous IgG purified before rebound exhibited poor neutralizing activity against rebound pseudoviruses, with all but one $IC_{50}$ values exceeding 100 µg ml$^{-1}$ (Fig. 7c). Autologous IgG from week 135 exhibited an $IC_{50}$ value of 12.5 µg ml$^{-1}$ (IIP at 10 mg ml$^{-1}$ of 5) against pseudovirus ID142.Rebound1, which represents contemporaneous virus (Fig. 7c,d). This is the threshold for viral replication in vivo (Supplementary Fig. 3f,g). Autologous IgG from weeks 135 and 140 exhibited higher $IC_{50}$ values of >20 µg ml$^{-1}$ and IIP values <5 against the other post-rebound pseudoviruses, which is insufficient to control replication. This indicates that plasma rebound viruses had escaped aNAb-mediated pressure.

**Fig. 5 | Effective HIV-1-specific CD8+ T cells maintain long-term ART-free control in PICs. a**, Batch-corrected uniform manifold approximation and projection (UMAP) of single-cell transcriptome data showing AIM-sorted T cells from the 16 samples sourced from two PICs (ID107 and ID142) and two non-controllers (NCs; ID104 and ID112) (total $n = 40,942$). CM, central memory; MAIT cell, mucosal-associated invariant T cell; TM, memory T cell. **b**, Heatmap showing scaled average expression of the virus-specific activation module per cluster. Rows (all module genes) are hierarchically clustered based on expression, whereas columns (clusters) are ordered by median module score. **c**, UMAP projection displaying virus-specific CD8 activation module score values per cell. **d**, UMAP projections showing DSB-normalized expression of selected surface protein markers ($n = 14,243$ cells from the two PICs). **e**, Bar plots of transcriptional cell subset distribution at pre-ATI in the two PICs and two NCs. *$P = 0.002$, median Benjamini–Hochberg-adjusted across 100 bootstrapped replicates; empirical Bayes moderated $t$-test, two-sided. **f**, Top 25 upregulated genes in cluster 1 ($n = 7,278$ cells) compared to clusters 10 (top; $n = 853$ cells) and 3 (bottom; $n = 4,634$ cells). Rows represent genes hierarchically clustered by expression level. **g**, Top enriched Gene Ontology (GO) terms in cluster 1 based

on differentially expressed genes compared to clusters 10 (top) and 3 (bottom). Top five terms ranked by $-\log_{10}$(Benjamini–Hochberg-adjusted $P$ value) from a one-sided hypergeometric test using the expressed genes in the dataset as background; ties shown. Terms are ordered by fold enrichment (observed/expected frequency). 'TCR-based immune response*' abbreviates the full GO term 'adaptive immune response based on somatic recombination of immune receptors built from immunoglobulin superfamily domains'. **h**, Alluvial plot showing the frequencies of each participant's ten largest clones over time. Clones are colored by within-participant rank and are not shared between participants. Weeks are plotted as categorical factors on the $x$ axis, which is not proportionally scaled to time. Panels show data from all collected time points (ID107, ATI weeks −57, 0, 14, 29, 95, 181, 281; ID142, −2, 51, 78; ID104, −57, 0, 5; ID112, −57, 0, 3; one AIM experiment per time point) except **e**, which includes only the latest pre-ATI time point (ATI week 0 for ID107, ID104 and ID112; ATI week −2 for ID142). All data represent biological replicates analyzed as single cells for cluster-level comparisons, as clonotypes for TCR repertoires or as whole samples for condition-level comparison. Cell counts for each cluster and donor are listed in Supplementary File 1.

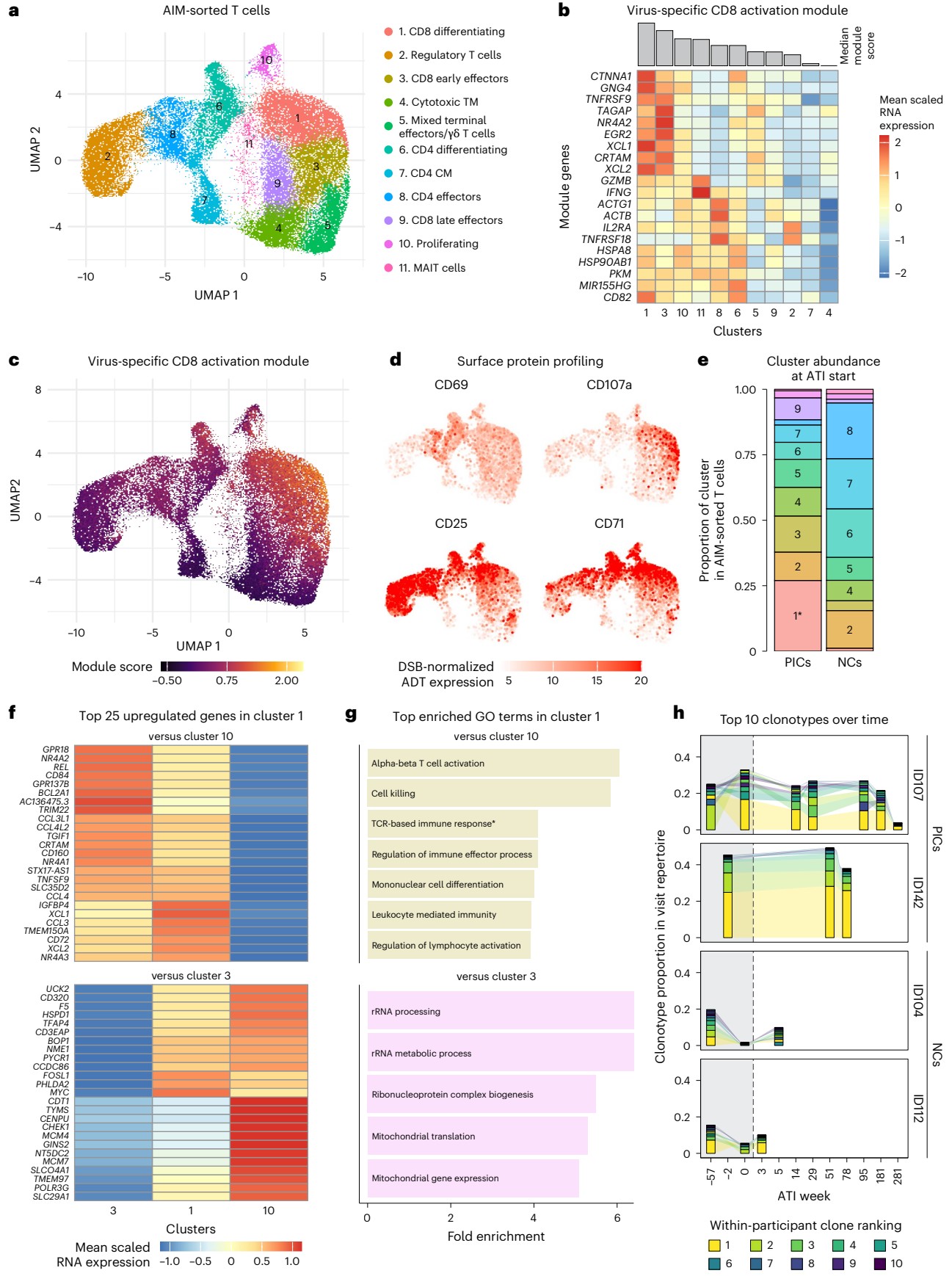

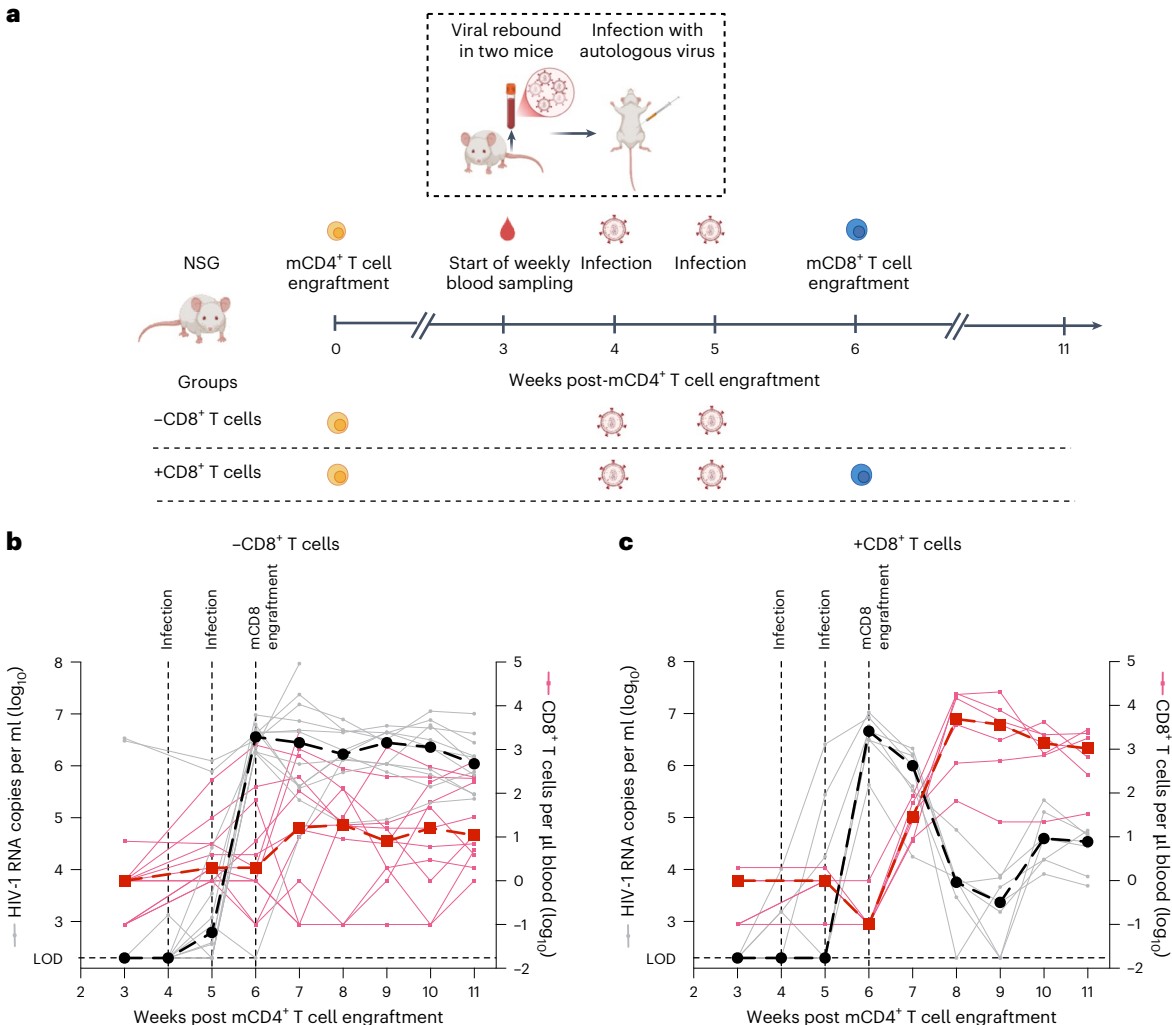

**Fig. 6 | HIV-1-specific CD8⁺ T cells suppress viral rebound in a HIV-1 participant-derived xenograft model. a**, Overview of PDX mouse model experiment assessing the in vivo suppressive capacity of ID107 mCD8⁺ T cells from ID107. Created in BioRender. Fisher, K. https://BioRender.com/y07rh63 (2026). **b,c**, Plasma HIV-1 viral load and CD8⁺ T cell count over time in NSG mice receiving mCD4⁺ T cells alone (n = 12) (**b**) or mCD4⁺ T cells and mCD8⁺ T cells (n = 7) (**c**). HIV-1 viral load is shown in gray (left y axis) and CD8⁺ T cell count is shown in red (right y axis). Dashed lines represent median viral load or CD8⁺ T cell count over time in the relevant color.

## Rebounding virus also carries mutations within HLA-matched CTL epitopes

We hypothesized that the sudden viral rebound observed in ID142 may also harbor evidence of escape from HIV-1-specific T cell responses. Comparing HIV-1 Gag, Pol and Nef amino acid sequences between pre- and post-rebound HIV-1 sequences, we identified multiple mutations within some HLA-restricted CD8⁺ T cell epitopes (Extended Data Fig. 9 and Extended Data Table 1). Specifically, we identified one HIV-1 Gag epitope that was mutated within 100% of post-rebound sequences, three HIV-1 Pol epitopes that were mutated within 82–100% of post-rebound sequences, and one mutation within a HIV-1 Nef epitope that was mutated within 95% of post-rebound sequences (Extended Data Fig. 9). Several of these mutated epitopes were found within regions previously shown to confer cytotoxic T lymphocyte (CTL) escape, or have emerged as an escape variant in humanized mouse models[37–39]. These results suggest that the viral variant that emerged during the rebound also carried mutations within HLA-matched CTL epitopes, suggestive of escape from the CD8⁺ T cell response. In support of this, we observed no change in the magnitude or quality of the HIV-1-specific CD8⁺ and CD4⁺ T cell response against heterologous HIV-1 peptide pools at viral rebound (Fig. 7e,f,h).

## Discussion

Our work investigating PIC in three PLWH has shown that suppressed viremia in the absence of ART is maintained by a potent neutralizing antibody response, as well as the presence of polyfunctional, effective HIV-1-specific T cells before ART interruption that responded rapidly to antigen exposure. All PICs harbored inducible infectious virus that persisted during ATI, and was biased toward integration within nongenic/centromeric regions of the genome, reflective of immune-mediated selection. This provides a model for how the immune system can suppress viral replication in the absence of ART and thus serves as a foundation for optimizing HIV-1 cure strategies.

We have provided strong evidence for the role of aNAbs in sustaining ART-free virologic control, consistent with studies identifying a role of aNAbs in the maintenance of PTC of HIV-1 in the absence of additional therapeutic intervention[13,40], and recently in the context of ATI and bNAb therapy[41]. aNAbs from two PICs were able to prevent exponential viral outgrowth of autologous virus in vitro. This contrasts with similar experiments conducted previously using sampling from individuals on suppressive ART, where, among different individuals, 0–96% of reservoir viruses could be blocked in the presence of autologous IgG[26]. We acknowledge that qVOA induces a small fraction of

genetically intact proviruses that persist in individuals on ART[42], and therefore there may be additional variants that we have not identified in this study that could be resistant to neutralization by autologous IgG. Furthermore, in both ID107 and ID142, aNAbs consistently demonstrated IIP values well above the 5 − log threshold typically observed with suppressive ART regimens, and potent neutralization of autologous replication-competent viral isolates was also observed for ID9254. When viral rebound did occur in ID142, we demonstrated that the rebounding virus was likely sourced from a variant that was resistant to aNAbs. Hence, the high level of inhibition by aNAbs may be an important contributor to the maintenance of control in PICs.

We also provide strong support for a role of effective HIV-1-specific T cell responses in the maintenance of long-term virological control. Profiling of AIM+ HIV-1-specific T cells revealed that, before ART interruption, ID107 and ID142 harbored higher frequencies of HIV-1-specific CD8+ T cells primed for activation and proliferation upon HIV-1 peptide stimulation compared to non-controllers, suggesting that the early presence of such cells supports a rapid response to the rising antigen load following ART interruption. These findings are consistent with previous studies linking combined differentiation and effector capacity of CD8+ T cells to control[43–45]; however, given that our single-cell analyses are limited to two PICs and two non-controllers, further validation of these specific findings in larger cohorts is required. Similarly, populations of polyfunctional HIV-1-specific memory CD8+ T cells, which have been associated with spontaneous control of HIV-1[46,47], at the pre-ATI timepoint were also enriched in PICs for specific HIV-1 protein targets compared to ART-suppressed individuals. We have therefore demonstrated, using two independent methodological approaches, the presence and enrichment of populations of HIV-1-specific CD8+ T cells before ART interruption in PICs that are poised to expand and respond to antigen production at the time of ATI and maintain long-term virological control, consistent with recent findings in an SIV model of PTC[48]. Finally, we have also identified a role for polyfunctional HIV-1-specific, and particularly Gag-specific, CD4+ T cells in the maintenance of PIC. Polyfunctional CD4+ T cell responses have previously been associated with spontaneous control of HIV-1 (ref. 49), the development of neutralizing antibody responses in viremic controllers[50], and have been shown to be preserved by early ART initiation[51], but to our knowledge have not yet been directly associated with post-treatment or post-intervention control.

The results from our PDX mouse model further emphasizes the role of HIV-1-specific CD8+ T cells in the maintenance of viral control in ID107. The magnitude and duration of suppression observed with ex vivo ID107 CD8+ T cells exceeded those previously reported for HIV-specific T cell therapy products in the PDX model, and in our experiment utilizing T cells from a progressor individual with HIV-1 (ref. 36); however, we do acknowledge that our PDX experiment with T cells from ID107 utilized infection with autologous virus with an unknown 50% tissue culture infectious dose (TCID50), rather than superinfection with TCID50 10,000 HIV-1 JRCSF, as in other experiments. Nonetheless, peak viremia levels were comparable across all settings, indicating that CD8+ T cells from ID107 can suppress robust replication of autologous virus. Future studies that use matched infection parameters will be needed to rigorously compare the role of CD8+ T cells in viral suppression in non-controllers.

In our PICs, early ART initiation may have provided a more favorable environment for the induction of immune responses capable of controlling HIV-1(ref. 2) by limiting the size and diversity of the proviral reservoir[52,53], and improving HIV-1-specific CD8+ and CD4+ T and B cell responses[54–56]. We additionally observed that for ID107, aNAb responses matured over time on ART. This has been observed in other studies showing that aNAb responses matured during ART in people who start treatment early and then contribute to control of viral rebound following ART interruption[40,57]. Furthermore, all PICs received multiple infusions of one or two bNAbs, respectively, before ART interruption. This may have enhanced the development of HIV-1-specific T and B cell responses through a 'vaccinal effect', whereby the binding of the bNAbs to the virus can enhance the maturation of the immune response[3,58]. We and others have demonstrated that the administration of bNAbs before ATI increases time to loss of virological control and, in some cases, enhances HIV-1-specific CD4+ and CD8+ T cell responses[4–8,10,59,60]. Furthermore, for ID107, these effects may have been enhanced by 3BNC117 administration at ART initiation due to the increased availability of HIV-1 antigen at the time of ART initiation compared to during suppressive ART[61,62]. We therefore speculate that the administration of bNAbs for ID107, ID9254 and ID142, in combination with early ART initiation, has contributed to their respective periods of long-term virological control.

In ID142, despite the potent HIV-1-specific aNAb and T cell response detected, viral rebound occurred at ATI week 134. Initial viral rebound likely originated from a phylogenetically distinct minor variant not previously detected in the reservoir by qVOA or proviral sequencing, or as a result of low-level replication occurring in an anatomic location other than the peripheral blood, eventually leading to virus escape. ID142 initiated ART 5 months post-acquisition, which is 3 months later than ID107 but of similar timing to ID9254. Viral variants containing

**Fig. 7 | Viral rebound in ID142 caused by viral variant showing significant immune escape mutations. a**, Plasma HIV-1 viral load over time following ART interruption for ID142. **b**, Maximum likelihood phylogenetic tree of pre-rebound and rebound HIV-1 sequences sourced from multiple time points following ATI. Asterisk indicates branch support >70%. Scale bar indicates nucleotide substitutions per site. **c**, Dose–response neutralization curves for pseudoviruses sourced from post-rebound *env* sequences ID142.REBOUND1, ID142.REBOUND2 and ID142.REBOUND3. Pseudoviruses were tested against autologous IgG purified from longitudinal time points (see color keys; one IgG sample per time point). Data represent mean ± s.d., based on three replicates of culture wells containing pseudovirus and TZM-bl cells for each IgG concentration. For each experiment, control wells were included containing pseudovirus and TZM-bl cells with no IgG (six wells per experiment) and TZM-bl cells with no pseudovirus or IgG (six wells per experiment). The *env* sequences used to generate pseudoviruses ID142.REBOUND1, ID142.REBOUND2 and ID142. REBOUND3 are shown in Extended Data Fig. 2b. **d**, IIP values at 10 mg ml[−1] of longitudinally sampled autologous IgG for each pseudovirus from pre-rebound and rebound time points, and these are represented on the right *y* axis. The dashed line at IIP = 5 represents the threshold for effective suppression of in vivo viral replication. **e**, Spectral flow cytometric assessment of cytokine production by memory CD8+ T cells (top) and memory CD4+ T cells (bottom) in response to stimulation with HIV-1 Gag, Pol, Env or Nef peptide pools in ID142 between pre-rebound and rebound time points. Time points ATI weeks 135 and 140 are included and highlighted in black boxes. Color denotes positivity for cytokines and degranulation (CD107a), and the eight dominant responses are highlighted in bold in the legend. Each bar represents a single experiment per individual and time point, utilizing a minimum of 0.8 × 10[6] PBMCs per stimulation. The frequency of cells expressing each combination of cytokines is calculated by subtracting the value of responding cells to the specific stimulation by that within the negative control (stimulation of cells with DMSO). **f**, HIV-1-specific CD8+ T cell responses over time pre- and post-rebound as characterized by lymphocyte proliferation assay (column graphs; left *y* axis) and IFNγ ELISpot assay (symbols; right *y* axis) in response to stimulation with HIV-1 Gag, Pol and Nef peptide pools. Each data point represents a single experiment per stimulation and time point. **g**, HIV-1-specific CD4+ T cell responses over time pre- and post-rebound as characterized by lymphocyte proliferation assay in response to stimulation with HIV-1 Gag, Pol and Nef peptide pools. Each data point represents a single experiment per stimulation and time point. **h,i**, HIV-1-specific CD8+ (**h**) and CD4+ (**i**) T cell responses as characterized by the AIM assay in response to stimulation with HIV-1 Gag, Pol and Nef peptide pools. Each data point represents a single experiment per stimulation and time point. Gray shading indicates time on ART and blue shading indicates viremic time points during viral rebound. Analyzed samples are color-coded according to time point.

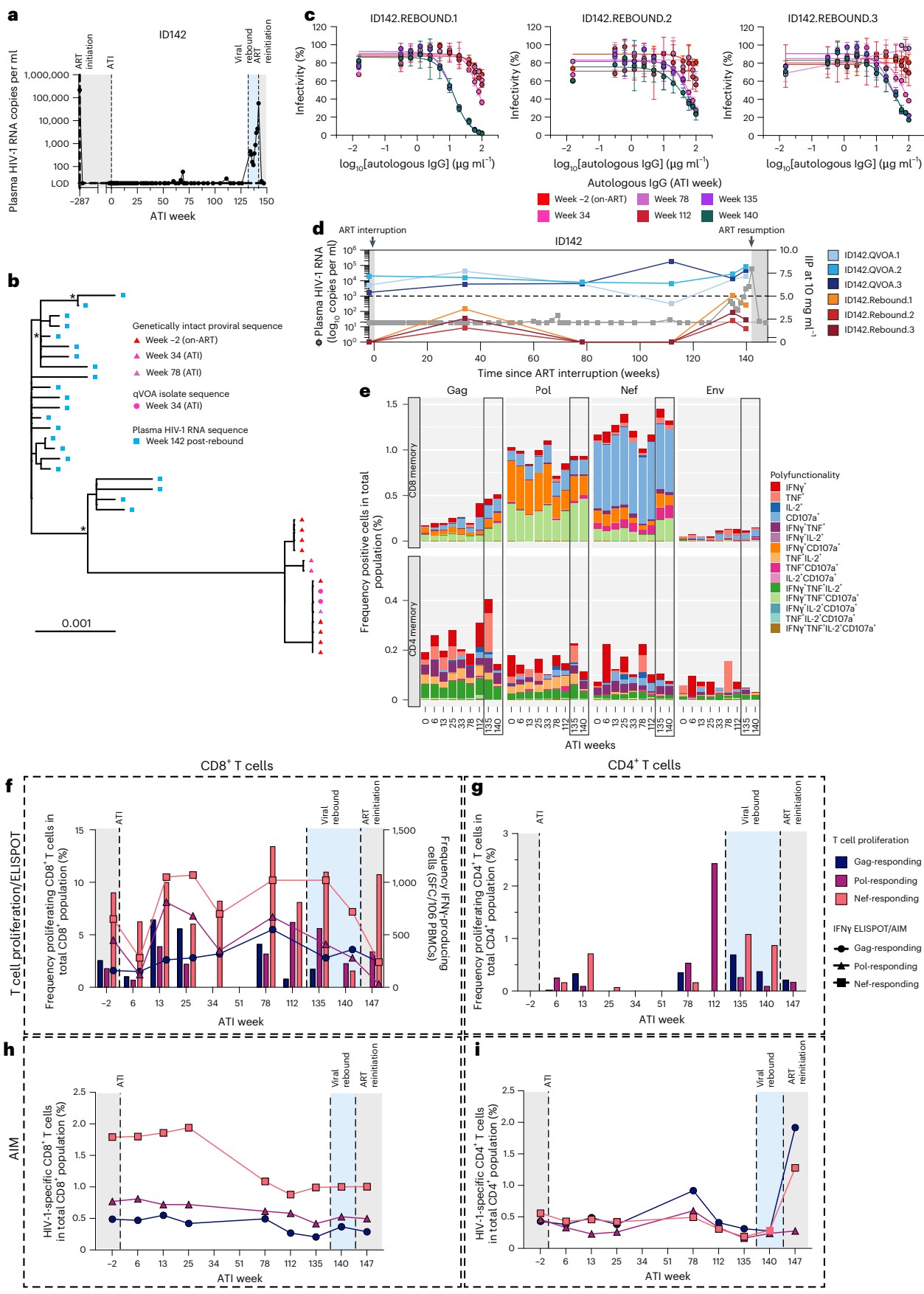

immune escape variants have been shown to emerge as early as 2–3 months post-acquisition[63]. During viral rebound, IIP values quickly decreased from 5 to 1.43, showing complete escape from aNAb pressure. In addition, comparison of pre-rebound and rebound HIV-1 sequences indicated mutations within HLA-matched CTL epitopes within the rebound population, which is suggestive of escape from the T cell response in this individual, though we acknowledge that we have not directly investigated whether these mutations caused a loss of recognition by HIV-1-specific CD8$^+$ T cells and therefore cannot conclude that these mutations caused escape from the HIV-1-specific CD8$^+$ T cell response. Though many HLA-matched epitopes were not mutated, different epitopes do mutate in response to CTL pressure at different rates[64,65]. We also acknowledge that other mechanisms, such as a reduced functional capacity of the HIV-1-specific CD8$^+$ T cells, may have contributed to the viral rebound, which we have not measured in our study. Thus, we conclude that viral evolution and seeding of minor variants in the reservoir with immune escape mutations may in some cases limit the long-term effectiveness of potent immune responses and lead to eventual viral rebound.

We acknowledge several limitations to our study. First, only three male individuals have been included in the study, and therefore the applicability of our results to others with HIV-1 will require further investigation. Furthermore, our sampling was limited to the peripheral blood. Though we speculate that activity in immune tissues such as the lymph nodes and gut-associated lymphoid tissue were heavily involved in the development of the potent immune responses observed, no tissue samples were collected, and this will therefore need to be addressed in future prospective studies.

In conclusion, our results have demonstrated that in three long-term PICs the development of potent immune responses, including aNAbs and HIV-1-specific CD4$^+$ and CD8$^+$ T cells, toward a susceptible and genetically restricted viral population contribute to the maintenance of long-term ART-free control of HIV-1. These observations will inform the optimization and development of HIV-1 curative strategies.

## Online content

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

Katie Fisher [1,2,22], Mauro A. Garcia[3,22], Giacomo S. Frattari [1,2,22], Chloé Naasz[4,22], Junlin Zhuo [3,22], Miriam Rosás-Umbert [1,2], Lisa L. Dietz [1,2], Anna Karina Juhl[1,2], Emma Falling Iversen [1,2], Rikke Olesen[1,2], Mariane H. Schleimann[1], Marie H. Pahus[1,2], Isik S. Johansen[5], Merle Henderson[6,7], Leah Carrere[4], Isabelle Roseto [4], Ce Gao [4,8], Xu G. Yu[4], Emily J. Fray [3], Beril Aydin [3], Donald Lubbeck [3], Jun Lai[3], Francesco R. Simonetti [3], Ali Danesh[9], Itzayana Miller [9], Pilar Mendoza[10], Julia Niessl [11,12], Christian Gaebler [13,14], Michael S. Seaman[15], Daniel E. Kaufmann[11,12,16], Clara Lehmann[17,18,19], Henning Gruell [18,20], Florian Klein [18,19,20], Marina Caskey [10], Michel C. Nussenzweig [10,21], Martin Tolstrup [1,2], R. Brad Jones [9], Jesper D. Gunst [1,2], Janet D. Siliciano [3], Mathias Lichterfeld [4,8], Robert F. Siliciano [3,21] & Ole S. Søgaard [1,2] ✉

[1]Department of Infectious Diseases, Aarhus University Hospital, Aarhus, Denmark. [2]Department of Clinical Medicine, Aarhus University, Aarhus, Denmark. [3]Department of Medicine, Johns Hopkins University School of Medicine, Baltimore, MD, USA. [4]Ragon Institute of MGH, MIT and Harvard, Cambridge, MA, USA. [5]Department of Infectious Diseases, Odense University Hospital, Odense, Denmark. [6]Department of Infectious Disease, Faculty of Medicine, Imperial College, London, UK. [7]Imperial College NIHR Biomedical Research Centre, London, UK. [8]Division of Infectious Diseases, Brigham and Women's Hospital, Harvard Medical School, Boston, MA, USA. [9]Division of Infectious Diseases, Weill Cornell Medicine, New York, NY, USA. [10]Laboratory of Molecular Immunology, The Rockefeller University, New York, NY, USA. [11]Research Centre of the Centre Hospitalier de l'Université de Montréal (CRCHUM), Montreal, Quebec, Canada. [12]Université de Montréal, Montreal, Quebec, Canada. [13]Laboratory of Translational Immunology of Viral Infections, Department of Infectious Diseases and Critical Care Medicine, Charité – Universitätsmedizin Berlin, Corporate Member of Freie Universität Berlin and Humboldt-Universität zu Berlin, Berlin, Germany. [14]Berlin Institute of Health, Berlin, Germany. [15]Center for Virology and Vaccine Research, Beth Israel Deaconess Medical Center, Harvard Medical School, Boston, MA, USA. [16]Division of Infectious Diseases, Department of Medicine, Lausanne University Hospital and University of Lausanne, Lausanne, Switzerland. [17]Department of Internal Medicine, Faculty of Medicine and University Hospital Cologne, University of Cologne, Cologne, Germany. [18]German Center for Infection Research (DZIF), Partner Site Bonn-Cologne, Cologne, Germany. [19]Center for Molecular Medicine Cologne (CMMC), University of Cologne, Cologne, Germany. [20]Institute of Virology, Faculty of Medicine and University Hospital Cologne, University of Cologne, Cologne, Germany. [21]Howard Hughes Medical Institute, Baltimore, MD, USA. [22]These authors contributed equally: Katie Fisher, Mauro A. Garcia, Giacomo S. Frattari, Chloé Naasz, Junlin Zhuo. ✉e-mail: olesoega@rm.dk

## Methods

### Participant samples

Longitudinal cryopreserved PBMCs and plasma and/or serum samples stored at −80 °C were utilized in this study. ID107, and six other individuals, including two non-controller individuals ID104 and ID112, initially participated in the eCLEAR trial (EudraCT:2015-002234-53, ClinicalTrials.gov NCT03041012)[4], and ID142 initially participated in the TITAN trial (EudraCT: 2018-001165-16, ClinicalTrials.gov NCT03837756)[5]. ID9254 initially participated in an open-label phase 1b study in people with HIV-1 on suppressive ART (EudraCT: 2016-002803-25, ClinicalTrials.gov NCT02825797)[10]. All individuals were found to harbor viruses that were sensitive to their respective bNAb treatment at enrollment and throughout follow-up, based on phenotypic and/or genotypic assessment methods[4,5,10] (Supplementary Table 6). Written informed consent was obtained from all individuals. All participants were offered financial compensation for lost income, time incurred for study procedures and/or for travel costs related to study visits. Continued ATI and leukapheresis for ID107 and ID142 was conducted under the trial protocol according to the National Committee on Health Research Ethics in Denmark (1-10-72-38-23). The decision to continue pausing ART was consensually made between the person or family, his physicians and the study team. Secondary use of samples from individuals who participated in eCLEAR was approved by the National Committee on Health Research Ethics in Denmark (1-10-72-110-16). For ID9254, the individual made the decision to remain off-ART after study completion except for a 3-day period of ART at week 34 after ART interruption, and is regularly monitored. Research blood samples were collected under an observational protocol approved by the Ethics Committee of the Medical Faculty of the University of Cologne (UKK 16-054).

### Modified IPDA

Genetically intact proviruses were quantified using a modified intact proviral DNA assay (IPDA) as described previously[4,5,18]. Primer–probe sets targeting the φ-region[18] and the rev-response element within *env*[17] were used to identify proviruses that are likely to be genetically intact in a duplex ddPCR assay using genomic DNA isolated from CD4+ T cells[18]. In parallel, two regions within the human *RPP30* gene were also quantified. All primer and probe sets can be found listed in Supplementary Table 7. The output of both the HIV-1 and RPP30 assays were then used to estimate the number of intact HIV-1 copies per $10^6$ CD4+ T cells, based on the quantity of input DNA. A detailed methodology can be found in the Supplementary Notes.

### Single-genome NFL proviral sequencing and matched integration site and proviral sequencing

Single-genome NFL proviral sequencing was performed using genomic DNA isolated from cryopreserved PBMCs as described previously[21]. In brief, isolated genomic DNA diluted to single-genome levels based on Poisson distribution statistics were subjected to single-genome amplification using Platinum Taq (Invitrogen, 10966018) and nested primers spanning NFL HIV-1 (HXB2 coordinates 638-9362). For a combined analysis of proviral sequences and corresponding chromosomal integration sites[21], full-genome amplification was preceded by a multiple displacement amplification step with phi29 polymerase (REPLI-g Single Cell kit, QIAGEN,150345) per the manufacturer's protocol. Proviral sequences were amplified from the unbiased WGA products using Platinum Taq (Invitrogen) and nested primers spanning NFL HIV-1 and were visualized by agarose gel electrophoresis (Quantify One and ChemiDOC MP Image Lab, Bio-Rad 12003154). Amplification products were subjected to Illumina MiSeq sequencing at the MGH DNA Core facility. Resulting short reads were de novo assembled and aligned to HXB2 to identify genomic defects using an automated in-house pipeline (https://github.com/BWH-Lichterfeld-Lab/Intactness-Pipeline). The presence or absence of hypermutations associated

with APOBEC3G or APOBEC3F was determined using the Los Alamos HIV sequence Database Hypermut 3.0 program (https://www.hiv.lanl.gov/content/sequence/HYPERMUT/hypermutv3.html). Viral sequences that lacked all the defects described above were termed 'intact.' Proviral species that were completely sequence-identical were considered clonal.

### Proviral integration site analysis

Integration sites associated with individual viral sequences were obtained by integration site loop amplification (ISLA) assays as previously described[67]. DNA produced by WGA was used as a template. Resulting PCR products of the ISLA reaction were subjected to next-generation sequencing using Illumina MiSeq. MiSeq paired-end FASTQ files were demultiplexed; small reads (142 bp) were then aligned simultaneously to the human reference genome GRCh38 and the HIV-1 reference genome HXB2 using bwa-mem (v.0.7.19)[68]. Biocomputational identification of integration sites was performed according to previously described procedures[67]. The final list of integration sites and corresponding chromosomal annotations was obtained using Ensembl (v.113; www.ensembl.org), the UCSC Genome Browser (www.genome.ucsc.edu) and GENCODE (v.47; www.gencodegenes.org). Chromosomal coordinates of integration sites were indicated using the Hg38 reference genome nomenclature.

### Quadruplex qPCR

Genetically intact proviral genomes were isolated and sequenced from PBMC samples sourced from ID9254 by Q4PCR as previously described[22]. In brief, CD4+ T cells were isolated from PBMCs by negative selection using the CD4+ T Cell Isolation kit (Miltenyi Biotec) and genomic DNA was isolated (Gentra Puregene Cell kit (QIAGEN)). Genomic DNA was then assayed by qPCR for the presence of HIV-1 *gag* to determine the limiting dilution of the DNA. Then, at the limiting dilution, NFL proviruses were amplified using primers targeting the 5′ and 3′ LTRs as described previously[22]. The products of these outer NFL PCRs were subjected to four-plex qPCR (targeting HIV-1 packaging signal, *env*, *gag* and *pol*) to identify amplicons that were most likely to contain a genetically intact provirus. NFL amplicons that were positive for at least two of four HIV-1 targets were used as template for the inner NFL PCR. These products were then subject to sequencing on the Illumina MiSeq platform. Amplicons were assembled using an in-house pipeline described previously (https://github.com/stratust/DIHIVA)[22].

### Quantitative viral outgrowth assay

qVOA was performed to quantify the frequency of inducible provirus for ID107 and ID142 as described previously[69–72], with minor modifications[59]. A detailed description of laboratory methods can be found in Supplementary Notes. CD4+ T cells were isolated from PBMCs by negative selection and were plated at 50,000 cells per well in round-bottom 96-well culture plates. Cells were stimulated with PHA and then cultured for 12 days, before culture wells positive for inducible, infectious virus were identified using TZM-bl assays and quantified using limiting dilution analysis as previously described[66].

For ID9254, the similar quantitative and qualitative viral outgrowth assay (Q²VOA)[24] was used to quantify the inducible proviral reservoir, details of which can be found in the Supplementary Notes.

### NFL sequencing of HIV-1 RNA from qVOA cultures

NFL (~8.7 kb; HXB2 coordinates 817-9501) HIV-1 RNA genomes were sequenced using a previously described assay based on single-genome sequencing: plasma-derived RNA using long-range sequencing (PRLS) assay[25]. Approximately ten positive qVOA cultures were randomly chosen from ID107 timepoints at ATI weeks 29, 227 and 281. For ID142, two positive qVOA cultures were used for sequencing from ATI week 34. Detailed descriptions of methodology can be found in the Supplementary Notes. Following assembly of contigs, a consensus sequence

was generated from at least three sequences per qVOA culture, and this was used to compare the virus present in individual qVOA wells to one another and to genetically intact proviruses generated by MIP-seq. For this comparison, all sequences were trimmed to the ~8.7-kb region sequenced by PRLS. Genetically identical sequences were identified using the Los Alamos webtool ElimDupes (https://www.hiv.lanl.gov/content/sequence/elimdupesv2/elimdupes.html). Phylogenetic trees were generated using PhyML[73], using the HKY85 nucleotide substitution model and a gamma rate of 4. Branch support was inferred using 1,000 bootstraps. Phylogenetic trees were visualized using ggTree[74] as described previously[25].

## Single-genome sequencing of residual viremia
Characterization of residual viremia HIV-1 sequences in plasma was performed as described previously[75], with some modifications. In brief, plasma was first spun at 3,500$g$ for 15 min at 4 °C to remove cell debris, lipids and fibrinogen. The supernatant was then transferred to new tubes and spun at 21,000$g$ for 2 h at 4 °C. Viral pellets then underwent RNA extraction[76]. HIV-1 RNA was used immediately for reverse transcription with SuperScript III in HIV-1 *env* using the primer *env*_RO 5′-GCARATGAGTTTTCYAGAGCA-3′ as previously described[77]. cDNA was then diluted to the end point and used for single-genome sequencing of a partial region of the HIV-1 *env* gene (HXB2 position 6980-8036) as previously described[77]. Primer sequences are listed in Supplementary Table 8. PCR products were sequenced by Sanger sequencing using the nested PCR primers (Azenta Life Sciences).

## Single-genome sequencing of proviral *env*
Single-genome sequencing of proviral *env* was performed as described previously[78]. In brief, following extraction of genomic DNA from resting CD4$^+$ T cells, proviral *env* DNA was amplified using a two-step nested PCR protocol targeting HXB2 5983-8882 as described previously[78]. Detailed methodology, including primer sequences and cycling conditions are listed in the Supplementary Notes.

## Modified qVOA and single-genome *env* sequencing of positive cultures and plasma-derived genomes
Modified qVOAs (mQVOAs) were conducted as previously described[26]. In brief, purified resting CD4$^+$ T cells were seeded into qVOA culture wells at 200,000 cells per well with up to 107 replicate wells and 1 negative control well per qVOA arm. Resting CD4$^+$ T cells were cultured with either no IgG, HIV-donor IgG (50 µg ml$^{-1}$) or autologous IgG (50 µg ml$^{-1}$). Cultures were carried out for 14 days as previously described, and assayed for HIV-1 p24.

Plasma viral RNA was extracted from p24$^+$ qVOA supernatants using the Viral RNA Isolation kit (Zymo Research, R1041) as previously described[78]. cDNA was synthesized using the *env*-specific primer OFM19 (5′-GCACTCAAGGCAAGCTTTATTGAGGCTTA-3′) with SuperScript III Reverse Transcriptase and RNaseOUT (Invitrogen) according to the manufacturer's instructions, using the program 55 °C for 50 min followed by 85 °C for 10 min. Single-genome amplification (SGA) of HIV-1 *env* was then performed at limiting dilution as previously described[4]. A detailed methodology is provided in the Supplementary Notes.

For plasma HIV-1 RNA *env* sequencing, viral RNA extraction was performed as described previously for residual viremia *env* sequencing[76]. cDNA was synthesized using Induro Reverse Transcriptase (New England Biolabs, M0681L). RNA and primer (OFM19) were first denatured at 65 °C for 5 min then snap-cooled at −20 °C. Reverse transcription was conducted in 20-µl reactions containing 1× Induro RT buffer, 1 mM dNTPs, 5 mM dithiothreitol (DTT) and 200 U Induro RT, and incubated at 55 °C for 50 min, followed by 95 °C for 10 min. SGA of *env* was performed using the same PCR conditions and primer sets described for mQVOA-positive wells, with a detailed methodology provided in Supplementary Notes.

## Isolation of autologous IgG antibodies
For ID107 and ID142, autologous IgG antibodies were isolated from heat-inactivated plasma using a NAb Protein A Plus Spin Column (Thermo Scientific, 89956). IgG samples were dialyzed in 1× phosphate-buffered saline (PBS) solution, pH 7.2, at 4 °C to remove residual ART drugs. Final IgG concentrations were measured using Nanodrop 2000 Spectrophotometer (Thermo Scientific).

For ID9254, IgG were purified from heat-inactivated serum (56 °C for 40 min) using Protein G Sepharose 4 Fast Flow (GE Healthcare, 1706180x) followed by buffer exchange to PBS using Amicon Ultra Centrifual Filters (Merck Millipore) and sterile filtration using Ultrafree-MC columns (Merck Millipore).

## Production of HIV-1 Env-expressing pseudoviruses
*Env* sequences of selected viral isolates were cloned into pcDNA 3.4 TOPO vector (Thermo Fisher Scientific, A14697) using either TOPO-TA Cloning (Thermo Fisher Scientific, 450071) or In Fusion Snap Assembly (Clontech, 638948) in accordance with the manufacturer's instruction. When available, original PCR amplicons were directly cloned. Otherwise, for several isolates, we designed synthetic double-stranded DNA (gBlock) templates based on sequences derived from plasma, then amplified and cloned these using In Fusion Snap Assembly. The resulting Env-expression vector driven by a cytomegalovirus (CMV) promoter (12.5 µg) was transfected into HEK293T cells together with 15 µg pNL4-3-ΔEnv−eGFP packaging construct[27] in the presence of 2.5 µg pAdvantage construct (Promega, E1711) to boost protein expression. HEK293T cells were incubated at 37 °C, 5% CO$_2$ for 72 h, and culture supernatant containing the isolate-specific Env-expressing pseudoviruses was collected, centrifuged to remove cell debris, filtered through a 0.45-µm Steriflip (Millipore Sigma, SE1M003M00) and flash frozen.

## In vitro pseudoviruses neutralization assay
Linear ranges for isolate-specific pseudoviruses were determined by titrating the pseudoviruses on TZM-bl cells (10,000 TZM-bl cells per culture well) with 3−6 technical replicates for each condition. In vitro neutralization assays were conducted with a pseudovirus concentration within the linear dynamic range. Dilutions of autologous IgG antibodies (10 ng ml$^{-1}$ to 100 µg ml$^{-1}$) were pre-incubated with pseudoviruses at 37 °C with 5% CO$_2$ for 90 min. TZM-bl cells in 50 µg ml$^{-1}$ DEAE-dextran were added and incubated at 37 °C with 5% CO$_2$ for 48 h. For each experiment, control wells were included containing pseudovirus and TZM-bl cells with no IgG (six wells per experiment) and TZM-bl cells with no pseudovirus or IgG (six wells per experiment). Viral infection was measured by luciferase production with Bright-Glo Reagent (Promega, E2610) 48 h later. For each pseudovirus, the percentage of infection ($f_u$) was calculated as a fraction of maximum infection (no antibodies present) after background signal was normalized. Both IC$_{50}$ and IIP values were calculated as previously described[27] to evaluate antibody potency and the in vivo efficacy of aNAbs. A detailed description of IIP calculations is provided in the Supplementary Notes.

## In vitro neutralization assay using replication-competent viral isolates
For ID9254, neutralization sensitivity of replication-competent proviruses was assessed using a similar TZM-bl-based neutralization assay as described above for ID107 and ID142, but using replication-competent viral isolates generated using viral outgrowth assays. Replication-competent viral isolates were cultured from positive Q2VOA wells from the time point ATI week 102, as described previously[10]. Then, sensitivity of these viral isolates to purified IgG sourced from the time point ATI week 102, and the bNAbs 3BNC117 and 10-1074, was assessed by TZM-bl neutralization using serial threefold dilutions of purified IgG, as previously described[10,79].

## Immunoassays

Lymphocyte proliferation, IFNγ ELISpot and AIM assays were performed on cryopreserved PBMCs that were thawed and washed with RPMI glutamine supplemented with penicillin–streptomycin and 10% FBS (cRPMI). PBMCs used for ELISpot and AIM were rested for 3 h at 37 °C, while PBMCs used for the lymphocyte proliferation assay were processed immediately. Detailed descriptions of the methodology of each immunoassay are provided in the Supplementary Notes.

## Spectral flow cytometry intracellular cytokine staining

**Cell stimulation and staining.** A 27-color spectral flow cytometry-based intracellular cytokine staining assay was used to characterize HIV-1-specific memory CD4$^+$ and CD8$^+$ T cell responses. PBMCs were thawed and rested overnight at 37 °C. The next day, cells were stimulated with HIV-1 peptide mix (Pol, Gag, Nef or Env) or negative control (DMSO) together with co-stimulatory antibodies, secretion inhibitors and anti-CD107a antibody, and were incubated for 6 h at 37 °C. Next, cells were stained with viability dye, blocked and stained for markers of lineage/differentiation (CD3, CD4, CD8, CD16, CD19, CD45RA, CCR7, CD27 and CD95), chemokine receptors (CCR4 and CXCR5), cytokines (IFNγ, TNF and IL-2), proliferation (Ki67), transcription (FOXP3, T-bet and TCF1), effector proteins (perforin, GzmB, GzmK and granulysin) and immune checkpoints (TIGIT and PD-1). Details of all antibodies used are listed in Supplementary Table 9. For intracellular markers, cells were fixed and permeabilized using the eBioscience Foxp3/Transcription Factor Fixation/Permeabilization Concentrate and Diluent (Invitrogen, 00-5521-00) according to the manufacturer's instructions and stained overnight at 4 °C. The next day, cells were washed and acquired within 3 h on a five-laser Sony ID7000 Spectral Analyser (SONY Biotechnologies).

The following controls were used for unmixing: unstained PBMCs for autofluorescence spectrum characterization, single-stained PBMCs with LIVE/DEAD Fixable Blue Dead Cell Stain (Invitrogen, L23105) and single-stained beads (UltraComp eBeads Plus Compensation Beads; Invitrogen, 01-3333-42) for all fluorochrome conjugated antibodies. When switching to a new antibody lot, we checked whether the spectral curved matched our existing single-stain control and if deviations were found, we acquired a new single-stain control for the unmixing.

A detailed description of all post-acquisition analysis steps can be found in Supplementary Notes.

## Single-cell transcriptome, surface protein and TCR sequencing of AIM-sorted T cells

**Sample size.** Two PICs and two non-controllers with sufficient longitudinal material were included. Following quality filtering, 40,942 cells from 16 samples were retained (Supplementary Fig. 5), with a median of 2,094 cells per sample (interquartile range 1,573–3,572).

**Sorting.** The AIM assay was performed as described above, except that cells were stimulated with a combined HIV-1 Gag, Pol and Nef peptide pool at a final total peptide concentration of 0.7 µg ml$^{-1}$. After stimulation and staining, T cells were enriched using the Pan T cell Isolation kit (Miltenyi Biotec, 130-096-535) and sorted on a MACSQuant Tyto. Cells coexpressing 4-1BB and PD-L1, or 4-1BB and CD69 were collected.

**Library preparation, sequencing and read alignment.** After sorting, samples were stained with Total-Seq C Human Universal Cocktail v.1.0 (BioLegend, 399905; Supplementary File 1). Single-cell libraries were prepared using Chromium Next GEM Single Cell 5′ v.2 reagents (10x Genomics, 1000265, 1000252 and 1000541) according to the manufacturer's instructions (CG000330 Rev F). Library quality was assessed by TapeStation D5000 Screen Tape (Agilent, 5067-5592, G2991AA) and Qubit dsDNA High Sensitivity Assay (Thermo Fisher Scientific, Q32851 and Q33238). Libraries failing quality control were excluded (Supplementary File 1). Sequencing was performed on an Illumina NovaSeq

6000 at the Department of Molecular Medicine, Aarhus University Hospital. Reads were aligned with Cell Ranger v.8.0.1 (10x Genomics) to a custom GRCh38-2020-A reference supplemented with autologous or subtype-matched HIV-1 sequences (Los Alamos National Laboratory (LANL) HIV Database, https://www.hiv.lanl.gov/, accession numbers K03455 and MH705134). TCR and ADT reads were processed using the human V(D)J reference (GRCh38) and the Total-Seq-C Human Universal Cocktail v.1.0 barcode list (Supplementary File 1).

**Transcriptomic analysis.** Cells with ≤500 genes, ≤1,000 unique molecular identifiers (UMIs) or ≥7.5% mitochondrial content were excluded. Doublets were identified using scDblFinder (v.1.18.0)[80]. Cells with >3 productive TCR chains were additionally flagged as doublets and removed. Ambient RNA contamination was assessed using SoupX (v.1.6.2)[81], no correction was applied (levels <5%; Supplementary File 1). Filtered singlet datasets were integrated and clustered to isolate high-quality T cells (Supplementary File 1).

RNA counts were normalized and scaled in Seurat v.5 (Log-Normalize, ScaleData), with cell-cycle effects regressed out. Principal-component analysis (PCA) was performed on highly variable genes (2,000 during preprocessing, 1,500 in the final dataset), excluding nonprotein-coding and mitochondrial genes. PCA embeddings were batch-corrected using Harmony (v.1.2.3)[82].

Clustering was performed using FindNeighbors (20 neighbors and 15 principal components) and FindClusters (Leiden)[83] with clustree[84]-optimized resolutions of 0.6 (doublet discrimination), 0.8 (contaminants removal) and 0.7 (final). UMAP embeddings were generated using RunUMAP. Clusters were annotated manually using differentially expressed genes (DEGs). Differences in cluster proportions were assessed using Scanpro[85] with 100 bootstrapped runs, reporting the median adjusted $P$ value (Supplementary File 1).

Virus-specific activation scores were calculated using AddModuleScore with a published gene set[32,33]: *ACTB, ACTG1, CD82, CRTAM, CTNNA1, EGR2, GNG4, GZMB, HSP90AB1, HSPA8, IFNG, IL2RA, MIR155HG, NR4A2, PKM, TAGAP, TNFRSF18, TNFRSF9, XCL1* and *XCL2*.

DEGs for cluster annotation were identified using FindAllMarkers (log$_2$fold change (FC) ≥ 0.25 and expression ≥ 25%) with the Wilcoxon rank-sum test during preprocessing. For the final analysis, MAST (v.1.30.0)[86] was used with donor origin as a covariate. For analyses of clusters 1, 3 and 10, genes with log$_2$FC ≥ 0.5, expression ≥ 10% and adjusted $P < 0.01$ were retained and used for gene overrepresentation analysis with enrichGO (clusterProfiler v.4.12.6)[87] on GO Biological Process terms (C5: GO:BP, human; minimum gene set size of 15). Terms with adjusted $P < 0.01$ and $q < 0.05$ were retained. Redundancy was reduced using pairwise_termsim (enrichplot v.1.24.4)[88] and simplify (clusterProfiler). All DEGs and enriched GO terms are reported in Supplementary File 1.

**Surface protein analysis.** DSB-normalized (v.2.0.0)[89] counts were used for marker selection, protein-positivity thresholding and visualization. Centered log-ratio (CLR)-normalized counts were used for differential testing with FindAllMarkers using the Wilcoxon rank-sum and receiver operating characteristic tests. Following previous studies[90,91], only markers with $Z > 2$ versus isotype controls were tested. Markers with adjusted $P < 0.01$ and area under the curve > 0.7 were retained for visualization (Supplementary File 1). Hierarchical clustering was performed using hclust on Euclidean distances calculated with dist (stats v.4.4.0).

**TCR analysis.** Clonotypes were defined by shared VJ usage and identical CDR3 nucleotide sequences in productive αβ pairs, without merging three- and two-chain clonotypes. At least one chain was recovered in 33,059 cells (80%), and a productive αβ pair in 26,345 cells (64%). Clone frequency ranks were calculated per participant. Clonality was quantified using the Gini coefficient[35] (DescTools v.0.99.56), based on bootstrapped, downsampled repertoires ($n = 100$). Clusters with fewer

than 50 TCRs were excluded. Clonal overlap was quantified using the Morisita–Horn index[35] in R (v.4.4.0).

## HIV-participant-derived xenograft mouse model

All animal procedures were conducted according to a protocol approved by the Weill Cornell Medical College Institutional Animal Care and Use Committee (protocol 2018-0027). Detailed standardized operating procedures for this model have recently been published[92]. NOD.Cg-Prkdc$_{scid}$ Il2rg$_{tm1Wjl}$/SzJ mice (stock 005557), commonly referred to as NSG mice, were purchased from The Jackson Laboratory. Female (6–8-week-old) NSG mice were used in all studies. Mice were co-housed in ventilated cages with wood chip bedding and maintained in a temperature-controlled environment with a 12-h light–dark cycle at the facilities of Weill Cornell Medical College. The temperature of the holding rooms was maintained between 70–74 °F. The relative humidity was maintained between 30–70%.

In brief, mCD4$^+$ T cells were isolated from PBMCs by negative selection and NSG female mice were engrafted with $5 \times 10^6$ mCD4$^+$ T cells. Three weeks post-mCD4$^+$ T cell engraftment, the mice were bled weekly to quantify human cell counts and viral load. Spontaneous viral rebound occurred in two mice and therefore plasma from these two viremic mice was used to infect the remaining mice. A week following two rounds of infection, mice were divided across two groups for equivalent CD4$^+$ T cell counts and viral loads (Supplementary Table 10). One group of mice received an engraftment of $5 \times 10^6$ autologous mCD8$^+$ T cells. Viral load was quantified weekly. A detailed methodology is provided in the Supplementary Notes. The statistical comparison between the plasma HIV-1 RNA load of the individual mice at week 11 post-mCD4$^+$ T cell engraftment in the −CD8$^+$ T cells group ($n = 11$) and the +CD8$^+$ T cells group ($n = 7$) was performed using an unpaired $t$-test (two-tailed) on log-transformed data in GraphPad Prism. Normal distribution of log-transformed data was confirmed using a Shapiro–Wilk test.

We also included data from a PDX-HIV model using PBMCs from a person living with HIV-1 who did not exhibit signs of virological control in the absence of ART (clinical characteristics shown in Supplementary Table 11). This participant was recruited from Whitman Walker Health, Washington DC, through a protocol approved by the George Washington University Institution Review Board. Secondary use of PBMCs was approved by the Weill Cornell Medicine Institutional Review Board. The individual gave written informed consent.

Memory CD4$^+$ T cells were isolated by negative selection from PBMCs and 5.2 million live cells were engrafted to NSG mice via tail vein injection. All mice ($n = 11$) were infected with 10,000 TCID$_{50}$ HIV-JRCSF and one group of mice ($n = 5$) received 5.0 million live memory CD8$^+$ T cells on day 41 post engraftment. Blood was collected by tail nick on a weekly basis.

## Identification of unique viral variant following viral rebound in ID142

Plasma-derived HIV-1 RNA genomes were sequenced from ID142 ATI week 142 plasma using the PRLS assay as described above for positive qVOA cultures[25]. Genetically intact plasma-derived genomes were identified using the GeneCutter tool on the Los Alamos HIV database (https://www.hiv.lanl.gov/content/sequence/GENE_CUTTER/cutter.html). Phylogenetic trees including all genetically intact genomes sourced from post-rebound plasma and sequences derived from pre-rebound time points using MIP-seq and PRLS were generated as described above for qVOA-derived NFL sequences.

To check for the possibility of superinfection with a genetically unrelated viral strain causing the viral rebound in ID142, two analyses were performed, details of which are in the Supplementary Notes.

Phylogenetic trees of *env* sequences isolated from pre-rebound and post-rebound time points by MIP-seq, PRLS or SGA were generated using this nucleotide alignment using PhyML[73], using the GTR substitution model and a gamma rate of 4. Branch support was inferred with

1,000 bootstraps. An amino acid alignment was also generated, and the Highlighter tool[93] was used to visualize genetic changes in specific *env* regions, with the lowest diversity sequence in the alignment used as the reference.

HLA-matched wild-type and experimentally verified escape mutations within HIV-1 Gag, Pol and Nef were initially identified as previously described, utilizing the Los Alamos 'Best-defined CTL/CD8$^+$ Epitope Summary' (https://www.hiv.lanl.gov/content/immunology/tables/optimal_ctl_summary.html), Los Alamos 'CTL/CD8$^+$ Epitope Variants and Escape Mutations' table (http://www.hiv.lanl.gov/content/immunology/variants/ctl_variant.html) and the Biopython SeqUtils package (https://github.com/biopython/biopython/tree/master/Bio/SeqUtils)[94]. Additional amino acid changes between pre- and post-rebound Gag, Pol and Nef sequences were cross-referenced with the Los Alamos HIV-1 database tool HIV Molecular Immunology Database Search (https://www.hiv.lanl.gov/mojo/immunology/search/ctl/form.html). Mutations were visualized using Highlighter plots[93], using a consensus sequence of all pre-rebound variants as the reference.

## bNAb sensitivity screening

All sequences reported in this manuscript for the PICs were subjected to a genotypic assessment for bNAb sensitivity using the bioinformatic tool bNAb-ReP (v.1.1-4)[95], with a threshold for bNAb sensitivity of >0.8 for the prediction score of each single sequence and threshold of >90% of sequences from each individual having a prediction score of >0.8.

## Reporting summary

Further information on research design is available in the Nature Portfolio Reporting Summary linked to this article.

## Data availability

HIV-1 proviral, qVOA and plasma-derived sequences have been deposited in GenBank with the accession numbers PX465444-PX466083, MK115946-MK116091, MN090734-MN090850 and MW063053-MW063065. HIV-1 *env* sequences utilized to check for superinfection can be found in GenBank with the accession numbers OR014534-OR014555, OR014635-OR014658, OR014662-OR014681, OR014984-OR015005, OR015084-OR015110, OR015113-OR015155 and PX892135-PX892225. Paired single-cell transcriptome and TCR sequencing data have been uploaded to the European Genome-Phenome Archive (accession number EGAS50000001570). Access to data will be assessed on a case-by-case basis and made available upon signing a Data Access Agreement. Any additional information required can be accessed by contacting the corresponding author, O.S.S. (olesoega@rm.dk). Source data are provided with this paper.

## Code availability

All original code is available on GitHub at https://github.com/SoegaardLab/2026_Fisher_Garcia_Frattari_Naasz_et_al. The complete computational environment used for all single-cell transcriptome and TCR sequencing analyses is available as a Docker image on Docker Hub (gsf70/fisher_garcia_frattari_naasz_26:published). Any additional information can be accessed by contacting the corresponding author, O.S.S. (olesoega@rm.dk).

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

## Acknowledgements

We acknowledge with gratitude the generosity of the PLWH who participated in this study. Spectral flow cytometry was performed using the five-laser SONY ID7000 at the FACS Core Facility, Aarhus University, Denmark. The five-laser ID7000 is a generous gift from the Novo Nordisk Foundation, grant number NNF210C0066798. Single-cell transcriptome, surface protein and TCR sequencing was performed at the Department of Molecular Medicine (MOMA), Aarhus University Hospital. Illumina sequencing for MIP-seq was performed at MGH DNA Core facility. Figures 2 and 6 were created in BioRender. Sanger sequencing of HIV-1 *env* amplicons was performed by Azenta Life Sciences. HIV-1 *env* sequencing of linear amplicons was performed by Plasmidsaurus using Oxford Nanopore Technology with custom analysis and annotation. M.A.G. acknowledges the support of C.E.C.A. Hop and C. Khojasteh (DMPK Department, Genentech) and the Institute for Basic Biomedical Sciences at Johns Hopkins University. We also thank A. Edwards for her administrative assistance. Work in the O.S.S. laboratory was funded by the Danish Council for Independent Research (grants nos. 7016-00022 and 3101-00360B OSS), Central Region Denmark Research Fund, the Lundbeck Foundation (J.D.G. R381–2021–1405 and O.S.S. R313-2019-790), and Aarhus University. Research reported in this publication was also supported by the National Institute of Allergy and Infectious Diseases of the National Institutes of Health (NIH) (award number UM1AI164565 to R.B.J. and O.S.S., award number 1R01AI184285 to O.S.S. and R.B.J. and award number 1P01AI178376; O.S.S.). Work in the laboratory of M.L. was supported by NIH grants AI155233, AI152979, AI176579, AI184094, AI155171, MH134823 and DA047034 (all to M.L.). The laboratory of R.F.S. and J.D.S. is supported by NIH Martin Delaney grants UM1 AI126603, UM1 AI126620 and UM1 AI12661 (all to R.F.S. and J.D.S.), NIH grant U54AI170752 (to R.F.S. and J.D.S.) and the Howard Hughes Medical Institute. F.R.S. is supported by the Office of the NIH Director and National Institute of Dental & Craniofacial Research (DP5OD031834), the Martin Delaney Collaboratory PAVE (UM1AI164566), the Johns Hopkins University CFAR (P30AI094189) and the W.W. Smith Charitable Trust. Work in the laboratory of M.S.S. was supported by Gates Foundation award no. INV-036842. Work in the laboratory of D.E.K. was supported by the Canadian Institutes of Health Research grant 152977 and NIH grant UM1 AI-144462 (CHAVD). Work in the laboratory of M.C.N. and M.C. was supported by NIH grants (UM1 AI100663 and R01 AI129795 to M.C.N.), REACH Delaney (UM1 AI164565 to M.C.), the Einstein Rockefeller-CUNY Center for AIDS Research (1P30 AI124414-01A1), BEAT-HIV Delaney (UM1 AI126620 to M.C.), the Bill & Melinda Gates Foundation (INV-008540 and INV-002705) and the Stavros Niarchos Foundation through its grant to the SNF Institute for Global Infectious Disease Research at The Rockefeller University. M.C.N. is an HHMI investigator. C.G. was supported by the HJH-Foundation, the Hector-Foundation and by the NIH (REACH Delaney grant UM1 AI164565 subaward). M.R.U. was

supported by the Gilead Sciences Research Scholars Program in HIV International. C. Gaebler is a Charité-Foundation Recruiting Grantee and received support by the European Research Council under the European Union's Horizon 2020 research and innovation programme (grant agreement no. 101162138, Project 'HIV CURE MISSION'). None of the specific sources of funding had any role in the conceptualization, design, data collection, analysis, decision to publish or preparation of the manuscript.

## Author contributions

K.F., M.A.G., G.S.F., C.N., J.Z., M.R.U., L.L.D., A.K.J., E.F.I., R.O., M.H.S., M.H.P., M.H. L.C., I.R., C. Gao, E.J.F., B.A., D.L., A.D., I.M., P.M., J.N., C. Gaebler and M.S.S. performed experiments and analyzed data. M.A.G., G.S.F. and L.L.D. contributed to manuscript writing. J.D.G., I.S.J., C.L. and H.G. recruited participants. K.F. wrote the manuscript. X.G.Y., J.L., F.R.S., M.S.S., D.E.K., F.K., M.C., M.C.N., M.T., R.B.J., J.D.S., M.D.L. and R.F.S. provided resources and supervision. C. Gaebler, D.E.K., F.K., M.C., M.C.N., R.B.J., J.D.S., M.L., R.F.S. and O.S.S. obtained funding. O.S.S. conceptualized and supervised the work.

## Competing interests

O.S.S. declares serving on scientific advisory boards for Abbvie, ViiV Healthcare, Gilead Sciences, Immunocore and Merck. Aspects of the IPDA are the subject of patent application PCT/US16/28822 filed by J.H.U. with R.F.S. as an inventor and licensed to AccelevirDx. F.R.S. received payments from Gilead for participating at scientific meetings. R.B.J. has served as an advisor to ViiV Healthcare and received payment for this role. M.L. has acted as a paid consultant for ViiV, Merck and MPM BioImpact. H.G. and F.K. are inventors on patent applications on HIV-neutralizing antibodies filed by the University of Cologne and have received compensation from the University of Cologne for licensed patents. P.M. is currently employed by Pfizer Vaccines, which researches and develops antiviral therapeutics and vaccines for profit. J.N. is currently employed by BioNTech SE. C.L. received payments from Gilead for participating at scientific meetings and declares serving on scientific advisory boards for Pfizer, ViiV Healthcare, Gilead Sciences and Merck. M.C. served on scientific advisory boards for Gilead, Merck and ViiV. M.C.N. is an inventor on antibody patents licensed by the Rockefeller University to Gilead. C. Gaebler received travel support from Gilead Sciences for participating at scientific meetings. The remaining authors declare no competing interests.

## Additional information

**Extended data** is available for this paper at https://doi.org/10.1038/s41590-026-02448-z.

**Correspondence and requests for materials** should be addressed to Ole S. Søgaard.

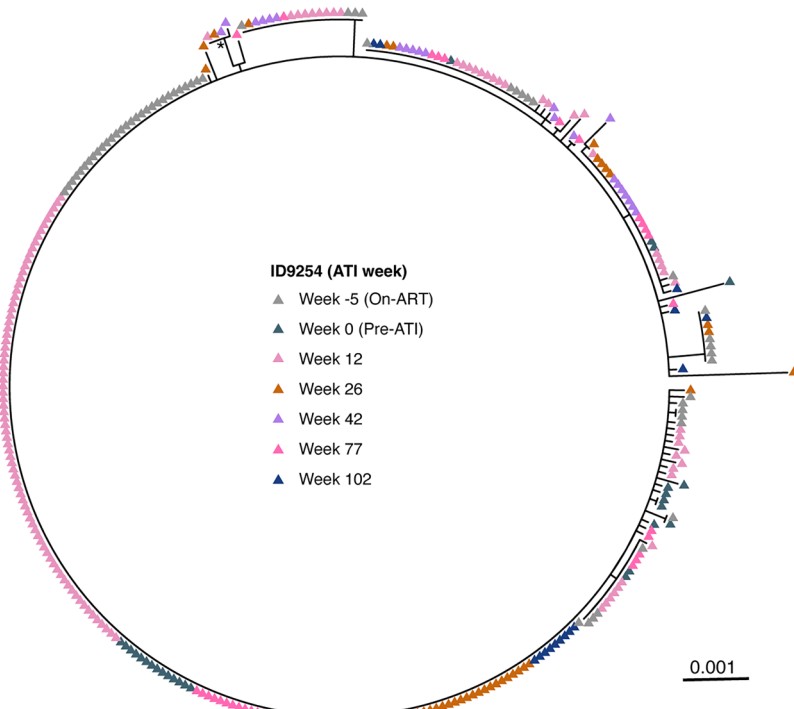

**Extended Data Fig. 1 | Phylogenetic tree of genetically intact proviruses for ID9254.** Maximum likelihood phylogenetic trees showing all genetically intact proviruses for ID9254 isolated by Q4PCR. All sequences are sourced from PBMCs and showed as triangles, with colors representing timepoint before or after ART interruption/ATI. Scale bar represents nucleotide substitutions per site. * indicates branch support value > 70%.

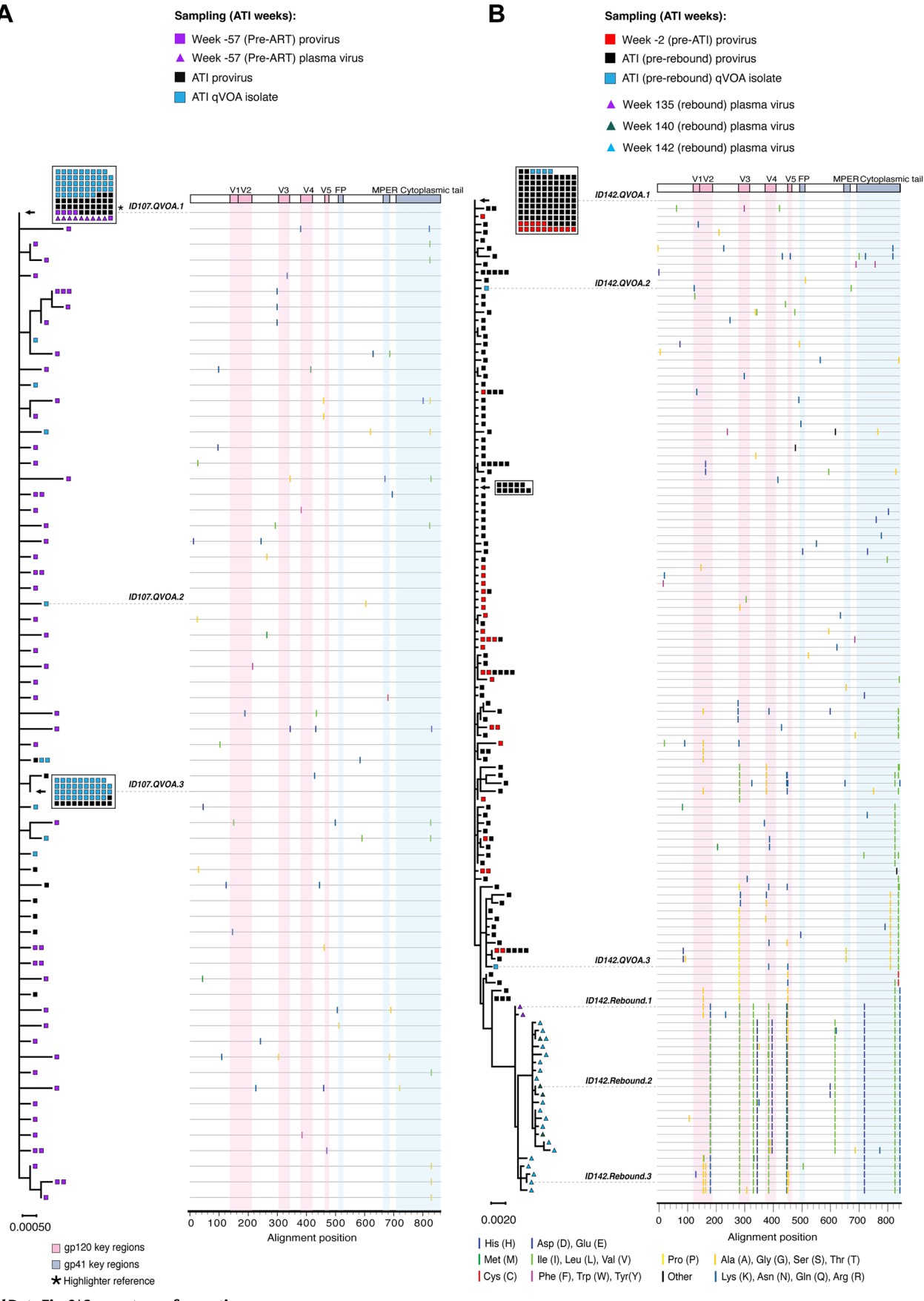

**Extended Data Fig. 2 | See next page for caption.**

**Extended Data Fig. 2 | Overview of *env* region and generation of pseudoviruses for aNAb assays for ID107 and ID142.** (**A**) Maximum likelihood phylogenetic tree of full-length HIV-1 *env* sequences from ID107 pre-ART plasma and proviral/qVOA reservoir isolates. The *env* sequences used to generate pseudoviruses ID107. QVOA.1, ID107.QVOA.2 and ID107.QVOA.3 in Fig. 3c are labeled. (**B**) Maximum likelihood phylogenetic tree of full-length HIV-1 *env* sequences from ID142 outgrowth, proviral and rebound HIV-1 isolates. The *env* sequences used to generate pseudoviruses ID142.QVOA1, ID142.QVOA2 and ID142.QVOA3 in Fig. 3e, and ID142.Rebound1, ID142.Rebound2 and ID142.Rebound3 in Fig. 7c are labeled.

Both phylogenetic trees are rooted to the dominant sequences recovered from the pre-ART (**A**) or pre-ATI (**B**) timepoint. The bars indicate substitutions per site converted to nucleotide units. Large black arrows indicate a large set of independent, identical HIV-1 *env* sequences. A highlighter plot with sequences corresponding to their position on the phylogenetic tree was also generated, using the root of the tree as the master reference. Hypervariable regions, V1-V5, of gp120 are highlighted in light pink, and key regions in gp41 (fusion peptide; FP, membrane proximal external region; MPER and cytoplasmic tail) are highlighted in light blue.

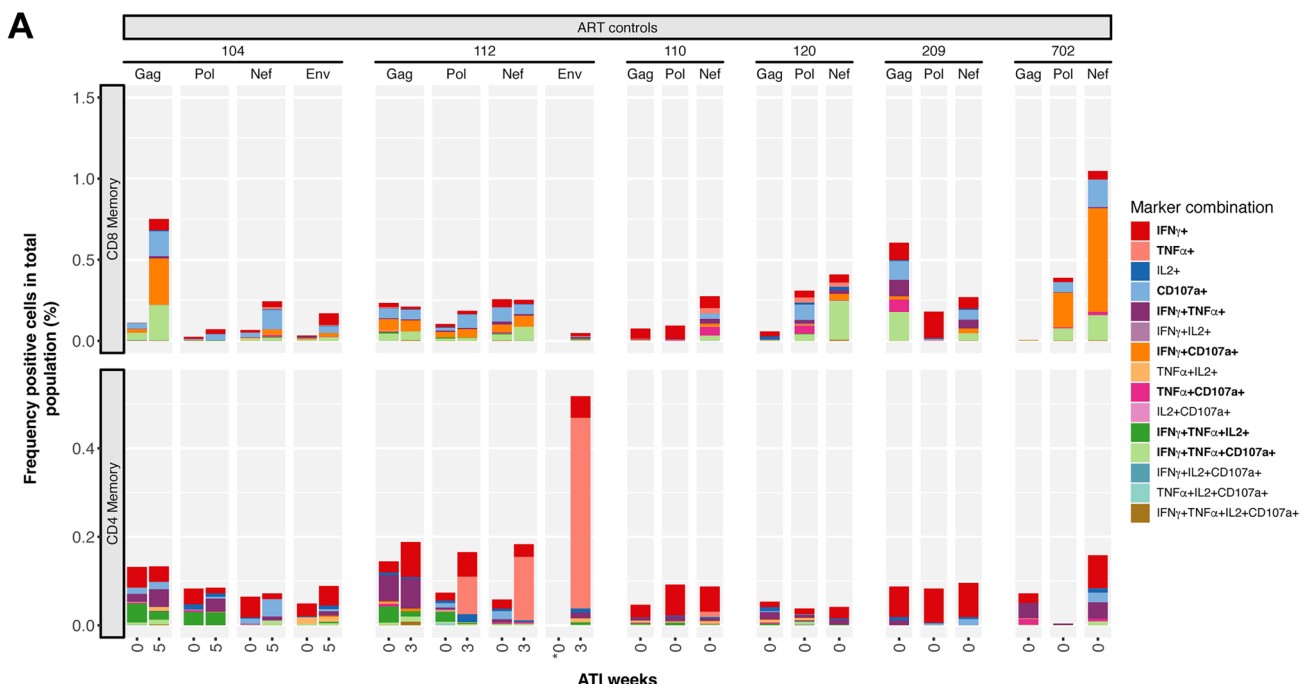

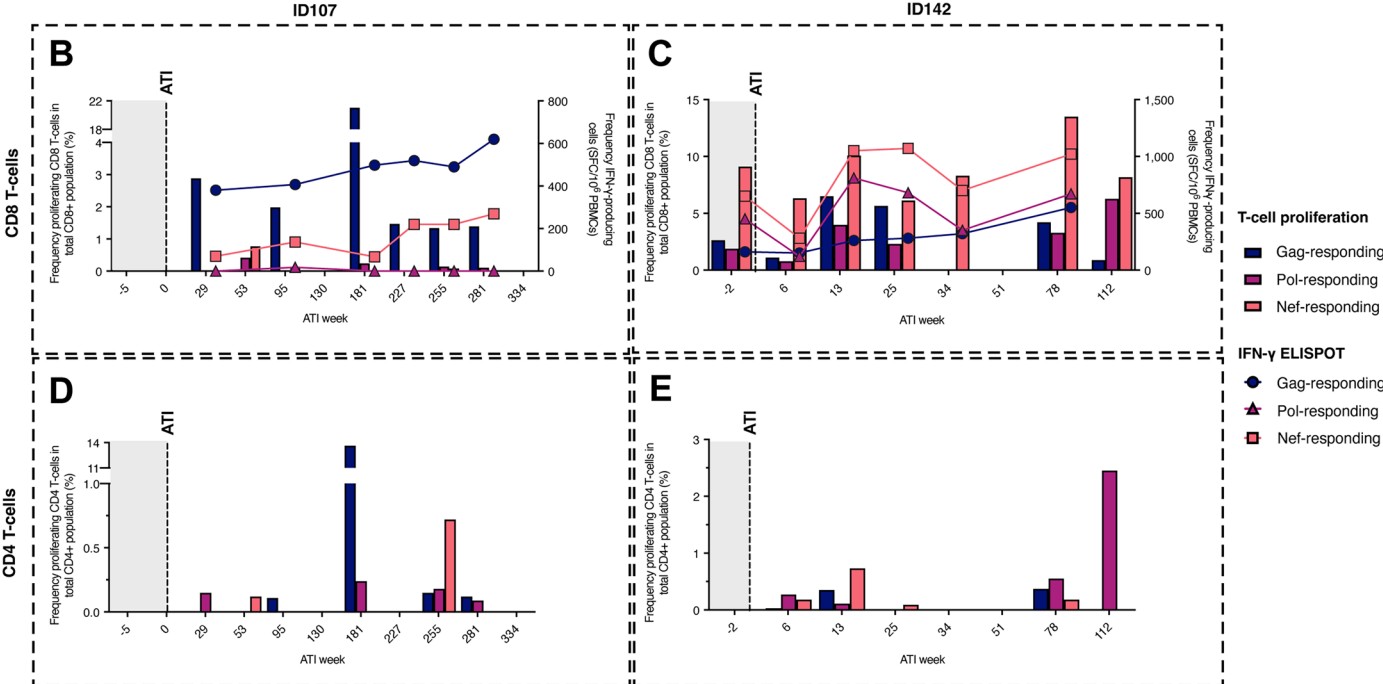

**Extended Data Fig. 3 | HIV-1-specific CD4 and CD8 T cells. (A)** Spectral flow cytometric assessment of cytokine production by memory CD8 T cells (top panel) and memory CD4 T cells (bottom panel) (see Methods and Supplementary Notes) in response to stimulation with HIV-1 Gag, Pol, Env or Nef peptide pools in ART-suppressed individuals. For ID104 and ID112, the viral rebound timepoint was also included. Color denotes positivity for cytokines and degranulation (CD107a), and the eight dominant responses are highlighted in bold in the legend. Each bar represents a single experiment per individual and timepoint, utilizing a minimum of 0.8 x 106 PBMCs per stimulation. The frequency of cells expressing each combination of cytokines is calculated by subtracting the value of responding cells to the specific stimulation by that within the negative control

(stimulation of cells with DMSO). * denotes excluded data due to insufficient number of cells in the PBMC vials. **(B-C)** HIV-1-specific CD8 T cell responses over time on- and off-ART for **(B)** ID107 and **(C)** ID142 as characterized by lymphocyte proliferation assay (column graphs; left y-axis) and IFN-γ ELISpot assay (symbols; right y-axis) in response to stimulation with HIV-1 Gag, Pol and Nef peptide pools. Each datapoint represents a single experiment per stimulation and timepoint. **(D-E)** HIV-1-specific CD4 T cell responses over time on- and off-ART for ID107 **(D)** and ID142 **(E)** as characterized by lymphocyte proliferation assay in response to stimulation with HIV-1 Gag, Pol and Nef peptide pools. Each datapoint represents a single experiment per stimulation and timepoint. Analyzed samples are color-coded according to timepoint. Gray shading indicates time on ART.

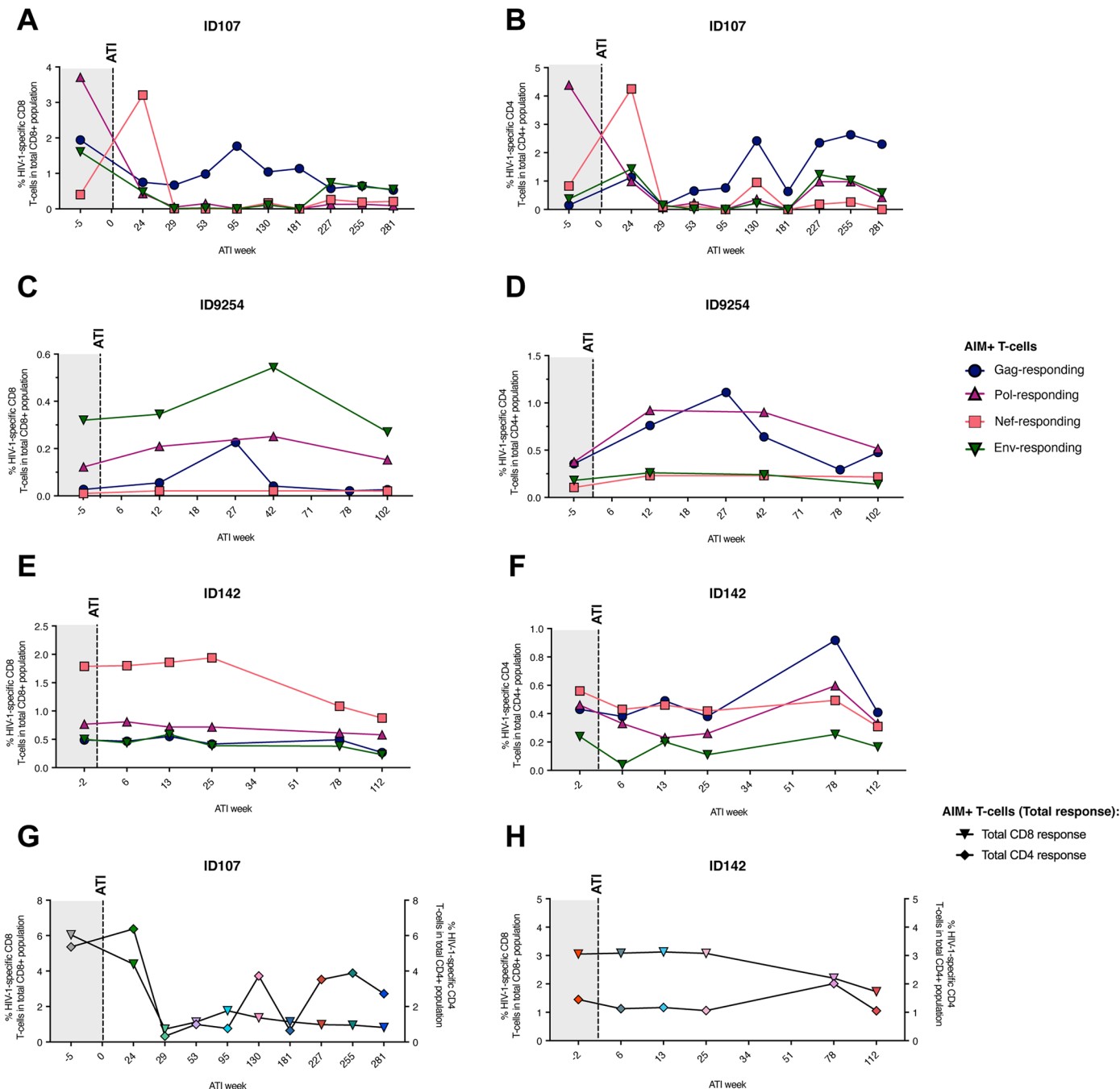

**Extended Data Fig. 4 | HIV-1-specific CD4 and CD8 T cells from ID107, ID9254 and ID142 quantified by the AIM assay.** (**A-B**) Peptide-stratified HIV-1-specific CD8 (**A**) and CD4 (**B**) T cell responses in ID107 were quantified by the AIM assay in response to stimulation with HIV-1 Gag, Pol, Nef and Env peptide pools. (**C-D**) Peptide-stratified HIV-1-specific CD8 (**A**) and CD4 (**B**) T cell responses in ID9254 were quantified by the AIM assay in response to stimulation with HIV-1 Gag, Pol, Nef and Env peptide pools. (**E-F**) Peptide-stratified HIV-1-specific CD8

(**E**) and CD4 (**F**) T cell responses in ID142 before viral rebound were quantified by the AIM assay in response to stimulation with HIV-1 Gag, Pol, Nef and Env peptide pools. (**G-H**) Total HIV-1-specific CD4 and CD8 T cell responses were plotted over time for ID107 (**G**) and ID142 before viral rebound (**H**), calculated as the sum of the responses to HIV-1 Gag, Pol and Nef. Analyzed samples are color-coded according to timepoint. Gray shading indicates time on ART. Each datapoint represents a single experiment per stimulation and timepoint.

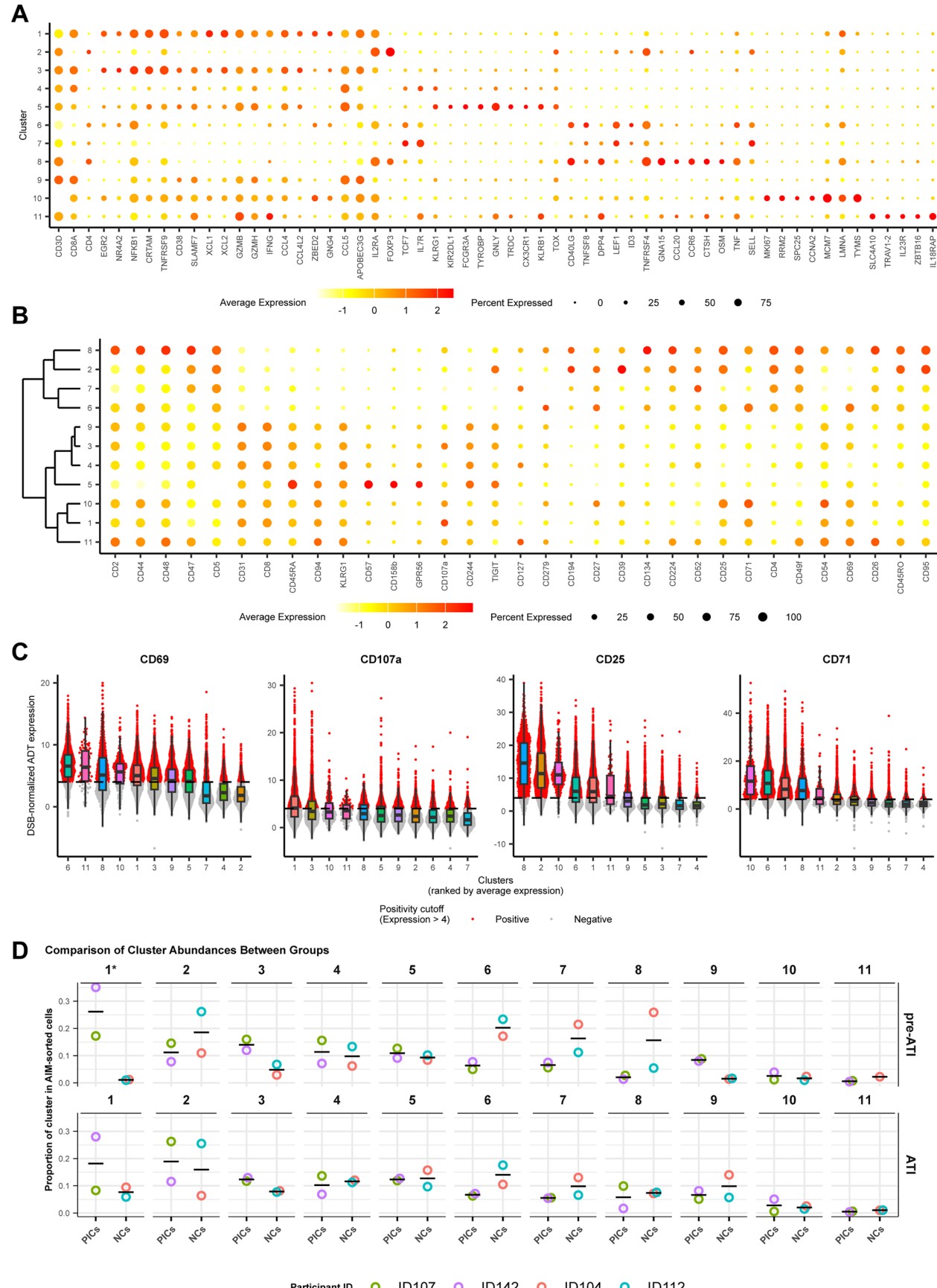

**Extended Data Fig. 5 | See next page for caption.**

**Extended Data Fig. 5 | Single-cell sequencing of AIM + T cells.** (**A**) Dot plot showing key marker gene expression per cluster. (**B**) Dot plot showing key surface protein expression per cluster. Columns (surface markers) and rows (clusters) are hierarchically clustered according to the markers' average expression, which is further illustrated for the rows by the dendrogram on the left. The percentage of cells positive for each marker was determined using a DSB-normalized positivity cutoff value of 4. (**C**) DSB-normalized expression of selected surface protein markers across clusters. Each dot represents a single cell (n = 14,243 cells from 2 PICs). Box plot depict value distributions across clusters, with boxes showing the median and interquartile range (IQR) and the whiskers extending up to 1.5 x IQR from the hinge. For each marker, clusters are ranked by average expression. Horizontal dashed lines indicate the DSB-normalized positivity cutoff value of 4. (**D**) Frequencies of each cluster in each participant before ATI (week 0 for ID107, ID104, and ID112; week −2 for ID142) and during ATI (weeks 14 for ID107, 51 for ID142, 5 for ID104, and 3 for ID112). Participants are grouped by control status, with group frequency distributions represented by horizontal lines. * Median Benjamini–Hochberg-adjusted p value = 0.002 across 100 bootstrapped replicates; Empirical Bayes moderated t-test, two-sided. Panels show data from all collected time points (ID107: ATI weeks −57, 0, 14, 29, 95, 181, 281; ID142: −2, 51, 78; ID104: −57, 0, 5; ID112: −57, 0, 3) except panel D, which includes only the latest pre-ATI time point (ATI week 0 for ID107, ID104, and ID112; ATI week −2 for ID142) and the earliest ATI time point (ID107: ATI week 14; ID142: 51; ID104: 5; ID112:3). All data represents biological replicates analyzed as single cells for cluster-level comparisons, as clonotypes for TCR repertoires, or as whole samples for condition-level comparisons. Cell counts for each cluster and donor are listed in Supplementary File 1.

## A · Clonal Rank Distribution by Cluster and Participant

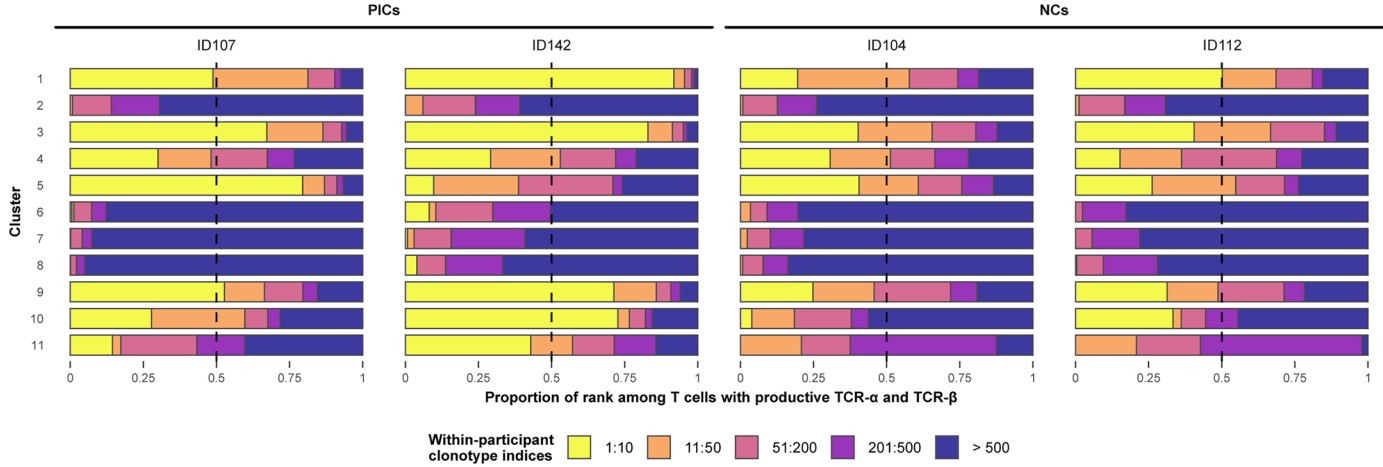

## B · Median Cluster Clonality Across Visits

## C · TCR Repertoire Overlap Between Clusters

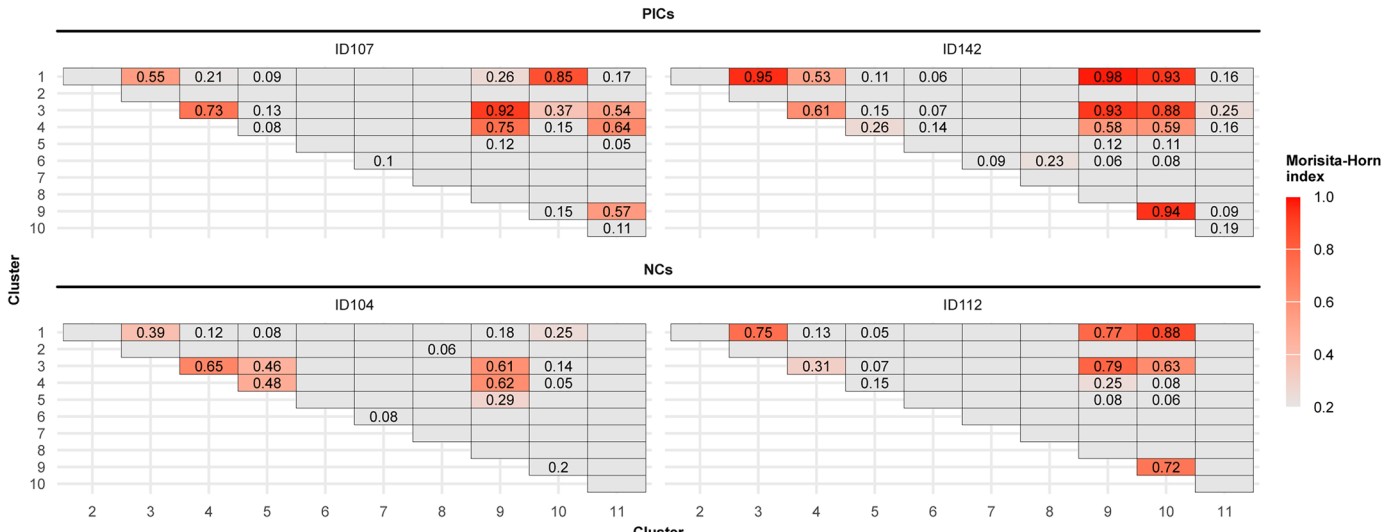

**Extended Data Fig. 6 | See next page for caption.**

**Extended Data Fig. 6 | TCR repertoire characteristics of AIM + T cells. (A)** Bar plots showing the distribution of clone rank index intervals within each cluster. Each bar represents a cluster, with fill colors indicating rank index intervals. Ranks were assigned per participant based on the total number of cells per clonotype across all visits. (**B**) Median Gini coefficient across all visits for each cluster in each participant. Participants are grouped by control status. Gini coefficients were calculated only for TCR repertoires containing at least 50 cells. Missing cluster-visit combinations were excluded from median calculations.

(**C**) Heatmap showing TCR repertoire overlap between clusters per participant, measured by the Morisita–Horn index. Morisita–Horn values are also displayed as labels. To improve readability, overlaps with values < 0.2 are shaded in gray, and values < 0.05 are not labeled. Panels show data from all collected time points (ID107: ATI weeks −57, 0, 14, 29, 95, 181, 281; ID142: −2, 51, 78; ID104: −57, 0, 5; ID112: −57, 0, 3). All data represents biological replicates analyzed at the single-cell, clonotype, or sample level, depending on the comparison.

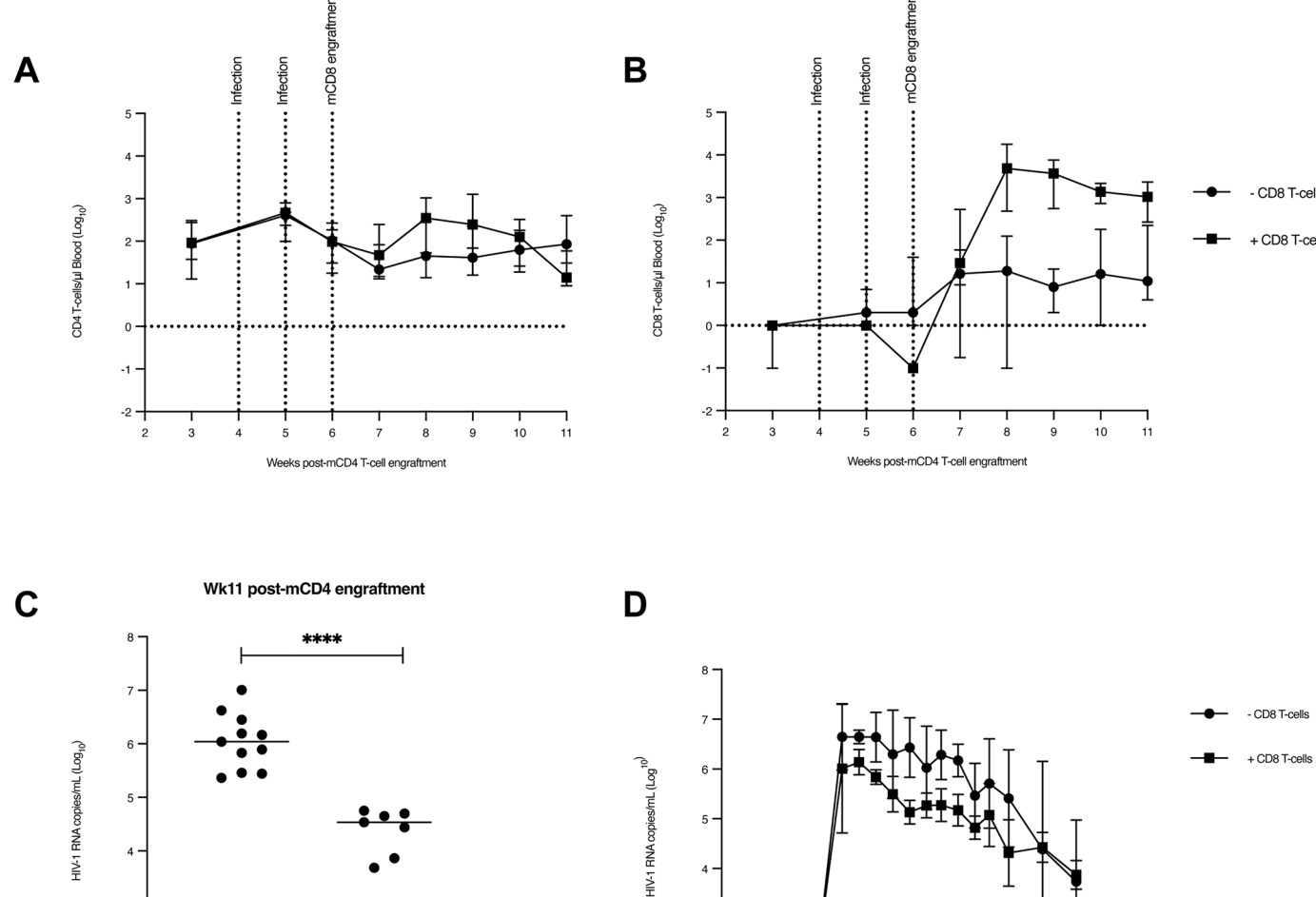

**Extended Data Fig. 7 | CD4 and CD8 T cell counts for PDX mouse experiments.**
(**A-B**) CD4 (**A**) and CD8 (**B**) T cell counts development over 11 weeks following engraftment of NSG mice with mCD4 T cells sourced from ID107. NSG mice were infected with plasma sourced from mice that experienced a spontaneous viral rebound following the mCD4 T cell engraftment, and one group of mice then also received an engraftment of mCD8 T cells sourced from ID107 ( + CD8 + T cell group). Datapoints represent the median CD4 (**A**) and CD8 (**B**) T cell counts at each timepoint for the mice within the – CD8 T cells group and the + CD8 T cells group (n = 11 and n = 7 biological replicates, respectively) and error bars represent the IQR. (**C**) Comparison of plasma HIV-1 RNA viral load between mice in the – CD8 + T cells group and the + CD8 + T cells group at week 11 post-mCD4

T cell engraftment. Bars represent the median plasma HIV-1 RNA load. The statistical comparison between the plasma HIV-1 RNA load of the individual mice at week 11 post-mCD4 T cell engraftment in the – CD8 T cells group and the + CD8 T cells group (n = 11 and n = 7 biological replicates, respectively) was performed using an unpaired t-test (two-tailed) on log-transformed data. (**D**) NSG mice were engrafted with mCD4 T cells sourced from a progressor individual with HIV-1, with or without mCD8 T cells sourced from the same individual. The mice were infected with HIV-JRCSF, and the viral load was monitored for 22 weeks following mCD4 + T cell engraftment. Datapoints represent the median viral load at each timepoint for all mice within the – CD8 T cells group and the + CD8 T cells group (n = 6 and n = 5 biological replicates, respectively) and error bars represent ± IQR.

**A**

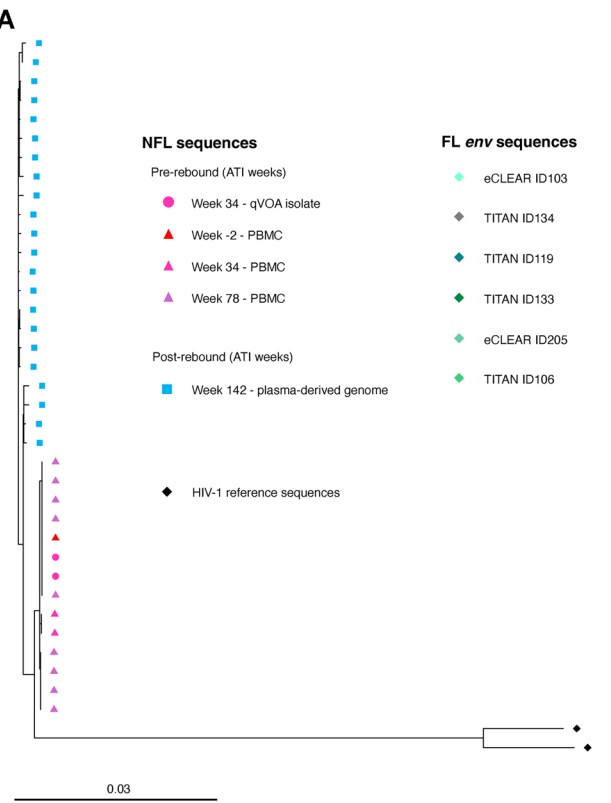

**NFL sequences**

Pre-rebound (ATI weeks)

- ● Week 34 - qVOA isolate
- ▲ Week -2 - PBMC
- ▲ Week 34 - PBMC
- ▲ Week 78 - PBMC

Post-rebound (ATI weeks)

- ■ Week 142 - plasma-derived genome

- ◆ HIV-1 reference sequences

**FL *env* sequences**

- ◆ eCLEAR ID103
- ◆ TITAN ID134
- ◆ TITAN ID119
- ◆ TITAN ID133
- ◆ eCLEAR ID205
- ◆ TITAN ID106

0.03

**B**

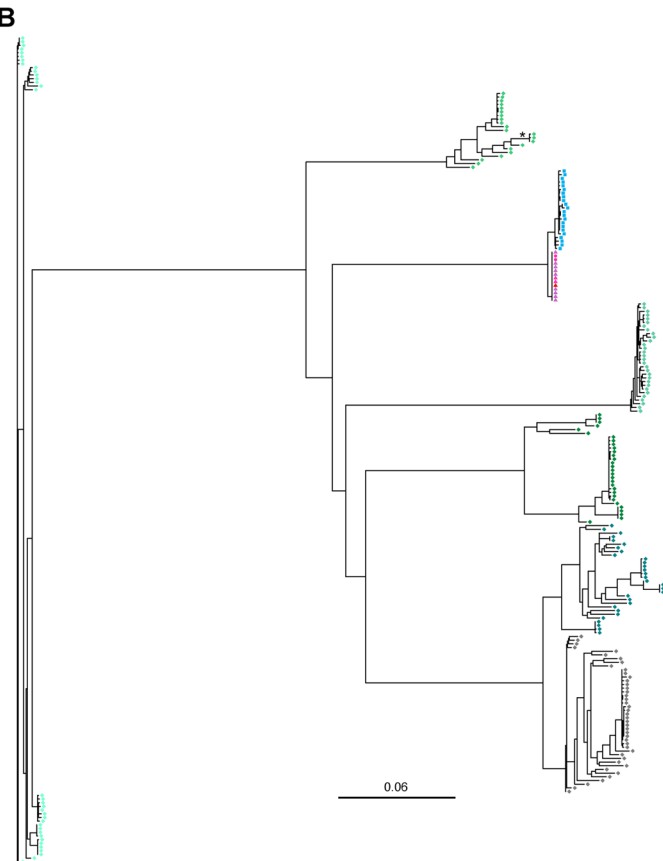

0.06

**Extended Data Fig. 8 | Viral rebound in ID142 is not caused by a superinfection.** (**A**) Maximum likelihood phylogenetic tree of all near-full-length sequences isolated from ID142 pre-rebound and rebound timepoints aligned to the lab strain viruses NL4-3 and HXB2 and trimmed to the overlapping region (~8.7 kb). (**B**) Maximum likelihood phylogenetic tree of all NFL sequences isolated from

ID142 pre-rebound and rebound timepoints trimmed to full-length *env* and aligned with *env* sequences sourced from six other individuals with HIV-1 subtype B. * indicates branch support of > 70%. Scale bars indicate nucleotide substitutions per site.

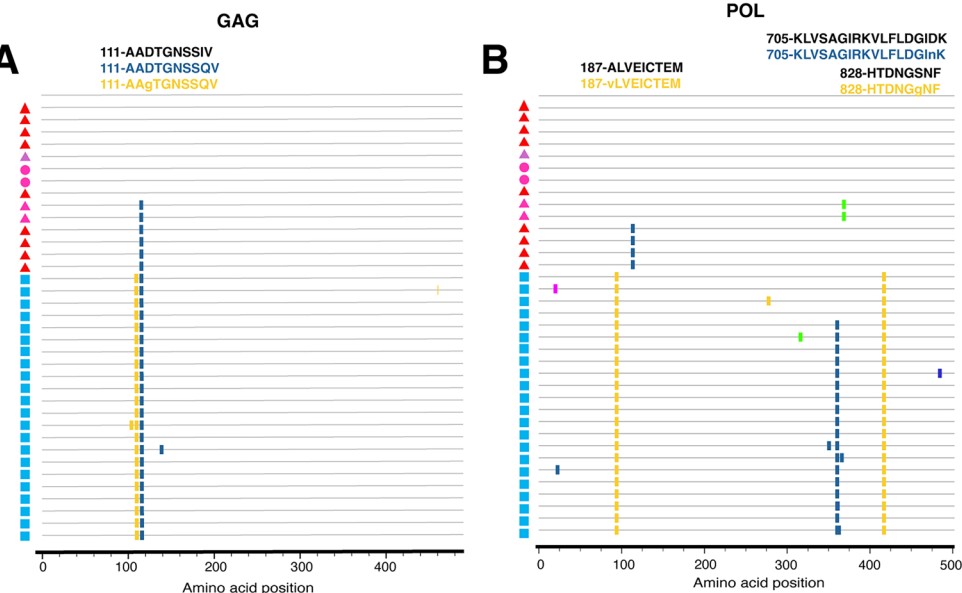

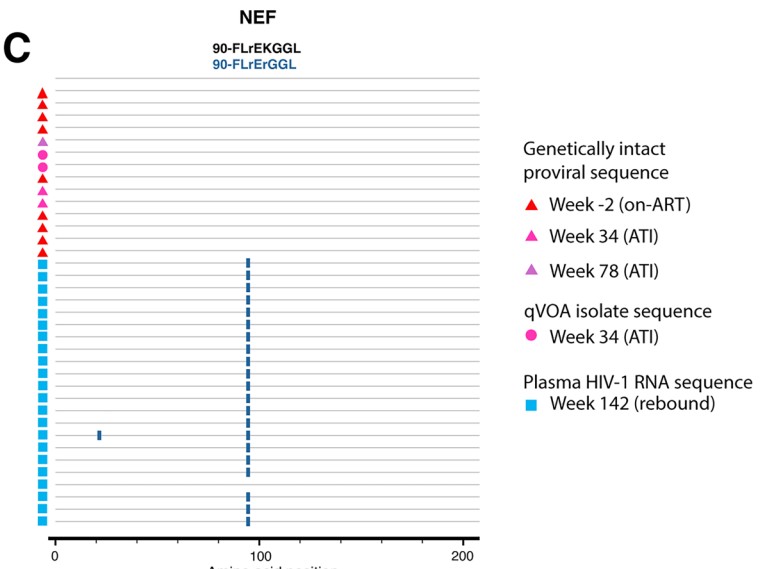

**Extended Data Fig. 9 | HLA-matched CD8 T cell epitope changes in ID142 sequences isolated pre-rebound and during rebound.** Highlighter plots for Gag (**A**), Pol (**B**) and Nef (**C**) protein sequences from pre- and post-rebound near-full-length sequences. Sequences are ordered as shown in the phylogenetic tree in Fig. 7b. Each ID142 sequence (indicated by colored symbol) is compared to the master sequence, which is a consensus of all pre-rebound sequences. Identified HLA-matched CTL epitopes are reported above the graphs, with the epitope sequence found in the master shown in black, and the mutated epitope shown in color. Sequences that carry the mutated epitope are indicated by a line in the same color underneath the epitope.

**Extended Data Table 1 | Identified HLA-matched wild type and escape CTL epitopes in ID142 HIV-1 sequences**

| Epitope | HLA restriction | Wild type/Escape documented |
|---|---|---|
| Gag | | |
| 66-(K)SLYNTIATL(Y) | A*02 | Escape |
| 150-EEKAFSPEV | B*44 | Wild type |
| 250-EIYKRWII | B*08 | Wild type |
| 284-(RD)YVDRFYKTL | A*02, B*44 | Wild type |
| 296-AEQASQEVKNWM | C*05 | Wild type |
| 319-DCKTILKAL | B*08 | Wild type |
| Pol | | |
| 172-GPKVKQWPL | B*08 | Wild type |
| 281-YTAFTIPSI | A*02 | Wild type |
| 333-VIYQYMDDL | A*02 | Wild type |
| 463-ILKEPVHGV | A*02 | Wild type |
| 973-(VVPR)RKAKIIRDYGK | B*08 | Wild type |
| Nef | | |
| 83-GALDLSHFL | A*02 | Escape |
| 187-SRLAFHHMA | B*08 | Wild type |

# Reporting Summary

## Statistics

For all statistical analyses, confirm that the following items are present in the figure legend, table legend, main text, or Methods section.

| n/a | Confirmed | |
|---|---|---|
| ☐ | ☒ | The exact sample size (*n*) for each experimental group/condition, given as a discrete number and unit of measurement |
| ☐ | ☒ | A statement on whether measurements were taken from distinct samples or whether the same sample was measured repeatedly |
| ☐ | ☒ | The statistical test(s) used AND whether they are one- or two-sided *Only common tests should be described solely by name; describe more complex techniques in the Methods section.* |
| ☐ | ☒ | A description of all covariates tested |
| ☐ | ☒ | A description of any assumptions or corrections, such as tests of normality and adjustment for multiple comparisons |
| ☐ | ☒ | A full description of the statistical parameters including central tendency (e.g. means) or other basic estimates (e.g. regression coefficient) AND variation (e.g. standard deviation) or associated estimates of uncertainty (e.g. confidence intervals) |
| ☐ | ☒ | For null hypothesis testing, the test statistic (e.g. *F*, *t*, *r*) with confidence intervals, effect sizes, degrees of freedom and *P* value noted *Give P values as exact values whenever suitable.* |
| ☐ | ☒ | For Bayesian analysis, information on the choice of priors and Markov chain Monte Carlo settings |
| ☒ | ☐ | For hierarchical and complex designs, identification of the appropriate level for tests and full reporting of outcomes |
| ☒ | ☐ | Estimates of effect sizes (e.g. Cohen's *d*, Pearson's *r*), indicating how they were calculated |

*Our web collection on statistics for biologists contains articles on many of the points above.*

## Software and code

Policy information about availability of computer code

| | |
|---|---|
| Data collection | QuantaSoft software (Bio-Rad, version 1.7.4), 5-laser Sony ID7000 Spectral Analyser (SONY Biotechnologies, San Jose, CA), QuantStudio 7 Pro Real-Time PCR system (Applied Biosystems, A43183), LSRII flow cytometer (BD Biosciences) |
| Data analysis | Custom code, utilising published R packages, was used in this manuscript to analyse spectral flow cytometry data and paired single-cell transcriptome and TCR sequencing data. These codes can be found at the following page: https://github.com/SoegaardLab/2025_Fisher_Garcia_Frattari_Naasz_et_al. For analysis of spectral flow cytometry data, the following packages were used: R v.4.4.0, R version 4.3.2 with RStudio version 2023.12.1.402, flowCore v2.20.0 (https://github.com/RGLab/flowCore), flowWorkspace v4.20.0 (https://github.com/RGLab/flowWorkspace), ggcyto v1.36.1 (https://github.com/RGLab/ggcyto), PeacoQC v1.18.0 (https://github.com/saeyslab/PeacoQC), CytoExploreR v1.1.0 (https://github.com/DillonHammill/CytoExploreR), openCyto v2.20.1 (https://github.com/RGLab/openCyto), CATALYST v1.32.1 (https://github.com/HelenaLC/CATALYST), SingleCellExperiment v1.30.1 (https://github.com/drisso/SingleCellExperiment), SummarizedExperiment v1.38.1 (https://bioconductor.org/packages/release/bioc/html/SummarizedExperiment.html), FlowSOM v2.16.0 (https://www.bioconductor.org/packages/release/bioc/html/FlowSOM.html), uwot v0.2.3 (https://github.com/jlmelville/uwot) and scater v1.35.0 (https://www.bioconductor.org/packages/release/bioc/html/scater.html). For analysis of paired single-cell transcriptome and TCR sequencing data, the following packages were used: Cell Ranger v8.0.1 (10x Genomics; https://www.10xgenomics.com/support/software/cell-ranger/latest), Scanpro 0.4.0 (https://github.com/loosolab/scanpro), scDblFinder v1.18.0 (https://github.com/plger/scDblFinder), SoupX v1.6.2 (https://github.com/constantAmateur/SoupX), Seurat v5 (https://github.com/satijalab/seurat), Harmony v1.2.3 (https://github.com/immunogenomics/harmony), MAST v1.30.0 (https://github.com/RGLab/MAST), clusterProfiler v4.12.6 (https://github.com/YuLab-SMU/clusterProfiler), enrichplot v1.24.4 (https://github.com/YuLab-SMU/enrichplot), DSB v2.0.0 (https://github.com/niaid/dsb), DescTools v0.99.56 (https://cran.r-project.org/web/packages/DescTools/index.html) and Aarhus University HPC-facility GenomeDK (https://genome.au.dk/). An in-house pipeline was used for assembly of NFL HIV sequences generated by PRLS was used (https://github.com/laulambr/virus_assembly), utilising multiqc v1.23 (https://github.com/MultiQC/MultiQC), bbmap v39.01 (https:// |

github.com/BioInfoTools/BBMap), samtools v1.18 (https://github.com/samtools/samtools), qualimap v2.3 (https://anaconda.org/channels/bioconda/packages/qualimap/overview), megahit v1.2.9 (https://github.com/voutcn/megahit), blast v1.16.0, mafft v7.520 (https://mafft.cbrc.jp/alignment/software/), and seqtk v1.4 (https://github.com/lh3/seqtk). An in-house pipeline for assembly and classification of HIV NFL sequences by Q4PCR (https://github.com/stratust/DIHIVA), utilising BBtools package v38.72 (https://sourceforge.net/projects/bbmap/), Trim Galore package v0.6.4 (https://github.com/FelixKrueger/TrimGalore) and SPAdes v3.13.1 (https://github.com/ablab/spades). An in-house intactness pipeline was used for classification of NFL HIV sequences generated by MIP-seq (https://github.com/BWH-Lichterfeld-Lab/Intactness-Pipeline). Additional code used in the paper has been cited appropriately.  Additional tools and packages used for analysis: bwa-mem v0.7.19 (https://github.com/lh3/bwa), Ensembl (v113, www.ensembl.org), the UCSC Genome Browser (www.genome.ucsc.edu), GENCODE (v47, www.gencodegenes.org), Los Alamos webtools ElimDupes (https://www.hiv.lanl.gov/content/sequence/elimdupesv2/elimdupes.html), GeneCutter (https://www.hiv.lanl.gov/content/sequence/GENE_CUTTER/cutter.html), Hypermut v3.0 (https://www.hiv.lanl.gov/content/sequence/HYPERMUT/hypermutv3.html) and Highlighter (https://www.hiv.lanl.gov/content/sequence/HIGHLIGHT/highlighter_top.html), and the HIV Molecular Immunology Database Search (https://www.hiv.lanl.gov/mojo/immunology/search/ctl/form.html), PhyML v3.3.20220408 (https://github.com/stephaneguindon/phyml), ggTree v3.12.0 (https://guangchuangyu.github.io/software/ggtree/), Geneious Prime v2025.1.2, FlowJo v10.10.0 and v10.5.0 Software (BD Biosciences), ID7000 Software version 2.0.2 (SONY Biotechnologies, San Jose, CA), GraphPad Prism v 10.2.0, IUPM algorithm (http://silicianolab.johnshopkins.edu), bNAb-ReP v1.1-4 (https://github.com/RedaRawi/bNAb-ReP), the Los Alamos "Best-defined CTL/CD8+ Epitope Summary" (https://www.hiv.lanl.gov/content/immunology/tables/optimal_ctl_summary.html), Los Alamos "CTL/CD8+ Epitope Variants and Escape Mutations" table (http://www.hiv.lanl.gov/content/immunology/variants/ctl_variant.html) and the Biopython SeqUtils package (https://github.com/biopython/biopython/tree/master/Bio/SeqUtils).

For manuscripts utilizing custom algorithms or software that are central to the research but not yet described in published literature, software must be made available to editors and reviewers. We strongly encourage code deposition in a community repository (e.g. GitHub). See the Nature Portfolio guidelines for submitting code & software for further information.

# Data

Policy information about availability of data

All manuscripts must include a data availability statement. This statement should provide the following information, where applicable:

- Accession codes, unique identifiers, or web links for publicly available datasets
- A description of any restrictions on data availability
- For clinical datasets or third party data, please ensure that the statement adheres to our policy

HIV-1 proviral, qVOA and plasma-derived sequences have been deposited in GenBank with the accession numbers PX465444-PX466083, MK115946-MK116091, MN090734-MN090850 and MW063053-MW063065. HIV-1 env sequences utilised to check for superinfection can be found in Genbank with the accession numbers OR014534-OR014555, OR014635-OR014658, OR014662-OR014681, OR014984-OR015005, OR015084-OR015110, OR015113-OR015155 and PX892135-PX892225. Paired single-cell transcriptome and TCR sequencing data has been uploaded to the European Genome-Phenome Archive (accession number EGAS50000001570). Access to data will be assessed on a case-by-case basis and available upon signing of a Data Access Agreement. Any additional information required can be accessed by contacting the corresponding author, Ole Schmeltz Søgaard (olesoega@rm.dk).

# Research involving human participants, their data, or biological material

Policy information about studies with human participants or human data. See also policy information about sex, gender (identity/presentation), and sexual orientation and race, ethnicity and racism.

| Reporting on sex and gender | People living with HIV-1 included in this manuscript included three post-intervention controllers and six ART-suppressed individuals/non-controllers, 5/6 of which were of male sex (self-reported/identified from medical records). One additional progressor individual with HIV-1 was also included in the manuscript, who was of female sex. Due to the nature of the study investigating post-intervention control of HIV-1, investigations could only utilise sampling from identified post-intervention controllers for whom samples were available, all of which were male. Secondary use of samples from ART-suppressed individuals who participated in the eCLEAR trial was restricted by availability of biological sampling. |
|---|---|
| Reporting on race, ethnicity, or other socially relevant groupings | Due to the nature of our investigation into post-intervention control of HIV-1, only a small number of post-intervention controllers (and an appropriate number of non-controlling individuals) were included in the study. This is because investigations could only utilise sampling from identified post-intervention controllers for whom samples were available. Information regarding the race/ethnicity of all individuals included in the Methods section and Supplementary Tables 1 and 5 of the manuscript. |
| Population characteristics | Our manuscript is based on the identification of three cases of HIV-1 post-intervention control. Therefore, no specific measures could be taken to ensure equal distribution of population characteristics as investigation was based on sample availability. Population characteristics of all individuals can be found in Supplementary Tables 1 and 5. |
| Recruitment | Our manuscript is based on the identification of three cases of HIV-1 post-intervention control. Therefore, specific recruitment into this study is not relevant. |
| Ethics oversight | National Committee on Health Research Ethics in Denmark (continued ATI, leukapheresis and secondary use of samples from eCLEAR clinical trial), Ethics Committee of the Medical Faculty of the University of Cologne, the George Washington University Institution Review Board and Weill Cornell Medicine Institutional Review Board. Written informed consent was given by all individuals. |

Note that full information on the approval of the study protocol must also be provided in the manuscript.

# Field-specific reporting

Please select the one below that is the best fit for your research. If you are not sure, read the appropriate sections before making your selection.

☒ Life sciences     ☐ Behavioural & social sciences     ☐ Ecological, evolutionary & environmental sciences

For a reference copy of the document with all sections, see nature.com/documents/nr-reporting-summary-flat.pdf

# Life sciences study design

All studies must disclose on these points even when the disclosure is negative.

| | |
|---|---|
| Sample size | Sample size calculations are not appropriate for our study as the design was based on three identified HIV-1 post-intervention controllers. |
| Data exclusions | Data were only excluded on the basis of insufficient material available to run assays at certain timepoints. |
| Replication | All assays were performed once at each timepoint due to limited biological sample availability, except for neutralisation experiments which were performed using duplicates or triplicates. However, multiple assays were used at multiple timepoints to corroborate our findings and claims. For the participant-derived xenograft experiments, a total of 19 mice were included in the study. |
| Randomization | For all experiments involving sampling from people with HIV-1, randomisation was not appropriate as the study was based on three identified HIV-1 post-intervention controllers, and included six ART-suppressed/non-controller individuals for comparison. For the participant-derived xenograft experiments, mice were divided into two groups (one to receive autologous memory CD8 T cells from ID107 and one who did not receive memory CD8 T cells) following infection of all mice with autologous virus. The division of the two groups ensured equal distribution of plasma HIV-1 viral loads and CD4 T cell counts following establishment of infection, and therefore was not randomised. |
| Blinding | All participants were assigned unique IDs prior to participation in the previous clinical trials, and these were utilised while conducting the experiments included in this manuscript. These IDs were used for identification of all samples while experiments were conducted, and therefore blinding was not relevant to this manuscript. |

# Reporting for specific materials, systems and methods

We require information from authors about some types of materials, experimental systems and methods used in many studies. Here, indicate whether each material, system or method listed is relevant to your study. If you are not sure if a list item applies to your research, read the appropriate section before selecting a response.

## Materials & experimental systems

| n/a | Involved in the study |
|---|---|
| ☐ | ☒ Antibodies |
| ☐ | ☒ Eukaryotic cell lines |
| ☒ | ☐ Palaeontology and archaeology |
| ☐ | ☒ Animals and other organisms |
| ☐ | ☒ Clinical data |
| ☒ | ☐ Dual use research of concern |
| ☒ | ☐ Plants |

## Methods

| n/a | Involved in the study |
|---|---|
| ☒ | ☐ ChIP-seq |
| ☐ | ☒ Flow cytometry |
| ☒ | ☐ MRI-based neuroimaging |

# Antibodies

| | |
|---|---|
| Antibodies used | 1. CD3: PE/Dazzle 594, OKT3, Biolegend, catalog #317345, lot #B443448<br>2. CD4: PE/Fire 700, SK3, Biolegend, catalog #344665, lot #B437043<br>3. CD8: APC, SK1, Biolegend, catalog #344721, lot #B441271<br>4. CD3: PerCP/Cy5.5, SK7, BioLegend, catalog #300429, lot #B354280<br>5. CD4: BV650, RPA-T4, BioLegend, catalog #300535, lot #B366745<br>6. CD8: BV605, RPA-T8, BioLegend, catalog #301039, lot #B435856<br>7. 4-1BB: PE, 4B4-1, BioLegend, catalog #309803, lot #B367290<br>8. CD69: APC, FN50, BioLegend, catalog #310909, lot #B359249<br>9. PD-L1: BV421, B7-H1, BioLegend, catalog #374507, lot #B388684<br>10. IL2: BUV395, MQ1-17H12, eBioscience, catalog #363-7029-42, lot #2851058<br>11. CD4: BUV496, SK3, BD bioscience, catalog #612936, lot #3247822, 4204806<br>12. CD16: BUV563, 3G8, BD bioscience, catalog #568289, lot #4095102, 4124989, 4185605<br>13. CD3: BUV615, SK7, BD bioscience, catalog #751252, lot #4243045, 4243046, 4243047<br>14. CXCR5: BUV661, RF8B2, BD bioscience, catalog #741559, lot #5058675<br>15. Tbet: BUV737, O4-46, BD bioscience, catalog #568166, lot #4102178, 4178340<br>16. IFNg: BV421, B27, BD bioscience, catalog #562988, lot #4234096<br>17. CCR4: BV510, L291H4, Biolegend, catalog #359416, lot #B377907<br>18. CD19: BV570, HIB19, Biolegend, catalog #302236, lot #B386458 |

19. TNFa, BV605, Mab11, Biolegend, catalog #502936, lot #B422747
20. CD27: BV650, O323, Biolegend, catalog #302828, lot #B375145, B412778
21. PD-1: BV711, EH12.2H7, Biolegend, catalog #329928, lot #B410254
22. CCR7: BV785, G043H7, Biolegend, catalog #353230, lot #B415814
23. Granulysin: AF488, RB1, BD bioscience, catalog #558254, lot #333150
24. CD8: Sparkblue-574, SK1, Biolegend, catalog #344786, lot #B430783
25. Granzyme B: PerCP, QA18A28, Biolegend, catalog #396416, lot #B429745
26. TCF1: Realblue705, S33-966, BD bioscience, catalog #570635, lot #418397
27. CD45RA: Realblue780, HI100, BD bioscience, catalog #569081, lot #3341167, 4151115
28. IL4: PE, MP4-25D2, Biolegend, catalog #500810, lot #255580
29. IL13: PE, JES10-5A2, Biolegend, catalog #501903, lot #3109515, B380784
30. Ki67: Pe-dazzle594, ki67, Biolegend, catalog #35053431, lot #B376072
31. CD95: PE-fire640, DX2, Biolegend, catalog #305658, lot #B420907
32. Granzyme K: PE-Cy7, GM26E7, Biolegend. Catalog #370516, lot #B424505
33. CD39: Pe-fire810, A1, Biolegend, catalog #328245, lot #B427223
34. CD107a: APC, H4A3, Miltenyi, catalog #130-119-869, lot #5240808332
35. FoxP3: SparkNIR-685, 206D, Biolegend, catalog #320130, lot #B42255
36. TIGIT: R718, TgMab-2, BD bioscience, catalog #569038, lot #4304734
37. Perforin: APC-fire750, B-D48, Biolegend #353318, lot #B400537
38. CD3: PerCP/eFluor710, SK7, eBioscience #46-0036, lot #1941534
39. CD4: BUV496, SK3, BD Biosciences #564651, lot #9080989
40. CD8: BV711, RPA-T8, BioLegend #301044, lot #B237121
41. 4-1BB: PE-Cy7, 4B4-1, BioLegend #309818, lot #B258325
42. CD69: BUV395, FN50, BD Biosciences #564364, lot #564364
43. PD-L1: BV421, 29E2A3, BioLegend #329714, lot #B258010

Validation

All verifications below are as found on the relevant manufacturers' website.
1. CD3 PE/Dazzle 594; Verified reativity: Human (Biolegend); Application: Flow cytometry (Quality tested, Biolegend)
2. CD4 PE/Fire 700; Verified reativity: Human (Biolegend); Application: Flow cytometry (Quality tested, Biolegend)
3. CD8 APC, SK1; Verified reativity: Human, Cynomolgus, Rhesus (Biolegend); Application: Flow cytometry (Quality tested, Biolegend)
4. CD3 PerCP/Cy5.5, SK7; Verified reativity: Human (Biolegend); Application: Flow cytometry (Quality tested, Biolegend)
5. CD4 BV650, RPA-T4; Verified reativity: Human (Biolegend); Application: Flow cytometry (Quality tested, Biolegend)
6. CD8 BV605, RPA-T8; Verified reativity: Human, Cynomolgus, Rhesus (Biolegend); Application: Flow cytometry (Quality tested, Biolegend)
7. 4-1BB PE, 4B4-1; Verified reativity: Human (Biolegend); Application: Flow cytometry (Quality tested, Biolegend)
8. CD69 APC, FN50; Verified reativity: Human (Biolegend); Application: Flow cytometry (Quality tested, Biolegend)
9. PD-L1 BV421, B7-H1; Verified reativity: Human (Biolegend); Application: Flow cytometry (Quality tested, Biolegend)
10. IL2: BUV395. Human. Applications Tested: Tested by intracellular staining followed by flow cytometry
11. CD4: BUV496. Human (QC Testing), Flow cytometry (Routinely Tested)
12. CD16: BUV563. Human (QC Testing), Rhesus, Cynomolgus, Baboon (Tested in Development). Flow cytometry (Routinely Tested)
13. CD3: BUV615. Human (Tested in Development). Flow cytometry (Qualified)
14. CXCR5: BUV661. Human (Tested in Development). Flow cytometry (Qualified)
15. Tbet: BUV737. Human (QC Testing), Mouse (Tested in Development). Intracellular staining (flow cytometry) (Routinely Tested)
16. IFNg: BV421. Human (QC Testing), Rhesus, Cynomolgus, Baboon (Tested in Development). Intracellular staining (flow cytometry) (Routinely Tested)
17. CCR4: BV510. Verified Reactivity Human. Flow cytometry (Qualified)
18. CD19, BV570. Verified Reactivity Human. Flow cytometry (Qualified)
19. TNFa, BV605. Verified Reactivity Human. Intracellular Flow cytometry (Qualified)
20. CD27: BV650. Verified Reactivity Human, Cynomolgus, Rhesus. Flow cytometry (Qualified)
21. PD-1: BV711. Verified Reactivity Human. Flow cytometry (Qualified).
22. CCR7: BV785. Verified Reactivity Human. Flow cytometry (Qualified).
23. Granulysin: AF488. Human (QC Testing). Intracellular staining (flow cytometry) (Routinely Tested)
24. CD8: Sparkblue-574. Verified Reactivity Human, Cynomolgus, Rhesus. Flow cytometry (Qualified).
25. Granzyme B: PerCP. Verified Reactivity Human, mouse. Intracellular Flow cytometry (Qualified)
26. TCF1: Realblue705. Mouse (QC Testing), Human (Tested in Development). Intracellular staining (flow cytometry) (Routinely Tested)
27. CD45RA: Realblue780. Human (QC Testing), Flow cytometry (Routinely Tested)
28. IL4: PE. Verified Reactivity Human. Intracellular Flow cytometry (Qualified)
29. IL13: PE. Verified Reactivity Human. Intracellular Flow cytometry (Qualified)
30. Ki67: Pe-dazzle594. Verified Reactivity Human. Intracellular Flow cytometry (Qualified)
31. CD95: PE-fire640. Verified Reactivity Human, Cynomolgus, Rhesus. Flow cytometry (Qualified).
32. Granzyme K: PE-Cy7. Verified Reactivity Human. Intracellular Flow cytometry (Qualified)
33. CD39: Pe-fire810. Verified Reactivity Human, Cynomolgus, Rhesus. Flow cytometry (Qualified).
34. CD107a: APC. Species tested: Human. Compatible applications. IF, IHC, ICC.
35. FoxP3: SparkNIR-685. Verified Reactivity Human. Intracellular Flow cytometry (Qualified)
36. TIGIT: R718. Human (QC Testing), Flow cytometry (Routinely Tested)
37. Perforin: APC-fire750. Verified Reactivity Human. Intracellular Flow cytometry (Qualified)
38. CD3: Verified reactivity: Human, Chimpanzee. Application: Flow Cytometry.
39. CD4: Verified reactivity: human. Application: Flow cytometry (routinely tested)
40. CD8: Verified reactivity: Human, Cynomolgus, Rhesus. Application: Flow Cytometry (Quality tested)
41. 4-1BB: Verified reactivity: Human. Application: Flow Cytometry (Quality Tested)
42: CD69: Reactivity: Human (QC Testing), Rhesus, Cynomolgus, Baboon (Tested in Development). Application: Flow Cytometry (Routinely Tested)
43: PD-L1: Verified reactivity: Human. Application: Flow Cytometry (Quality Tested)

# Eukaryotic cell lines

Policy information about cell lines and Sex and Gender in Research

| | |
|---|---|
| Cell line source(s) | TZM-bl cells (Cat No. 8129) and MOLT-4/CCR5 cell lines (Cat No 4984) were both obtained from the NIH HIV Reagent Program (https://www.beiresources.org/HIV.aspx). |
| Authentication | The cell lines above were not authenticated. |
| Mycoplasma contamination | TZM-bl cells and MOLT-4/CCR5 cell lines are negative for mycoplasma. |
| Commonly misidentified lines (See ICLAC register) | No commonly misidentified cell lines were used in this study. |

# Animals and other research organisms

Policy information about studies involving animals; ARRIVE guidelines recommended for reporting animal research, and Sex and Gender in Research

| | |
|---|---|
| Laboratory animals | NOD.Cg-Prkdcscid Il2rgtm1Wjl/SzJ mice (stock 005557), commonly referred to as NSG mice, were purchased from The Jackson Laboratory. Female (6–8-week-old) NSG mice were used in all studies. Mice were co-housed in ventilated cages with wood chip bedding and maintained in a temperature-controlled environment with a 12-h light/dark cycle at the facilities of Weill Cornell Medical College. The temperature of the holding rooms are maintained between 70-74 degrees Fahrenheit. The relative humidity is maintained between 30% – 70%. |
| Wild animals | The study did not involve wild animals. |
| Reporting on sex | Only female mice were included in this study as all mice need to be the same sex to ensure even distribution of variables such as plasma HIV viral load and CD4 T cell count when assigning groups for the engraftment of memory CD8 T cells. Additionally, all mice are of the same sex to ensure no impact of the sex of the mouse on the assessment of function of human immune cells within the experimental conditions. |
| Field-collected samples | The study did not involve sample collection from the field. |
| Ethics oversight | All animal procedures were conducted according to a protocol approved by the Weill Cornell Medical College Institutional Animal Care and Use Committee (protocol 2018-0027). |

Note that full information on the approval of the study protocol must also be provided in the manuscript.

# Clinical data

Policy information about clinical studies

All manuscripts should comply with the ICMJE guidelines for publication of clinical research and a completed CONSORT checklist must be included with all submissions.

| | |
|---|---|
| Clinical trial registration | This manuscript does not report on the findings of any clinical trials. However, the people with HIV-1 who were included in the manuscript were originally enrolled in other clinical trials.<br>eCLEAR (IDs 107, 104, 112, 110, 702, 120 and 209): EudraCT:2015-002234-53, Clinicaltrials.gov: NCT03041012<br>TITAN (ID142): EudraCT: 2018-001165-16, Clinicaltrials.gov: NCT03837756<br>ID9254: EudraCT: 2016-002803-25, Clinicaltrials.gov: NCT02825797 |
| Study protocol | Study protocols are available upon request. |
| Data collection | Participants ID107, ID104, ID112, ID110, ID702, ID120 and ID209 were enrolled in the eCLEAR trial. ID107 continued to be followed longitudinally following completion of the eCLEAR trial until 2025. ID142 was originally enrolled in the TITAN trial, and continued to be followed longitudinally until 2024. Continued ATI for ID107 and ID142 was done under the trial protocol and leukapheresis was done according to the National Committee on Health Research Ethics in Denmark (#1-10-72-38-23). Secondary use of samples from individuals who participated in eCLEAR was approved by the National Committee on Health Research Ethics in Denmark (1-10-72-110-16). ID9254 was enrolled in a different clinical trial, and continued to be monitored longitudinally until 2025. Research blood samples were collected under an observational protocol approved by the Ethics Committee of the Medical Faculty of the University of Cologne (UKK #16-054). |
| Outcomes | Outcomes of clinical trials are not relevant for this manuscript. |

# Plants

Seed stocks

*Report on the source of all seed stocks or other plant material used. If applicable, state the seed stock centre and catalogue number. If plant specimens were collected from the field, describe the collection location, date and sampling procedures.*

Novel plant genotypes

*Describe the methods by which all novel plant genotypes were produced. This includes those generated by transgenic approaches, gene editing, chemical/radiation-based mutagenesis and hybridization. For transgenic lines, describe the transformation method, the number of independent lines analyzed and the generation upon which experiments were performed. For gene-edited lines, describe the editor used, the endogenous sequence targeted for editing, the targeting guide RNA sequence (if applicable) and how the editor was applied.*

Authentication

*Describe any authentication procedures for each seed stock used or novel genotype generated. Describe any experiments used to assess the effect of a mutation and, where applicable, how potential secondary effects (e.g. second site T-DNA insertions, mosiacism, off-target gene editing) were examined.*

# Flow Cytometry

## Plots

Confirm that:

☒ The axis labels state the marker and fluorochrome used (e.g. CD4-FITC).

☒ The axis scales are clearly visible. Include numbers along axes only for bottom left plot of group (a 'group' is an analysis of identical markers).

☒ All plots are contour plots with outliers or pseudocolor plots.

☒ A numerical value for number of cells or percentage (with statistics) is provided.

## Methodology

Sample preparation

Cryopreserved PBMCs were thawed and washed in RPMI supplemented with FBS and P/S. Treatment of samples following thawing and washing depended on the experiment performed (i.e. lymphocyte proliferation assay, AIM assay or spectral flow cytometry intracellular cytokine staining assay), and is described in the Methods section.

Instrument

MACSQuant16 (Miltenyi Biotec) (lymphocyte proliferation assay, AIM assay), MACSQuant Tyto Cell Sorter (Miltenyi Biotec) (sorting of AIM+ cells), 5-laser Sony ID7000 Spectral Analyser (SONY Biotechnologies, San Jose, CA) (spectral flow cytometry), LSRII flow cytometer (AIM assay).

Software

Flow data acquired for the AIM and lymphocyte proliferation assay was analysed using FlowJo 10.10.0 or FlowJo 10.5.0. Spectral flow unmixing was performed using ID7000 Software version 2.0.2 (SONY Biotechnologies, San Jose, CA). All analyses downstream of spectral unmixing were conducted in R version 4.3.2 with RStudio version 2023.12.1.402 using an in-house pipeline built through the assembly of several algorithms (described in the Methods section).

Cell population abundance

200,000 PBMCs were seeded in each well for stimulation in the lymphocyte proliferation assay. 1 x 10^6 PBMCs were seeded in each well for the AIM assay for ID107 and ID142, and at 10 x 10^6 cells/mL in each well for the AIM assay for ID9254. For spectral flow cytometry ICS, the input ranged from 0.8 x 10^6 PBMCs to 1.3 x 10^6 PBMCs.

Gating strategy

In the lymphocyte proliferation assay, cells were gated as follows: Live cells were gated as the negative population in a dead cell stain. Single cells were gated in a SSC-A/SSC-H plot. Lymphocytes were gated in a SSC-A/FSC-A plot. CD3 postive cells were gated in a SSC-A/CD3-A. Single positive CD4 and CD8 cells were gated in a CD8-A/CD4-A. Percentage proliferating cells was gated for both CD4 and CD8 single positive cells as populations dimmer than the undivided peak.
In the AIM assay for ID107 and ID142, cells were gated as follows: Live cells were gated as the negative population in a dead cell stain. Single cells were gated in a SSC-A/SSC-H plot. Lymphocytes were gated in a SSC-A/FSC-A plot. CD3 postive cells were gated in a SSC-A/CD3-A. Single positive CD4 and CD8 cells were gated in a CD8-A/CD4-A. Boolean gating of the three AIMs was then applied to both CD4 and CD8 positive cells to detect double- and triple positive cells.  For the AIM assay for ID9254, the gating strategy is reported in Niessl et al., Nature Medicine, 2020.
For spectral flow cytometry, a detailed description of the gating strategy is given in the Methods section. In brief, cells were pregated in a SSC-A/FSC-A plot to identify cells and remove debris, live cells were gated as the negative population in a dead cell stain, singlets in a FSC-A/FSC-H plot, and lymphocytes in a FSC-A/SSC-A plot. The lymphocyte population was then clustered into main populations using canonical lineage markers (CD3, CD8, CD4, CD16, CD19). Memory CD4 and CD8 T-cell clusters were identified by their expression of CD45RA, CCR7, CD27, and CD95, and cells were gated for the degranulation marker CD107a and the cytokines IFN-g, TNF-a and IL-2.

☒ Tick this box to confirm that a figure exemplifying the gating strategy is provided in the Supplementary Information.

