## [Peer Review File · Nature Immunology]

Autologous neutralising antibodies and polyfunctional T-cells contribute to long-term HIV-1 post-intervention control

Corresponding Author: Dr Katie Fisher

Version 0:

Reviewer comments:

Reviewer #1

(Remarks to the Author)

This study provides an in-depth characterization of two individuals who demonstrated viral control for several years following analytical interruption in the context of their participation in the eCLEAR and TITAN clinical trials. The authors performed a broad set of experiments to investigate the mechanisms associated with durable viral control without ART. They revealed the presence of reactivable intact proviruses in both individuals after ART interruption, underscoring the critical role of effective immune pressure during this period. Consistently, the study presents strong evidence of T cells and antibodies with autologous antiviral capacity. Although the number of individuals studied is a limitation, the thoroughness of the sequential analyses and the consistency of the results palliate this limitation.

One strong aspect of the study is the analysis of the direct inhibitory capacity of antibodies and CD8 T cells (for one of the individuals) against autologous viruses. However, some controls are needed in order to truly appreciate these capacities. Specifically:

- in the case of the antibody neutralizing activities, it is not shown whether these activities differ from those in other people who participate in the trials but did not control infection without ART. For instance from ID104 and ID112 whose cells were used later.

- Moreover, it would be important to know if unrelated IgGs (from other people with HIV) could neutralize the viruses from these individuals.

- In the PDX mouse model experiments a control with cells from an unrelated donor are provided, but the conditions are not really comparable. In the control case, the animals were infected with a lab adapted strain HIV-JRCSF at a high TCID₅₀ (10 000) whereas in the case of participant ID107, the animals were infected with the autologous virus at an unknown TCID₅₀.

The authors mention (Lines 178-179) that (HIV-1-specific T-cells in ID107 and ID142 were primarily polyfunctional, whereas NCs were dominated by a monofunctional HIV-1-specific T-cell response (Fig. 4A)." However, this is not immediately evident from the figure, in particular in the case of the ID104NC whose CD8 Gag-response appeared to be largely composed of cells producing IFN γ +CD107+/-TNF α quite similar to that in ID107. Are these differences quantifiable?

The authors should acknowledge that comments comparing inhibitory potential of antibodies with inhibition by ART are based on estimates.

The frequency of HIV-specific T cells in the ID107 and ID142 controllers appeared to be relatively high at or before ATI in comparison to the non controllers. Did the authors have data regarding ultra-sensitive viral loads determinations or cell-associated viral RNA in these individuals? It is intriguing that whereas the frequency of HIV-specific CD8 T cells appeared to wane in ID107 it remained elevated for ID142 throughout the follow-up, which might suggest low level or localized viral replication in this individual which would explain viral escape later.

Considering the profile of the individuals it is unlikely but I wonder if the authors checked (in particular for ID142) the presence of escape mutations against the bNAbs used in the interventions

The role of autologous neutralizing antibodies or CD8 T cells after ART interruption has been proposed in previous studies (<https://doi.org/10.1126/scitranslmed.abq4490>; <https://insight.jci.org/articles/view/173864>; <https://www.nature.com/articles/s41467-023-44389-3>; <https://doi.org/10.1016/j.chom.2023.06.006>;) and could be discussed

by the authors.

The authors mention that the lack of changes in the magnitude of HIV-specific T cell responses following viral rebound in ID142 was likely related to the presence of CTL escape mutations in this virus (lines 322-323) but most epitopes did not appear to have signs of escape (extended data table 2). Expansion/activation of T cells was not evident either following ATI. This may be related to very localized response to antigens, but perhaps the authors could comment on this.

(Remarks on code availability)

Reviewer #2

(Remarks to the Author)

The manuscript by Fisher and Sogaard investigates the virology and immune response to HIV in two patients that achieved long-term post-treatment control of the virus. The authors pursue measurements of the viral reservoir, viral sequencing, integration sites, autologous neutralizing antibodies and T cell responses. The work is impressive not for the number of patients, which is acknowledged, but for the extraordinary depth of the investigation. The NSG mouse experiment takes cause and effect about as far as one can go in working out the mechanism of control in these patients. The authors are to be commended. There are some areas where the manuscript could be improved, mostly not in methodology but in interpretation and avoiding overstating the conclusions.

1. The conclusion that potent antibodies are a major contributor to control may be an overstatement. In cross-sectional analyses even the circulating virus of "elite neutralizers" is typically resistant to potent and broad antibodies isolated from that patient. The authors show that autologous antibodies can suppress in a QVOA like assay, and that the circulating virus that emerges in D142 is resistant. However, these are very different situations in that the QVOA showed suppression of what is likely single clones of sensitive viruses and in relatively few wells. The circulating virus is going to be dominated by a small species of resistant viruses that arose from a very large population of infected cells in the body, which the authors show. So, while these antibodies exert some selective pressure, we can't really conclude the net effect of aNabs on control. For example, if all the control was due to a potent cellular immune response, regardless of whether the antibody escape mutations predated the loss of control, one might observe the same result. I feel that this conclusion might be moderated in the title, body of the paper, and within the paragraph of caveats within the discussion.

2. The conclusion that immunologic control is lost in D142 because of mutational escape in class-I restricted epitopes isn't well founded. The authors find a few mutations in a minority of epitopes and do not investigate whether that resulted in loss of recognition but rely on citations. There are very good data from several centers now showing that there is no difference in the frequency of such mutations in key epitopes in MHC-matched cohorts with and without immunologic control. It appears that the presence of mutations didn't reduce overall recognition of the virus in D142 given the total HIV-specific CD8+ T cell response didn't decline. Rather recognition of the virus is what is keeping the numbers of HIV-specific cells high. It seems more plausible that the loss of control in this patient is more likely due to a loss of immune function than immune recognition. This loss of function might be detected in measures of CD8 proliferation, CD8-mediated killing or suppression. Given the very large amount of work in the paper I don't feel these are necessary for publication but the cause and effect conclusions for the loss of control should be moderated and receive appropriate caveats.

3. Figure 4a-The degree of polyfunctionality detected doesn't look different between controllers and non-controllers and appears to be driven by frequency. In other words, you are more likely to find cells making a given cytokine that is above threshold if you have a higher frequency response. Given most patients without control have similar frequencies of HIV-specific cells compared to those with control, this experiment should be repeated in larger numbers of patients better matched for CD8+ T cell frequency.

Minor

1. Fig. 1C would be more appropriately placed in supplemental.

(Remarks on code availability)

Reviewer #3

(Remarks to the Author)

The manuscript by Fisher, Garcia et al. describes in-depth characterization of samples from two early ART initiated post intervention controllers (PICs) that did not rebound for > 2.5 years after ATI following bNAb treatment. Intact proviruses were detected in both participants and reservoirs were located in gene deserts. In addition to T cell pressure, autologous neutralizing antibodies (aNAbs) at ART inhibited the virus in viral outgrowth assays in vitro but was unable to neutralize rebound pseudovirus. Further HIV-1 specific CD8 T cells in PICs were poised to respond to antigen and maintain control after ATI. The premise of this study to investigate viral control in PIC is strong since uncovering such mechanisms can help identify strategies for viral clearance. The paper is well written and logical.

Main concerns are related to the very small sample size (n=2) of this study, which makes it incredibly difficult to assess robustness of the analysis without statistical tests. Although multiple timepoints are available for the 2 donors, results from comparisons of 2 PICs and 2 NCs are debatable and seem more in line with observational case findings. Given that the frequency of PIC (10-20%) is higher than PTC (4%), propose the authors expand findings to additional PIC/NC for a robust

comparative analysis.

Both the PICs have HLA-B*44, which has previously been shown to control HIV in 3 different studies (PMID: 23677320, 11309482, 21715491). The presence of mutations in the rebound virus of the one participant that rebounded after 2.5 years is supportive that escape is contributing to loss of viral control. Is the T cell or Ab response able to inhibit virus from non-controllers to rule out that these donors are PICs for other reasons such as a weak virus, or the favorable HLA type.

To investigate HIV-1 specific T cell responses, the authors sort AIM+ T cells and perform scRNA-seq. Based on scRNA-seq gene expression, clusters 1 & 4 were determined as HIV-specific CD8 T cells, with cluster 1 frequency higher in PICs compared to NC pre-ATI. This activated cluster although potentially interesting, is questionable given the small n and the low recovery of cells in 1 of the 4 donors (600 versus >2000). The single cell analysis is quite limited, ADT data is underutilized, which perhaps is not surprising given the small n. Can the phenotype of cluster 1 be defined using the ADT surface markers, to then validate this cluster using flow cytometry or other methods like tetramer binding assays? But really, what is different about the cells in cluster 1 that could explain why it is protective in PICs?

Minor comments

Is it possible to infer glycosylation sites in the sequences of Env and are there differences comparing PIC and NC as shown previously in PTC by others?

CRF01 is not a HIV-1 subtype and please update to the recommended nomenclature. See <https://www.hiv.lanl.gov/components/sequence/HIV/crfd/crfs.comp>

(Remarks on code availability)

I was able to access the data and read the code in the paper. The processed .rds file which contains normalized data, variable features, dimensionality reduction, clustering results, cell type annotations and other metadata and analysis results in one object is missing. Without the .rds file, it is not possible to reproduce the exact results and aforementioned figures.

Decision Letter:

1st Aug 2025

Dear Dr Fisher,

Your Article, "Autologous neutralising antibodies and polyfunctional T-cells maintain long-term HIV-1 post-intervention control" has now been seen by 3 referees. You will see from their comments copied below that while they find your work of considerable potential interest, they have raised quite substantial concerns that must be addressed. In light of these comments, we cannot accept the manuscript for publication, but would be very interested in considering a revised version that addresses these serious concerns.

We hope you will find the referees' comments useful as you decide how to proceed. If you wish to submit a substantially revised manuscript, please bear in mind that we will be reluctant to approach the referees again in the absence of major revisions.

In particular, all the referees highlight the small sample number as a limitation. Therefore, we consider important adding another sample (controller) and some non-controllers for at least some of the assays. This will strengthen the manuscript, as suggested by the reviewers. Please also add a processed .rds file which contains normalized data, variable features, dimensionality reduction, clustering results, cell type annotations and other metadata and analysis results in one object, for the reviewers to assess the code. Please also address the rest of the reviewers' issues.

If you choose to revise your manuscript taking into account all reviewer and editor comments, please highlight all changes in the manuscript text file.

* If you have not done so already please begin to revise your manuscript so that it conforms to our Article format instructions at <http://www.nature.com/ni/authors/index.html>. Refer also to any guidelines provided in this letter.

The Reporting Summary can be found here:

Extended Data figures and tables are online-only (appearing in the online PDF and full-text HTML version of the paper), peer-reviewed display items that provide essential background to the Article but are not included in the printed version of the paper due to space constraints or being of interest only to a few specialists. A maximum of ten Extended Data display items (figures and tables) is typically permitted. When re-submitting your manuscript, please ensure that any supplementary figures and tables that are more critical to the manuscript's conclusions are converted to Extended data to increase these data's visibility.

Link Redacted

If you wish to submit a suitably revised manuscript we would hope to receive it within 6 months. If you cannot send it within this time, please let us know. We will be happy to consider your revision so long as nothing similar has been accepted for publication at Nature Immunology or published elsewhere.

Nature Immunology is committed to improving transparency in authorship. As part of our efforts in this direction, we are now requesting that all authors identified as 'corresponding author' on published papers create and link their Open Researcher and Contributor Identifier (ORCID) with their account on the Manuscript Tracking System (MTS), prior to acceptance. ORCID helps the scientific community achieve unambiguous attribution of all scholarly contributions. You can create and link your ORCID from the home page of the MTS by clicking on 'Modify my Springer Nature account'. For more information please visit www.springernature.com/orcid.

Thank you for the opportunity to review your work.

Sincerely,
Paula

Paula Jauregui, PhD
Senior Editor
Nature Immunology

Referee expertise:

Referee #1: HIV controllers, HIV reservoir

Referee #2: HIV immunology

Referee #3: Sequencing. HIV

Reviewers' Comments:

Reviewer #1 (Remarks to the Author):

This study provides an in-depth characterization of two individuals who demonstrated viral control for several years following analytical interruption in the context of their participation in the eCLEAR and TITAN clinical trials. The authors performed a broad set of experiments to investigate the mechanisms associated with durable viral control without ART. They revealed the presence of reactivable intact proviruses in both individuals after ART interruption, underscoring the critical role of

effective immune pressure during this period. Consistently, the study presents strong evidence of T cells and antibodies with autologous antiviral capacity. Although the number of individuals studied is a limitation, the thoroughness of the sequential analyses and the consistency of the results palliate this limitation.

One strong aspect of the study is the analysis of the direct inhibitory capacity of antibodies and CD8 T cells (for one of the individuals) against autologous viruses. However, some controls are needed in order to truly appreciate these capacities. Specifically:

- in the case of the antibody neutralizing activities, it is not shown whether these activities differ from those in other people who participate in the trials but did not control infection without ART. For instance from ID104 and ID112 whose cells were used later.
- Moreover, it would be important to know if unrelated IgGs (from other people with HIV) could neutralize the viruses from these individuals.
- In the PDX mouse model experiments a control with cells from an unrelated donor are provided, but the conditions are not really comparable. In the control case, the animals were infected with a lab adapted strain HIV-JRCSF at a high TCID₅₀ (10 000) whereas in the case of participant ID107, the animals were infected with the autologous virus at an unknown TCID₅₀.

The authors mention (Lines 178-179) that (HIV-1-specific T-cells in ID107 and ID142 were primarily polyfunctional, whereas NCs were dominated by a monofunctional HIV-1-specific T-cell response (Fig. 4A).” However, this is not immediately evident from the figure, in particular in the case of the ID104NC whose CD8 Gag-response appeared to be largely composed of cells producing IFNG+CD107+/-TNFα quite similar to that in ID107. Are these differences quantifiable?

The authors should acknowledge that comments comparing inhibitory potential of antibodies with inhibition by ART are based on estimates.

The frequency of HIV-specific T cells in the ID107 and ID142 controllers appeared to be relatively high at or before ATI in comparison to the non controllers. Did the authors have data regarding ultra-sensitive viral loads determinations or cell-associated viral RNA in these individuals? It is intriguing that whereas the frequency of HIV-specific CD8 T cells appeared to wane in ID107 it remained elevated for ID142 throughout the follow-up, which might suggest low level or localized viral replication in this individual which would explain viral escape later.

Considering the profile of the individuals it is unlikely but I wonder if the authors checked (in particular for ID142) the presence of escape mutations against the bNAbs used in the interventions

The role of autologous neutralizing antibodies or CD8 T cells after ART interruption has been proposed in previous studies (<https://doi.org/10.1126/scitranslmed.abq4490>; <https://insight.jci.org/articles/view/173864>; <https://www.nature.com/articles/s41467-023-44389-3>; <https://doi.org/10.1016/j.chom.2023.06.006>;) and could be discussed by the authors.

The authors mention that the lack of changes in the magnitude of HIV-specific T cell responses following viral rebound in ID142 was likely related to the presence of CTL escape mutations in this virus (lines 322-323) but most epitopes did not appear to have signs of escape (extended data table 2). Expansion/activation of T cells was not evident either following ATI. This may be related to very localized response to antigens, but perhaps the authors could comment on this.

Reviewer #2 (Remarks to the Author):

The manuscript by Fisher and Sogaard investigates the virology and immune response to HIV in two patients that achieved long-term post-treatment control of the virus. The authors pursue measurements of the viral reservoir, viral sequencing, integration sites, autologous neutralizing antibodies and T cell responses. The work is impressive not for the number of patients, which is acknowledged, but for the extraordinary depth of the investigation. The NSG mouse experiment takes cause and effect about as far as one can go in working out the mechanism of control in these patients. The authors are to be commended. There are some areas where the manuscript could be improved, mostly not in methodology but in interpretation and avoiding overstating the conclusions.

1. The conclusion that potent antibodies are a major contributor to control may be an overstatement. In cross-sectional analyses even the circulating virus of “elite neutralizers” is typically resistant to potent and broad antibodies isolated from that patient. The authors show that autologous antibodies can suppress in a QVOA like assay, and that the circulating virus that emerges in D142 is resistant. However, these are very different situations in that the QVOA showed suppression of what is likely single clones of sensitive viruses and in relatively few wells. The circulating virus is going to be dominated by a small species of resistant viruses that arose from a very large population of infected cells in the body, which the authors show. So, while these antibodies exert some selective pressure, we can't really conclude the net effect of aNAbs on control. For example, if all the control was due to a potent cellular immune response, regardless of whether the antibody escape mutations predated the loss of control, one might observe the same result. I feel that this conclusion might be moderated in the title, body of the paper, and within the paragraph of caveats within the discussion.

2. The conclusion that immunologic control is lost in D142 because of mutational escape in class-I restricted epitopes isn't well founded. The authors find a few mutations in a minority of epitopes and do not investigate whether that resulted in loss of recognition but rely on citations. There are very good data from several centers now showing that there is no difference in the frequency of such mutations in key epitopes in MHC-matched cohorts with and without immunologic control. It appears

that the presence of mutations didn't reduce overall recognition of the virus in D142 given the total HIV-specific CD8+ T cell response didn't decline. Rather recognition of the virus is what is keeping the numbers of HIV-specific cells high. It seems more plausible that the loss of control in this patient is more likely due to a loss of immune function than immune recognition. This loss of function might be detected in measures of CD8 proliferation, CD8-mediated killing or suppression. Given the very large amount of work in the paper I don't feel these are necessary for publication but the cause and effect conclusions for the loss of control should be moderated and receive appropriate caveats.

3. Figure 4a-The degree of polyfunctionality detected doesn't look different between controllers and non-controllers and appears to be driven by frequency. In other words, you are more likely to find cells making a given cytokine that is above threshold if you have a higher frequency response. Given most patients without control have similar frequencies of HIV-specific cells compared to those with control, this experiment should be repeated in larger numbers of patients better matched for CD8+ T cell frequency.

Minor

1. Fig. 1C would be more appropriately placed in supplemental.

Reviewer #3 (Remarks to the Author):

The manuscript by Fisher, Garcia et al. describes in-depth characterization of samples from two early ART initiated post intervention controllers (PICs) that did not rebound for > 2.5 years after ATI following bNAb treatment. Intact proviruses were detected in both participants and reservoirs were located in gene deserts. In addition to T cell pressure, autologous neutralizing antibodies (aNAbs) at ART inhibited the virus in viral outgrowth assays in vitro but was unable to neutralize rebound pseudovirus. Further HIV-1 specific CD8 T cells in PICs were poised to respond to antigen and maintain control after ATI. The premise of this study to investigate viral control in PIC is strong since uncovering such mechanisms can help identify strategies for viral clearance. The paper is well written and logical.

Main concerns are related to the very small sample size (n=2) of this study, which makes it incredibly difficult to assess robustness of the analysis without statistical tests. Although multiple timepoints are available for the 2 donors, results from comparisons of 2 PICs and 2 NCs are debatable and seem more in line with observational case findings. Given that the frequency of PIC (10-20%) is higher than PTC (4%), propose the authors expand findings to additional PIC/NC for a robust comparative analysis.

Both the PICs have HLA-B*44, which has previously been shown to control HIV in 3 different studies (PMID: 23677320, 11309482, 21715491). The presence of mutations in the rebound virus of the one participant that rebounded after 2.5 years is supportive that escape is contributing to loss of viral control. Is the T cell or Ab response able to inhibit virus from non-controllers to rule out that these donors are PICs for other reasons such as a weak virus, or the favorable HLA type.

To investigate HIV-1 specific T cell responses, the authors sort AIM+ T cells and perform scRNA-seq. Based on scRNA-seq gene expression, clusters 1 & 4 were determined as HIV-specific CD8 T cells, with cluster 1 frequency higher in PICs compared to NC pre-ATI. This activated cluster although potentially interesting, is questionable given the small n and the low recovery of cells in 1 of the 4 donors (600 versus >2000). The single cell analysis is quite limited, ADT data is underutilized, which perhaps is not surprising given the small n. Can the phenotype of cluster 1 be defined using the ADT surface markers, to then validate this cluster using flow cytometry or other methods like tetramer binding assays? But really, what is different about the cells in cluster 1 that could explain why it is protective in PICs?

Minor comments

Is it possible to infer glycosylation sites in the sequences of Env and are there differences comparing PIC and NC as shown previously in PTC by others?

CRF01 is not a HIV-1 subtype and please update to the recommended nomenclature. See <https://www.hiv.lanl.gov/components/sequence/HIV/crfd/crfs.comp>

Reviewer #3 (Remarks on code availability):

I was able to access the data and read the code in the paper. The processed .rds file which contains normalized data, variable features, dimensionality reduction, clustering results, cell type annotations and other metadata and analysis results in one object is missing. Without the .rds file, it is not possible to reproduce the exact results and aforementioned figures.

Version 1:

Reviewer comments:

Reviewer #1

(Remarks to the Author)

I thank the authors for the attention paid to the different issues raised during the initial evaluation of their manuscript. I do not have any further comments

(Remarks on code availability)

Reviewer #2

(Remarks to the Author)

10. The conclusion that potent antibodies are a major contributor to control may be an overstatement. In cross-sectional analyses even the circulating virus of “elite neutralizers” is typically resistant to potent and broad antibodies isolated from that patient. The authors show that autologous antibodies can suppress in a QVOA like assay, and that the circulating virus that emerges in D142 is resistant. However, these are very different situations in that the QVOA showed suppression of what is likely single clones of sensitive viruses and in relatively few wells. The circulating virus is going to be dominated by a small species of resistant viruses that arose from a very large population of infected cells in the body, which the authors show. So, while these antibodies exert some selective pressure, we can't really conclude the net effect of aNabs on control. For example, if all the control was due to a potent cellular immune response, regardless of whether the antibody escape mutations predated the loss of control, one might observe the same result. I feel that this conclusion might be moderated in the title, body of the paper, and within the paragraph of caveats within the discussion.

We thank the reviewer for this thoughtful comment. We agree that individuals classified as elite neutralisers, i.e. those whose plasma IgG neutralises a broad panel of heterologous HIV-1 strains, often harbor circulating plasma viruses resistant to their own contemporaneous antibodies (Gray et al., J Virol 2011; Landais et al., PLoS Pathog 2016). This well-known paradox arises from the prolonged virus-antibody co-evolution required for breadth, during which escape from even highly potent antibodies is expected, as suggested by the Reviewer. Thus, in classical elite neutralisers the terms “potency” and “breadth” refer to neutralisation of heterologous reference panels, not of the individual's contemporaneous plasma viruses, a distinction we believe underscores our point rather than contradicts it.

Participants ID107, ID142 and ID9254 differ fundamentally from the “elite neutraliser” phenotype. As demonstrated by our new data on cross-neutralisation (see comment 2), none of these individuals show heterologous cross-neutralisation breadth; autologous IgG purified from these participants fails to neutralise HXB2 Env-pseudotyped viruses, pseudovirus panels representing the global diversity of HIV-1, or viruses from unrelated individuals (Supplementary Figure 4; Supplementary Tables 4-5 of the revised manuscript). Instead, each controller exhibits exceptionally potent neutralising aNAb activity against autologous viral strains, with IC50 values of ~1 µg/mL against pseudoviruses representing their own inducible, replication-competent reservoir viruses. The fact that these participants started ART relatively early during the course of infection means that env diversity is limited, making it more likely that potent aNAb responses can control replication. Moreover, we have used estimates of inhibitory potential to show that in some cases the activity of aNabs is comparable to that of ART regimens and should be able to control replication even in the absence of other immune mechanisms.

These findings reveal a qualitatively different mechanism of control, narrowly directed but strikingly effective, participant-specific autologous neutralising IgG capable of suppressing inducible, infectious virus. We feel that our conclusion, therefore, does not overstate antibody involvement but rather that this story highlights a distinct scenario entirely separate from classical elite neutralisation or broadly neutralising development.

----The authors miss the key points to the comment and have not moderated any of the statements in the paper.

1. The point about elite neutralizers is not to compare them with PICs here. The point is that in the context of the whole reservoir in vivo the virus escapes even in patients with the most robust B cell responses.
2. Testing only a few viruses that arise in a QVOA is inherently different than a vast quasispecies arising in the whole patient. We have no idea what the relative role of these antibodies in control in vivo is. The data here are only suggestive.
3. We have no idea how to contextualize the finding in only 2-3 patients. In ref 30, 15 persons on ART were tested and 8 had very little aNab suppression, but the rest did. So if the authors performed the assay on another 10 PICs would they find all aNab suppressors or possibly no more?

For these reasons the claims in the paper need to be vastly moderated consistent with my original comment. The authors express enthusiasm for their results but don't have the data to support their claims; and moderation of them is the bare minimum for a high impact journal such as this one.

11. The conclusion that immunologic control is lost in D142 because of mutational escape in class-I restricted epitopes isn't well founded. The authors find a few mutations in a minority of epitopes and do not investigate whether that resulted in loss of recognition but rely on citations. There are very good data from several centers now showing that there is no difference in the frequency of such mutations in key epitopes in MHC-matched cohorts with and without immunologic control. It appears that the presence of mutations didn't reduce overall recognition of the virus in D142 given the total HIV-specific CD8+ T cell response didn't decline. Rather recognition of the virus is what is keeping the numbers of HIV-specific cells high. It seems more plausible that the loss of control in this patient is more likely due to a loss of immune function than immune recognition. This loss of function might be detected in measures of CD8 proliferation, CD8-mediated killing or suppression. Given the very large amount of work in the paper I don't feel these are necessary for publication but the cause and effect conclusions for the loss of control should be moderated and receive appropriate caveats.

We agree with the reviewer that it is possible that viral escape could be mediated by a loss of function of HIV-1-specific CD8 T-cells rather than, or in addition to, loss of recognition. As the reviewer suggested, we have included measures of CD8 T-cell function, including proliferation, expression of activation markers, and polyfunctionality (Figure 7 of the revised manuscript). We observed some loss of proliferative capacity of HIV-1-specific CD8 T-cells during the viral rebound, which recovered following ART re-initiation. However, there was no change in the frequency of AIM+ T-cells during the rebound, and no change in the functional profile of the T-cells, as measured by cytokine production. The only change observed by ICS was an increase in the overall frequency of cytokine-producing cells, consistent with the increase in viral load during the

rebound. Altogether, this suggests no major change in the functional capacity of the CD8 T-cells, though we acknowledge that we have not been able to directly assess the suppression capacity of the CD8 T-cells from ID142 during viral rebound. We have included an acknowledgement that viral rebound may be due to a reduced functional capacity of HIV-1-specific CD8 T-cells in lines 684-687 of the revised manuscript.

We have also amended lines 520-521 of the revised manuscript to read 'We hypothesised that the sudden viral rebound observed in ID142 may also harbour evidence of escape from HIV-1-specific T-cell responses', to be more consistent with the fact that our experiments cannot prove escape from CD8 T-cell responses, but can rather infer escape due to signatures of immune escape in the viral sequence. In addition, in line 532 of the revised manuscript, we have amended the sentence to read 'signatures of immune escape', to indicate that while there are some mutations within CTL epitopes that are suggestive of immune escape, the virus may not have entirely escaped from the CD8 T-cell response.

----I don't see data to support that these are "signatures of immune escape" given only one of the 3 references bothered to document whether the polymorphism affected recognition. Even if they were associated with a loss of recognition of the epitope in prior papers, that isn't necessarily the case here unless the mutations are in anchor positions. The problem with this literature is that few bother to test recognition, and even if they do and find a loss they don't mention the other 35 epitopes that are intact or the fact that the mutation isn't different between those with or without control of the virus. The authors should either test for loss of recognition experimentally and objectively interpret, or caveat this that the significance of these mutations is uncertain.

12. Figure 4a-The degree of polyfunctionality detected doesn't look different between controllers and non-controllers and appears to be driven by frequency. In other words, you are more likely to find cells making a given cytokine that is above threshold if you have a higher frequency response. Given most patients without control have similar frequencies of HIV-specific cells compared to those with control, this experiment should be repeated in larger numbers of patients better matched for CD8+ T cell frequency.

As mentioned in comment 4, we have now included additional individuals on suppressive ART in our ICS experiments to allow a more robust comparison of the contribution of polyfunctional memory CD4 and CD8 T-cells to post-intervention control of HIV.

----The new patients added are on ART and so have the same issue of frequency. The question is are the responses found during ATI of PICs different from progressors or elite controllers better matched for HIV-specific CD8+ T cell frequency. The authors should either properly address this experimentally or give it appropriate caveats in the limitations of the study.

(Remarks on code availability)

Reviewer #3

(Remarks to the Author)

11/17/2025

The clear author responses are appreciated. The authors have included additional samples for some of the assays, but the sample size remains the same for the AIM-transcriptomic assays. Addition of even the ATI beginning timepoint for Fig. 5E would have strengthened these findings related to CD8 differentiation. That these specific results were only performed in 2 participants per group and needs additional validation should be clearly stated in the limitations in the discussion. Same thing should be mentioned in the methods for scRNA-seq including sample size and timepoints in Fig. 5 legends.

(Remarks on code availability)

Code availability.

In the instructions_to_reproduce_the_results.html, the user is instructed to run the command "docker load -i reproduce_results/fisher_garcia_frattari_naasz_et_al_2025_docker_image.tar" to load the provided Docker file. Note that the name of the Docker file is incorrect and missing _image, "fisher_garcia_frattari_naasz_et_al_2025_docker.tar".

I could not find the GEO accession ID. Please upload single-cell sequencing data with the appropriate meta data to GEO database and include the accession ID.

Decision Letter:

Our ref: NI-A40549A

4th Dec 2025

Dear Dr. Fisher,

Thank you for submitting your revised manuscript "Autologous neutralising antibodies and polyfunctional T-cells maintain long-term HIV-1 post-intervention control" (NI-A40549A). It has now been seen by the original referees and their comments are below. The reviewers find that the paper has improved in revision, and therefore we'll be happy in principle to publish it in Nature Immunology, pending minor revisions to satisfy the referees' final requests and to comply with our editorial and

formatting guidelines.

We will now perform detailed checks on your paper and will send you a checklist detailing our editorial and formatting requirements in about a week. Please do not upload the final materials and make any revisions until you receive this additional information from us.

If you had not uploaded a Word file for the current version of the manuscript, we will need one before beginning the editing process; please email that to immunology@us.nature.com at your earliest convenience.

Thank you again for your interest in Nature Immunology Please do not hesitate to contact me if you have any questions.

Sincerely,
Paula

Paula Jauregui, PhD
Senior Editor
Nature Immunology

Reviewer #1 (Remarks to the Author):

I thank the authors for the attention paid to the different issues raised during the initial evaluation of their manuscript. I do not have any further comments

Reviewer #2 (Remarks to the Author):

10. The conclusion that potent antibodies are a major contributor to control may be an overstatement. In cross-sectional analyses even the circulating virus of “elite neutralizers” is typically resistant to potent and broad antibodies isolated from that patient. The authors show that autologous antibodies can suppress in a QVOA like assay, and that the circulating virus that emerges in D142 is resistant. However, these are very different situations in that the QVOA showed suppression of what is likely single clones of sensitive viruses and in relatively few wells. The circulating virus is going to be dominated by a small species of resistant viruses that arose from a very large population of infected cells in the body, which the authors show. So, while these antibodies exert some selective pressure, we can't really conclude the net effect of aNabs on control. For example, if all the control was due to a potent cellular immune response, regardless of whether the antibody escape mutations predated the loss of control, one might observe the same result. I feel that this conclusion might be moderated in the title, body of the paper, and within the paragraph of caveats within the discussion.

We thank the reviewer for this thoughtful comment. We agree that individuals classified as elite neutralisers, i.e. those whose plasma IgG neutralises a broad panel of heterologous HIV-1 strains, often harbor circulating plasma viruses resistant to their own contemporaneous antibodies (Gray et al., J Virol 2011; Landais et al., PLoS Pathog 2016). This well-known paradox arises from the prolonged virus-antibody co-evolution required for breadth, during which escape from even highly potent antibodies is expected, as suggested by the Reviewer. Thus, in classical elite neutralisers the terms “potency” and “breadth” refer to neutralisation of heterologous reference panels, not of the individual's contemporaneous plasma viruses, a distinction we believe underscores our point rather than contradicts it.

Participants ID107, ID142 and ID9254 differ fundamentally from the “elite neutraliser” phenotype. As demonstrated by our new data on cross-neutralisation (see comment 2), none of these individuals show heterologous cross-neutralisation breadth; autologous IgG purified from these participants fails to neutralise HXB2 Env–pseudotyped viruses, pseudovirus panels representing the global diversity of HIV-1, or viruses from unrelated individuals (Supplementary Figure 4; Supplementary Tables 4-5 of the revised manuscript). Instead, each controller exhibits exceptionally potent neutralising aNAb activity against autologous viral strains, with IC50 values of ~1 µg/mL against pseudoviruses representing their own inducible, replication-competent reservoir viruses. The fact that these participants started ART relatively early during the course of infection means that env diversity is limited, making it more likely that potent aNAb responses can control replication. Moreover, we have used estimates of inhibitory potential to show that in some cases the activity of aNabs is comparable to that of ART regimens and should be able to control replication even in the absence of other immune mechanisms.

These findings reveal a qualitatively different mechanism of control, narrowly directed but strikingly effective, participant-specific autologous neutralising IgG capable of suppressing inducible, infectious virus. We feel that our conclusion, therefore, does not overstate antibody involvement but rather that this story highlights a distinct scenario entirely separate from classical elite neutralisation or broadly neutralising development.

----The authors miss the key points to the comment and have not moderated any of the statements in the paper.

1. The point about elite neutralizers is not to compare them with PICs here. The point is that in the context of the whole reservoir in vivo the virus escapes even in patients with the most robust B cell responses.
2. Testing only a few viruses that arise in a QVOA is inherently different than a vast quasispecies arising in the whole patient. We have no idea what the relative role of these antibodies in control in vivo is. The data here are only suggestive.
3. We have no idea how to contextualize the finding in only 2-3 patients. In ref 30, 15 persons on ART were tested and 8 had very little aNab suppression, but the rest did. So if the authors performed the assay on another 10 PICs would they find all

aNab suppressors or possibly no more?

For these reasons the claims in the paper need to be vastly moderated consistent with my original comment. The authors express enthusiasm for their results but don't have the data to support their claims; and moderation of them is the bare minimum for a high impact journal such as this one.

11. The conclusion that immunologic control is lost in D142 because of mutational escape in class-I restricted epitopes isn't well founded. The authors find a few mutations in a minority of epitopes and do not investigate whether that resulted in loss of recognition but rely on citations. There are very good data from several centers now showing that there is no difference in the frequency of such mutations in key epitopes in MHC-matched cohorts with and without immunologic control. It appears that the presence of mutations didn't reduce overall recognition of the virus in D142 given the total HIV-specific CD8+ T cell response didn't decline. Rather recognition of the virus is what is keeping the numbers of HIV-specific cells high. It seems more plausible that the loss of control in this patient is more likely due to a loss of immune function than immune recognition. This loss of function might be detected in measures of CD8 proliferation, CD8-mediated killing or suppression. Given the very large amount of work in the paper I don't feel these are necessary for publication but the cause and effect conclusions for the loss of control should be moderated and receive appropriate caveats.

We agree with the reviewer that it is possible that viral escape could be mediated by a loss of function of HIV-1-specific CD8 T-cells rather than, or in addition to, loss of recognition. As the reviewer suggested, we have included measures of CD8 T-cell function, including proliferation, expression of activation markers, and polyfunctionality (Figure 7 of the revised manuscript). We observed some loss of proliferative capacity of HIV-1-specific CD8 T-cells during the viral rebound, which recovered following ART re-initiation. However, there was no change in the frequency of AIM+ T-cells during the rebound, and no change in the functional profile of the T-cells, as measured by cytokine production. The only change observed by ICS was an increase in the overall frequency of cytokine-producing cells, consistent with the increase in viral load during the rebound. Altogether, this suggests no major change in the functional capacity of the CD8 T-cells, though we acknowledge that we have not been able to directly assess the suppression capacity of the CD8 T-cells from ID142 during viral rebound. We have included an acknowledgement that viral rebound may be due to a reduced functional capacity of HIV-1-specific CD8 T-cells in lines 684-687 of the revised manuscript.

We have also amended lines 520-521 of the revised manuscript to read 'We hypothesised that the sudden viral rebound observed in ID142 may also harbour evidence of escape from HIV-1-specific T-cell responses', to be more consistent with the fact that our experiments cannot prove escape from CD8 T-cell responses, but can rather infer escape due to signatures of immune escape in the viral sequence. In addition, in line 532 of the revised manuscript, we have amended the sentence to read 'signatures of immune escape', to indicate that while there are some mutations within CTL epitopes that are suggestive of immune escape, the virus may not have entirely escaped from the CD8 T-cell response.

----I don't see data to support that these are "signatures of immune escape" given only one of the 3 references bothered to document whether the polymorphism affected recognition. Even if they were associated with a loss of recognition of the epitope in prior papers, that isn't necessarily the case here unless the mutations are in anchor positions. The problem with this literature is that few bother to test recognition, and even if they do and find a loss they don't mention the other 35 epitopes that are intact or the fact that the mutation isn't different between those with or without control of the virus. The authors should either test for loss of recognition experimentally and objectively interpret, or caveat this that the significance of these mutations is uncertain.

12. Figure 4a-The degree of polyfunctionality detected doesn't look different between controllers and non-controllers and appears to be driven by frequency. In other words, you are more likely to find cells making a given cytokine that is above threshold if you have a higher frequency response. Given most patients without control have similar frequencies of HIV-specific cells compared to those with control, this experiment should be repeated in larger numbers of patients better matched for CD8+ T cell frequency.

As mentioned in comment 4, we have now included additional individuals on suppressive ART in our ICS experiments to allow a more robust comparison of the contribution of polyfunctional memory CD4 and CD8 T-cells to post-intervention control of HIV.

----The new patients added are on ART and so have the same issue of frequency. The question is are the responses found during ATI of PICs different from progressors or elite controllers better matched for HIV-specific CD8+ T cell frequency. The authors should either properly address this experimentally or give it appropriate caveats in the limitations of the study.

Reviewer #3 (Remarks to the Author):

11/17/2025

The clear author responses are appreciated. The authors have included additional samples for some of the assays, but the sample size remains the same for the AIM-transcriptomic assays. Addition of even the ATI beginning timepoint for Fig. 5E would have strengthened these findings related to CD8 differentiation. That these specific results were only performed in 2 participants per group and needs additional validation should be clearly stated in the limitations in the discussion. Same thing should be mentioned in the methods for scRNA-seq including sample size and timepoints in Fig. 5 legends.

Reviewer #3 (Remarks on code availability):

Code availability.

In the instructions_to_reproduce_the_results.html, the user is instructed to run the command “docker load -i reproduce_results/fisher_garcia_frattari_naasz_et_al_2025_docker_image.tar” to load the provided Docker file. Note that the name of the Docker file is incorrect and missing _image, “fisher_garcia_frattari_naasz_et_al_2025_docker.tar”.

I could not find the GEO accession ID. Please upload single-cell sequencing data with the appropriate meta data to GEO database and include the accession ID.

Editor's comment:

In particular, all the referees highlight the small sample number as a limitation. Therefore, we consider important adding another sample (controller) and some non-controllers for at least some of the assays. This will strengthen the manuscript, as suggested by the reviewers. Please also add a processed .rds file which contains normalized data, variable features, dimensionality reduction, clustering results, cell type annotations and other metadata and analysis results in one object, for the reviewers to assess the code. Please also address the rest of the reviewers' issues.

We thank the Editor for considering our manuscript and for these suggestions, which we believe have greatly improved the manuscript. As suggested, we have included an additional post-intervention controller, ID9254, who has remained off ART following infusions of the bNAbs 3BNC117 and 10-1074 for >7.5 years (ongoing) (trial described in Mendoza et al, Nature, 2018). For this controller, we have been able to include a number of matching virological and immunological analyses such as intact and inducible provirus (Figure 2B and E; Extended Data Figure 1), autologous neutralising antibody responses (Figure 3G), and investigations into HIV-1-specific CD4 and CD8 T-cells (Figure 4; Extended Data Fig. 4C-D). In addition, we have added additional non-controllers and individuals on suppressive ART for some assays, which we have described in the responses to specific reviewers' queries below. All key findings remain unchanged in the revised manuscript but the addition of the new controller and the extra non-controllers has significantly strengthened the data analysis and re-enforced the conclusions.

Reviewers' Comments:

Reviewer #1 (Remarks to the Author):

This study provides an in-depth characterization of two individuals who demonstrated viral control for several years following analytical interruption in the context of their participation in the eCLEAR and TITAN clinical trials. The authors performed a broad set of experiments to investigate the mechanisms associated with durable viral control without ART. They revealed the presence of reactivable intact proviruses in both individuals after ART interruption, underscoring the critical role of effective immune pressure during this period. Consistently, the study presents strong evidence of T cells and antibodies with autologous antiviral capacity. Although the number of individuals studied is a limitation, the thoroughness of the sequential analyses and the consistency of the results palliate this limitation.

One strong aspect of the study is the analysis of the direct inhibitory capacity of antibodies and CD8 T cells (for one of the individuals) against autologous viruses. However, some controls are needed in order to truly appreciate these capacities. Specifically:

1. in the case of the antibody neutralizing activities, it is not shown whether these activities differ from those in other people who participate in the trials but did not control infection without ART. For instance from ID104 and ID112 whose cells were used later.

We thank the Reviewer for this comment. We do agree with the reviewer that comparison of the autologous antibody neutralising activity of our PICs to the specific non-controller individuals who experience a rapid viral rebound following ART interruption included elsewhere in the manuscript is important to truly appreciate the uniqueness of the neutralising activity of aNABs from PICs. Unfortunately, due to more limited sampling of non-controllers compared to PICs (who underwent leukapheresis during the ATI), we are not able to perform the mQVOA, and subsequently the TZMbl neutralisation assays against qVOA-derived *env* variants, using sampling from ID104 and ID112.

Importantly, many individuals do not develop robust aNAB responses. We have previously shown that autologous neutralising antibodies can block outgrowth of a substantial but variable fraction of reservoir viruses in mQVOA assays *ex vivo*, ranging from 0% to 96% in different participants (Bertagnolli et al., PNAS, 2020). Among 14 participants in that study, four have previously participated in a cure trial involving a bNAB, and all four experienced viral rebound within 3–9 weeks following treatment interruption. Although mQVOA experiments in this study were performed ~3 years following the ATI, *env* sequences sourced from the viral rebound were found to be identical to or genetically similar to *env* isolates that were resistant to autologous IgG in these individuals. Our recent studies indicate that in people on long term ART, few qVOA isolates are effectively neutralized by autologous IgG in TZMbl neutralisation assays, with a median of 92% of viruses in each individual being resistant to autologous IgG (McMyn et al., JCI, 2025). Together, these findings indicate that while aNABs can exert measurable pressure on the virus, they are generally insufficient to prevent rebound, which typically comes from the subset of variants that are not neutralised.

We have now included a description of our previous work investigating aNAB responses in individuals on suppressive ART in lines 561-566 of the revised manuscript to allow the distinction of the patterns seen in our PICs to those of other individuals with HIV-1.

2. Moreover, it would be important to know if unrelated IgGs (from other people with HIV) could neutralize the viruses from these individuals.

To directly address this point, we have now conducted neutralisation assays designed to assess the level of cross neutralisation (i.e. against heterologous HIV strains) using IgG purified from ID107 and ID142. Each participant's IgG was tested not only against autologous Env-expressing pseudoviruses, but also against the lab-adapted strain HXB2 and pseudoviruses from the other participant. Remarkably, each participant's IgG potently neutralised their own viruses with IC₅₀ values ~1 µg/mL, but failed to neutralise HXB2 or the other participant's pseudoviruses. These data are shown in Supplementary Figure 4 of the revised manuscript. Furthermore, for ID107 and ID9254 (the new controller), we assessed the neutralisation capacity of purified IgG against panels of pseudoviruses representing the global diversity of HIV-1 Env. We similarly observed no cross-neutralisation capacity of the autologous IgG against these pseudoviruses (shown in Supplementary Tables 3 and 4 of the revised manuscript). These results highlight the extraordinary strain-specific potency of the autologous antibodies, emphasising that these responses are uniquely tailored to the viral quasispecies of each individual and further distinguish post-intervention controllers from non-controllers, whose aNAb responses are insufficient to fully contain viral outgrowth. Cumulatively, these data provide strong mechanistic support for our statement that highly potent autologous neutralising antibodies can directly suppress viral outgrowth *ex vivo* and likely contribute to durable post-intervention control in these rare individuals.

The suggestions to include cross-neutralisation controls and to discuss the broader immune context were invaluable, allowing us to demonstrate the extraordinary specificity and functional potency of these antibodies that directly supports their potential role in maintaining prolonged ART-free remission. We have included a description of our assessments of cross-neutralisation capacity of IgG sourced from our PICs in lines 269-279 of the revised manuscript.

3. In the PDX mouse model experiments a control with cells from an unrelated donor are provided, but the conditions are not really comparable. In the control case, the animals were infected with a lab adapted strain HIV-JRCSF at a high TCID₅₀ (10 000) whereas in the case of participant ID107, the animals were infected with the autologous virus at an unknown TCID₅₀.

We agree with the Reviewer that the experimental conditions between these two PDX models were not completely identical. Specifically, mice engrafted with cells from the unrelated progressor individual were infected with the lab-adapted HIV-1 JR-CSF strain at a defined high titre, whereas those engrafted with cells from participant ID107 were infected with plasma-derived autologous virus of unknown TCID₅₀. We have now revised the discussion (lines 615-639) to clarify this explicitly. The revised text notes that although the inoculum titre for ID107 was unknown, infection produced robust viremia in nearly all mice, with peak viral loads comparable to those in the JR-CSF-infected controls

(>10⁶ copies mL⁻¹ within 1-2 weeks). Importantly, engraftment of autologous mCD8⁺ T cells from ID107 led to a 1,191-fold reduction in plasma viremia relative to mice that did not receive CD8⁺ T cells, and viral loads remained more than 2 logs lower than controls throughout the experiment. We interpret these findings as direct *in vivo* evidence that CD8⁺ T cells from ID107 can suppress replication of their autologous virus. Based on both our published data and broader experience with the PDX model, we believe this represents a notably strong degree of control within this system; however, our revised manuscript now provides balanced context, clearly noting the caveats and acknowledging that “future studies under fully matched infection parameters will be required to rigorously define the magnitude of this effect relative to non-controllers.”

4. The authors mention (Lines 178-179) that (HIV-1-specific T-cells in ID107 and ID142 were primarily polyfunctional, whereas NCs were dominated by a monofunctional HIV-1-specific T-cell response (Fig. 4A).” However, this is not immediately evident from the figure, in particular in the case of the ID104NC whose CD8 Gag-response appeared to be largely composed of cells producing IFNG+CD107+/-TNFa quite similar to that in ID107. Are these differences quantifiable?

We thank the Reviewer for this suggestion to clarify our results assessing the HIV-1-specific T-cell response in our PICs and NCs. To address this, we have quantified the percentage of the memory CD4 and CD8 T-cell population responding to HIV-1 stimulation that is either monofunctional or polyfunctional, and compared this between our now three PICs and six individuals on suppressive ART (Figure 4A-B of the revised manuscript). We have performed this comparison at the on-ART timepoint as we have this timepoint available for all individuals. At this timepoint, all individuals are virologically suppressed, and therefore their T-cell responses are not potentially biased by viremia, which could explain the similar CD8 Gag-response observed in ID104 at the viremic timepoint, as pointed out by the Reviewer. We have also performed this comparison with HIV-1 Gag, Pol and Nef stimulation combined to assess the quality of the total HIV-1-specific response, and each stimulation separately. By performing this analysis, we have confirmed that the PICs do have a higher frequency of polyfunctional CD8 and CD4 memory responses compared to NCs, with the difference in polyfunctional CD4 memory responses being substantially and significantly higher in PICs, suggesting that polyfunctional CD4 memory responses at the time of ART interruption are important in the maintenance of ART-free virological control (Figure 4A and lines 290-307 of the revised manuscript).

We found that each PIC has a strong polyfunctional memory CD8 T-cell response, but to a specific HIV-1 protein (Figure 4B and lines 308-319 of the revised manuscript). Additionally, PICs have a higher frequency of polyfunctional CD4 memory responses to

all stimulations, emphasising the potential important contribution of CD4 memory responses at the pre-ATI timepoint in the maintenance of control of viremia following ART interruption (Figure 4B and lines 320-337 of the revised manuscript). Interestingly, we find that the higher frequency of polyfunctional CD4 memory responses was largely driven by responses to Gag stimulation, which is consistent with previous work suggesting the importance of polyfunctional Gag-responding CD4 T-cells in spontaneous control of HIV-1 (Pereyra et al., J Infectious Diseases, 2008; Kannanganat et al., J Virology, 2007). We have included a discussion of these polyfunctional CD4 T-cell responses in lines 608-614 of the revised manuscript.

We thank the Reviewer for this suggestion to quantify the differences in monofunctional and polyfunctional CD4 and CD8 T-cell responses between NCs and PICs, as we have further identified that CD4 memory T-cell responses at the on-ART/pre-ATI timepoint are important in the maintenance of ART-free control of viremia in PICs.

5. The authors should acknowledge that comments comparing inhibitory potential of antibodies with inhibition by ART are based on estimates.

We acknowledge your point that comparisons between antibody inhibitory potential and ART are based on estimates. Lines 243-244 of the revised manuscript have been adjusted to clarify that IIP values calculated with physiological doses (~10 mg/mL of autologous IgG) are estimates of *in vivo* antibody efficacy.

The text has also been modified in the description of the *In vitro* Pseudoviruses Neutralisation Assay in line 1053 of the revised manuscript.

6. The frequency of HIV-specific T cells in the ID107 and ID142 controllers appeared to be relatively high at or before ATI in comparison to the non controllers. Did the authors have data regarding ultra-sensitive viral loads determinations or cell-associated viral RNA in these individuals ? It is intriguing that whereas the frequency of HIV-specific CD8 T cells appeared to wane in ID107 it remained elevated for ID142 throughout the follow-up, which might suggest low level or localized viral replication in this individual which would explain viral escape later.

We thank the reviewer for this comment, and agree that continued expression of HIV-1 proviruses during ART-free control is intriguing. We do not have ultrasensitive viral loads for any of our PICs, but for ID107 we have been able to amplify partial *env* genomes from the plasma at the timepoint ATI week 281, indicating that there is some expression of proviral genomes in this individual that is being controlled by the immune response. We have included a description of this in lines 216-222 of the revised manuscript, as well as a Supplementary Figure (Supplementary Fig. 2) comparing these partial *env* genomes

with genetically intact proviruses isolated from PBMCs from ID107. Unfortunately, we do not have similar sequences from suppressed timepoints for ID142, though we agree that the increased frequency of HIV-1-specific T-cells in this individual and the eventual rebound could be suggestive of continued expression of HIV-1 proviruses, and potential low-level replication, that may explain the later viral escape from the immune response. Of interest, we also have measurements of the plasma inflammatory markers soluble CD14 and neopterin at timepoints before and after ART interruption for ID107 and ID142, which we have included in Supplementary Figure 1D-E of the revised manuscript. Here we observe for both individuals that the levels of these plasma inflammatory markers remain unchanged after stopping ART, and were generally lower than at the on-ART timepoint, which may indicate that there is limited expression of HIV antigens compared to the on-ART timepoint during ART-free control (described in lines 223-224 of the revised manuscript).

7. Considering the profile of the individuals it is unlikely but I wonder if the authors checked (in particular for ID142) the presence of escape mutations against the bNAbs used in the interventions

We agree with the Reviewer that determining that the persisting reservoir in these PICs is sensitive to the bNAbs used in the interventions is of interest, particularly in the case of ID142 who later experienced viral rebound. For all three PICs, resistance to the interventional bNAbs was assessed by genotypic and/or phenotypic methods at trial enrolment, and this is reported in each of the clinical trials. To further assess the presence of escape mutations for this manuscript, we have additionally used the bNAb-ReP tool (Rawi et al, Scientific Reports, 2019) to genotypically assess sensitivity to each bNAb in all sequences isolated in our study. Using this tool, we found that all sequences for ID107 and ID142, including those sourced from the viral rebound for ID142, were sensitive to the respective bNAbs. For ID9254, we interestingly found that while all sequences were predicted to be sensitive to the bNAb 3BNC117, all sequences were predicted to be resistant to 10-1074 using the bNAb-ReP tool. However, TZMb-I neutralisation assays against Q²VOA-derived replication-competent isolates sourced from ATI week -5, week 12 (reported in Mendoza et al, Nature, 2018) and week 102 indicated that inducible infectious viruses persisting during the ATI are sensitive to both 10-1074 and 3BNC117 (Supplementary Table 2 of the revised manuscript). This suggests that proviral sequences that would likely contribute to viral rebound in ID9254 were sensitive to both bNAbs. We have modified the text within our revised manuscript to reflect the bNAb sensitivity of the proviral reservoir in lines 128-131 and 490-493, as well as in Supplementary Table 1, of the revised manuscript. We have also included an additional Supplementary Table (Supplementary Table 2) to report the IC₅₀ and IC₈₀ values of all Q²VOA-derived viral isolates for ID9254 against 3BNC117 and 10-1074.

8. The role of autologous neutralizing antibodies or CD8 T cells after ART interruption has been proposed in previous studies (<https://doi.org/10.1126/scitranslmed.abq4490>; <https://insight.jci.org/articles/view/173864>; <https://www.nature.com/articles/s41467-023-44389-3>; <https://doi.org/10.1016/j.chom.2023.06.006>;) and could be discussed by the authors.

We thank the reviewer for their suggestion to add more context to the role of autologous neutralising antibodies and CD8+ T-cells after ART interruption in the context of other studies. We have included discussions of the role of both aNAbs and CD8 T-cells in post-treatment control in lines 104-105 of the revised manuscript, including Li et al., JCI Insight, 2024 and Molinos-Albert et al., Nature Communications, 2022, as suggested by the Reviewer. Additionally, we have modified the manuscript to include additional discussions of the importance of both aNAbs and CD8 T-cells in control of viremia following ART interruption. In lines 558-559 of the revised manuscript, we acknowledge the role of aNAbs in post-treatment control of HIV-1 in the absence of any therapeutic treatment, as shown by Molinos-Albert et al., Nature Communications, 2022 and Esmailzadeh et al., Sci Trans Med, 2023. In lines 605-608 of the revised manuscript, we discuss the role of virus-specific CD8 T cells in the maintenance of post-treatment control of viremia in SIV infection, as discussed in Passeas et al., Nature Communications, 2024. In lines 643-647 of the revised manuscript, we discuss the importance of the development of aNAb responses during ART suppression that can contribute to control of viremia following ART interruption, as shown by Esmailzadeh et al., Sci Trans Med, 2023 and Whitehill et al., J Clin Invest, 2024.

9. The authors mention that the lack of changes in the magnitude of HIV-specific T cell responses following viral rebound in ID142 was likely related to the presence of CTL escape mutations in this virus (lines 322-323) but most epitopes did not appear to have signs of escape (extended data table 2). Expansion/activation of T cells was not evident either following ATI. this may be related to very localized response to antigens, but perhaps the authors could comment on this.

We agree with the Reviewer that not all HLA-matched CTL epitopes were mutated in our study. However, the identification of mutations within these CTL epitopes is likely indicative of immune pressure on these regions, potentially contributing to the emergence of escape mutations within these epitopes. Furthermore, it is well-known that even during acute HIV-1 infection, in the absence of ART, that not all CTL epitopes will develop escape mutations, and that the development of escape mutations within specific epitopes occurs at differing rates (Jones et al., JEM, 2004; Ganusov et al., J Virology, 2011; Sunshine et

al., J Virology, 2015; Brumme et al., J Virology, 2008). This is mostly related to the cost of the mutation to viral fitness, as well as differing rates of recognition loss of mutations. We believe that it is possible that if viremia had continued in the absence of ART, additional mutations within CTL epitopes would have emerged as the viral load continued to increase exponentially. We therefore speculate that the changes within HLA-matched CTL epitopes that we have observed here likely would have led to some loss of recognition, that in combination with the loss of effectiveness of the nAb response has allowed enough viral replication to occur for additional escape mutations to develop and eventual viral escape. We have included an acknowledgement of the fact that not all HLA-matched CTL epitopes were mutated in lines 680-682 of the revised manuscript.

Further, we have included an acknowledgement in lines 664-666 and in lines 682-687 of the revised manuscript that the viral rebound could have occurred in an anatomic location other than the blood, leading to viral rebound.

Reviewer #2 (Remarks to the Author):

The manuscript by Fisher and Sogaard investigates the virology and immune response to HIV in two patients that achieved long-term post-treatment control of the virus. The authors pursue measurements of the viral reservoir, viral sequencing, integration sites, autologous neutralizing antibodies and T cell responses. The work is impressive not for the number of patients, which is acknowledged, but for the extraordinary depth of the investigation. The NSG mouse experiment takes cause and effect about as far as one can go in working out the mechanism of control in these patients. The authors are to be commended. There are some areas where the manuscript could be improved, mostly not in methodology but in interpretation and avoiding overstating the conclusions.

10. The conclusion that potent antibodies are a major contributor to control may be an overstatement. In cross-sectional analyses even the circulating virus of “elite neutralizers” is typically resistant to potent and broad antibodies isolated from that patient. The authors show that autologous antibodies can suppress in a QVOA like assay, and that the circulating virus that emerges in D142 is resistant. However, these are very different situations in that the QVOA showed suppression of what is likely single clones of sensitive viruses and in relatively few wells. The circulating virus is going to be dominated by a small species of resistant viruses that arose from a very large population of infected cells in the body, which the authors show. So, while these antibodies exert some selective pressure, we can’t really conclude the net effect of aNabs on control. For example, if all the control was due to a potent cellular immune response, regardless of whether the antibody escape mutations

predated the loss of control, one might observe the same result. I feel that this conclusion might be moderated in the title, body of the paper, and within the paragraph of caveats within the discussion.

We thank the reviewer for this thoughtful comment. We agree that individuals classified as elite neutralisers, i.e. those whose plasma IgG neutralises a broad panel of heterologous HIV-1 strains, often harbor circulating plasma viruses resistant to their own contemporaneous antibodies (Gray et al., J Virol 2011; Landais et al., PLoS Pathog 2016). This well-known paradox arises from the prolonged virus-antibody co-evolution required for breadth, during which escape from even highly potent antibodies is expected, as suggested by the Reviewer. Thus, in classical elite neutralisers the terms “potency” and “breadth” refer to neutralisation of heterologous reference panels, not of the individual’s contemporaneous plasma viruses, a distinction we believe underscores our point rather than contradicts it.

Participants ID107, ID142 and ID9254 differ fundamentally from the “elite neutraliser” phenotype. As demonstrated by our new data on cross-neutralisation (see comment 2), none of these individuals show heterologous cross-neutralisation breadth; autologous IgG purified from these participants fails to neutralise HXB2 Env–pseudotyped viruses, pseudovirus panels representing the global diversity of HIV-1, or viruses from unrelated individuals (Supplementary Figure 4; Supplementary Tables 4-5 of the revised manuscript). Instead, each controller exhibits exceptionally potent neutralising aNAb activity against autologous viral strains, with IC₅₀ values of ~1 µg/mL against pseudoviruses representing their own inducible, replication-competent reservoir viruses. The fact that these participants started ART relatively early during the course of infection means that *env* diversity is limited, making it more likely that potent aNAb responses can control replication. Moreover, we have used estimates of inhibitory potential to show that in some cases the activity of aNAbs is comparable to that of ART regimens and should be able to control replication even in the absence of other immune mechanisms.

These findings reveal a qualitatively different mechanism of control, narrowly directed but strikingly effective, participant-specific autologous neutralising IgG capable of suppressing inducible, infectious virus. We feel that our conclusion, therefore, does not overstate antibody involvement but rather that this story highlights a distinct scenario entirely separate from classical elite neutralisation or broadly neutralising development.

11. The conclusion that immunologic control is lost in D142 because of mutational escape in class-I restricted epitopes isn’t well founded. The authors find a few mutations in a minority of epitopes and do not investigate whether that resulted in loss of recognition but rely on citations. There are very good data from several centers now showing that there is no difference in the frequency of such mutations

in key epitopes in MHC-matched cohorts with and without immunologic control. It appears that the presence of mutations didn't reduce overall recognition of the virus in D142 given the total HIV-specific CD8+ T cell response didn't decline. Rather recognition of the virus is what is keeping the numbers of HIV-specific cells high. It seems more plausible that the loss of control in this patient is more likely due to a loss of immune function than immune recognition. This loss of function might be detected in measures of CD8 proliferation, CD8-mediated killing or suppression. Given the very large amount of work in the paper I don't feel these are necessary for publication but the cause and effect conclusions for the loss of control should be moderated and receive appropriate caveats.

We agree with the reviewer that it is possible that viral escape could be mediated by a loss of function of HIV-1-specific CD8 T-cells rather than, or in addition to, loss of recognition. As the reviewer suggested, we have included measures of CD8 T-cell function, including proliferation, expression of activation markers, and polyfunctionality (Figure 7 of the revised manuscript). We observed some loss of proliferative capacity of HIV-1-specific CD8 T-cells during the viral rebound, which recovered following ART re-initiation. However, there was no change in the frequency of AIM+ T-cells during the rebound, and no change in the functional profile of the T-cells, as measured by cytokine production. The only change observed by ICS was an increase in the overall frequency of cytokine-producing cells, consistent with the increase in viral load during the rebound. Altogether, this suggests no major change in the functional capacity of the CD8 T-cells, though we acknowledge that we have not been able to directly assess the suppression capacity of the CD8 T-cells from ID142 during viral rebound. We have included an acknowledgement that viral rebound may be due to a reduced functional capacity of HIV-1-specific CD8 T-cells in lines 684-687 of the revised manuscript.

We have also amended lines 520-521 of the revised manuscript to read 'We hypothesised that the sudden viral rebound observed in ID142 may also harbour evidence of escape from HIV-1-specific T-cell responses', to be more consistent with the fact that our experiments cannot prove escape from CD8 T-cell responses, but can rather infer escape due to signatures of immune escape in the viral sequence. In addition, in line 532 of the revised manuscript, we have amended the sentence to read 'signatures of immune escape', to indicate that while there are some mutations within CTL epitopes that are suggestive of immune escape, the virus may not have entirely escaped from the CD8 T-cell response.

12. Figure 4a-The degree of polyfunctionality detected doesn't look different between controllers and non-controllers and appears to be driven by frequency. In other words, you are more likely to find cells making a given cytokine that is above threshold if you have a higher frequency response. Given most patients without

control have similar frequencies of HIV-specific cells compared to those with control, this experiment should be repeated in larger numbers of patients better matched for CD8+ T cell frequency.

As mentioned in comment 4, we have now included additional individuals on suppressive ART in our ICS experiments to allow a more robust comparison of the contribution of polyfunctional memory CD4 and CD8 T-cells to post-intervention control of HIV.

Minor

13. Fig. 1C would be more appropriately placed in supplemental.

We thank the Reviewer for this suggestion. However, given the large number of assays run at many timepoints for three PICs in this manuscript, we believe that it is important for the reader to be oriented to which assay was performed at what timepoint, early in the manuscript. Therefore, we would prefer to keep Figure 1C (now Figure 1D in the revised manuscript) within the main text.

Reviewer #3 (Remarks to the Author):

The manuscript by Fisher, Garcia et al. describes in-depth characterization of samples from two early ART initiated post intervention controllers (PICs) that did not rebound for > 2.5 years after ATI following bNAb treatment. Intact proviruses were detected in both participants and reservoirs were located in gene deserts. In addition to T cell pressure, autologous neutralizing antibodies (aNAbs) at ART inhibited the virus in viral outgrowth assays *in vitro* but was unable to neutralize rebound pseudovirus. Further HIV-1 specific CD8 T cells in PICs were poised to respond to antigen and maintain control after ATI. The premise of this study to investigate viral control in PIC is strong since uncovering such mechanisms can help identify strategies for viral clearance. The paper is well written and logical.

14. Main concerns are related to the very small sample size (n=2) of this study, which makes it incredibly difficult to assess robustness of the analysis without statistical tests. Although multiple timepoints are available for the 2 donors, results from comparisons of 2 PICs and 2 NCs are debatable and seem more in line with observational case findings. Given that the frequency of PIC (10-20%) is higher than PTC (4%), propose the authors expand findings to additional PIC/NC for a robust comparative analysis.

We thank the Reviewer for this comment, and we agree that increasing the number of PICs and NCs in this manuscript would increase our ability to perform robust statistical comparisons. As mentioned in our response to the Editor, we have included analyses from one additional PIC and additional controls to the revised manuscript.

15. Both the PICs have HLA-B*44, which has previously been shown to control HIV in 3 different studies (PMID: 23677320, 11309482, 21715491). The presence of mutations in the rebound virus of the one participant that rebounded after 2.5 years is supportive that escape is contributing to loss of viral control. Is the T cell or Ab response able to inhibit virus from non-controllers to rule out that these donors are PICs for other reasons such as a weak virus, or the favorable HLA type.

We thank the Reviewer for this comment, and we acknowledge that ID107 and ID142, as well as the additional PIC, ID9254, have HLA-B*44. However, the non-controllers that were used for many of the comparative analyses, ID104 and ID112, also have HLA-B*44. This highlights that HLA type alone cannot explain or predict PICs. We have included the HLA types of the non-controller individuals in Supplementary Table 6 of the revised manuscript.

To further exclude that control of viremia is not associated with a favourable HLA type, we have investigated the suppression capacity of CD8 T-cells from ID107 against CD4 T-cells infected *in vitro* with the HIV-1 lab strain HXB2 (see below). We found no evidence of an enhanced response compared to six individuals from the eCLEAR study who did not control following ART interruption, including ID104 and ID112. This is consistent with the CD8 T-cell suppressive capacity of the VISCONTI cohort of post-treatment controllers, who did not have CD8 T-cells with an enhanced suppressive capacity against infection of autologous CD4 T-cells with a lab-strain HIV virus compared to viremic and ART-suppressed individuals with HIV-1 (Sáez-Ciri3n et al., PLoS Pathogens, 2014). Furthermore, we have also demonstrated that IgG from our PICs is not able to cross-neutralise unrelated viral stains, as mentioned in comment 2.

Suppression of HXB2 infection by ID107-derived CD8 T-cells. CD4 T-cells from ID107 (from 7 timepoints before and after ART interruption) and 6 individuals from eCLEAR who did not control following ART interruption (from the pre-ART, pre-ATI and rebound timepoints) were infected with the lab-adapted virus HXB2, and viral suppression by participant-derived CD8 T-cells was assessed compared to infection in the absence of CD8 T-cells. CD8 T-cells were cultured with CD4 T-cells at a (A) 1:1 ratio or (B) 1:10 ratio. % inhibition was calculated based on p24 ELISA from cultures with and without CD8 T-cells.

Furthermore, we find it extremely unlikely that these PICs may be controlling viral replication due to a “weak” virus. All three PICs harboured a reservoir of infectious proviruses that was readily inducible, as demonstrated by qVOA experiments. Importantly, all three PICs had high viral loads prior to initiating ART, indicating that a productive infection occurred. Finally, in the PDX mouse experiment, spontaneous viral rebound occurred in two mice engrafted with memory CD4 T-cells from ID107, indicating that viral rebound could occur in the absence of immune responses and that the virus is not particularly weak. We have included an acknowledgement of the high pre-ART viral loads, indicating that the virus is not weak, in lines 125-127 and 540-543 of the revised manuscript.

16. To investigate HIV-1 specific T cell responses, the authors sort AIM+ T cells and perform scRNA-seq. Based on scRNA-seq gene expression, clusters 1 & 4 were

determined as HIV-specific CD8 T cells, with cluster 1 frequency higher in PICs compared to NC pre-ATI. (a) This activated cluster although potentially interesting, is questionable given the small n and the low recovery of cells in 1 of the 4 donors (600 versus >2000). (b) The single cell analysis is quite limited, ADT data is underutilized, which perhaps is not surprising given the small n. Can the phenotype of cluster 1 be defined using the ADT surface markers, to then validate this cluster using flow cytometry or other methods like tetramer binding assays? (c) But really, what is different about the cells in cluster 1 that could explain why it is protective in PICs?

(a) We thank the reviewer for raising this point regarding whether cluster 1 and its differential abundance in PICs reflect a biologically meaningful result rather than a computational artefact. To rule out potential analytical biases, we re-ran the analysis after optimising steps that could influence clustering outcomes. First, we applied stricter singlet selection, retaining only cells classified as singlets in three independent rounds. Then, we reintroduced HLA genes that had been excluded initially due to concerns that differential HLA gene expression across individuals might bias clustering, which we later confirmed were unfounded. These changes slightly affected the number of cells within individual clusters (and thus the numbering, with the original cluster 4 now corresponding to cluster 3), but the definition of the activated cluster was further consolidated. The overall structure and the higher abundance of cluster 1 in PICs persisted.

Several observations support the robustness of cluster 1:

- Cluster 1 was the largest cluster (7,278 cells; 17.8% of total).
- After harmony-based batch correction, cluster 1 was detected in all samples despite variation in total number of cells. Specifically, cluster 1 frequencies ranged from 1-38% across samples and 6-27% across donors, showing no correlation with total cell yield and thus excluding artifacts related to uneven recovery.

We acknowledge that the limited number of donors and the lower number of cells from one NC introduce uncertainty. Due to limited sampling from NCs as mentioned in comment 1, we do not have additional single-cell sequenced AIM-sorted samples, and comparable AIM-sorted HIV-1-specific CD8 T-cell datasets are not available for integration or joint analysis of new donors, which highlights the novelty of our work. However, to assess the impact of sampling variability, we performed 100 bootstrapped scanpro runs, each downsampled (without replacement) to the smallest sample. In all runs, cluster 1 was more abundant in PICs, reaching statistical significance (adjusted $p < 0.05$) in 89%. The consistent directionality across all iterations confirms that the bootstrap behaved as intended, introducing expected stochastic variability while validating the robustness of the observed enrichment. Details are provided in the Methods (lines 1391-1408 of the revised manuscript).

(b) We acknowledge the reviewer's comments regarding the limited extent of the single-cell analysis and the underutilization of ADT data. Given the wide range of data types integrated in the manuscript, we focused on the most robust results – specifically, the CD8 compartment, cluster 1, its differential abundance, and the associated TCR clonotype analysis – which provide complementary insights to the other T-cell response assays presented (ICS, AIM, proliferation, and the mouse xenograft model). In line with this focus, we initially excluded the ADT data, which were limited to a subset of samples obtained exclusively from PICs (detailed in Supplementary File 1). Consequently, comprehensive analyses such as multimodal weighted nearest-neighbor clustering or differential surface marker testing between PICs and NCs were not feasible. However, we welcome the reviewer's interest in surface protein expression and have therefore added a PIC-based surface protein profiling of the transcriptionally defined clusters in lines 372-377, Fig. 5D and Extended Fig. 5B-C of the revised manuscript.

This phenotypic profile should nevertheless be interpreted with caution when extrapolating to NCs. Furthermore, the 18-hour peptide stimulation used for the AIM assay likely alters the surface marker expression, limiting comparability with other assays, such tetramer-sorted cells at rest or ICS after 6-hour stimulation. Each of these assays provides a distinct characterization of T-cell responses, and they likely capture different subsets with distinct functional profiles. However, importantly, our single-cell analysis reveals a distinct transcriptional program characteristic of effector cells capable of both cytokine production and proliferation – features that may also be shared by cells with differing surface protein profiles.

(c) We thank the reviewer for drawing our attention to the fact that the key features of cluster 1 may not have been fully conveyed. We believe that two key features make cluster 1 particularly noteworthy:

1. A transcriptional signature indicative of activation and differentiation, suggesting that these cells have responded to HIV, exerted effector function, and remain poised to proliferate. We propose that this state represents a key element of a sustained antiviral T-cell response.
2. The timing of its differential abundance. Although cluster 1 is present in both PICs and NCs, in PICs it is already detectable at pre-ATI following stimulation, i.e., in the absence of viral replication. This suggests that PICs maintain a population of cells that can be rapidly reactivated upon viral rebound, enabling PICs to control replication before viral loads rise and immune dysfunction ensues.

To further support point 1, we added differential gene expression and gene ontology enrichment analyses comparing cluster 1 with two other key clusters with high virus-specific CD8 activation module scores – cluster 3 (early effector) and cluster 10 (proliferating) – showing that cluster 1 encompasses transcriptional features of both. This can be found in lines 393-406 and Fig. 5F-G of the revised manuscript. To address point

2, we refined the concluding statement of the corresponding Results section in lines 408-410 of the revised manuscript to better emphasize this interpretation.

Minor comments

17. Is it possible to infer glycosylation sites in the sequences of Env and are there differences comparing PIC and NC as shown previously in PTC by others?

We thank the Reviewer for this comment, and agree that this could be an intriguing explanation for the difference in control of viremia between PICs and NCs (see figure below). We have compared the mean number of N-glycosylation sites for proviral *env* sequences for ID107, ID142 and ID9254 to those of 13 individuals from the TITAN trial (Gunst et al., *Nature Medicine*, 2023) who did not receive bNABs and did not control viremia following ART interruption, and we found no difference in the mean number of N-glycosylation sites between these groups. This suggests that major differences in N-glycosylation sites cannot explain control of viremia by our PICs. Of note, we did not include eCLEAR ID104 and ID112 in this analysis because we do not have proviral *env* sequences for these participants that could be compared to the proviral *env* sequences we have for our PICs.

18. CRF01 is not a HIV-1 subtype and please update to the recommended nomenclature.

See <https://www.hiv.lanl.gov/components/sequence/HIV/crfdb/crfs.comp>

We have changed Supplementary Table 1 and Supplementary Table 6 of the revised manuscript to refer to ‘HIV-1 subtype/circulating recombinant form’ to be consistent with the recommended nomenclature.

Reviewer #3 (Remarks on code availability):

19. I was able to access the data and read the code in the paper. The processed .rds file which contains normalized data, variable features, dimensionality reduction, clustering results, cell type annotations and other metadata and analysis results in one object is missing. Without the .rds file, it is not possible to reproduce the exact results and aforementioned figures.

We are grateful to the reviewer for raising this important point regarding reproducibility. In response, we carefully reviewed the single-cell analysis scripts and ensured that all steps involving random number generation (e.g. doublet detection) are now appropriately seeded. To enable full reproducibility from the raw data, we have implemented a Docker container that can be rebuilt from the provided Dockerfile and is also distributed as a pre-built image, available to the reviewer for peer review and to other researchers for download at the time of publication. In addition, we are providing the Seurat object as an .rds file containing the required features, along with a README file describing the structure and content of the object. The Docker and .rds file can be found for peer review using the following link: https://share.genome.au.dk/IgAnmb_2Mw0.